# Tuning-Free One-Class Discriminant Learning for Tabular Anomaly Detection

**Xuan-Ha Nguyen** [1]  **Vu Nguyen Thai Duong** [2 3]  **Van-Hoi Nguyen** [2 3]  **Kim-Hung Le** [2 3]  **Nhien-An Le-Khac** [1]

## Abstract

Anomaly detection (AD) on real-world tabular data is challenged by diverse anomaly types, scarce labels, and high sensitivity to data-specific hyperparameter tuning. A central difficulty is that different anomaly types favor opposing representations: *compactness*—tightening the normal class to expose pointwise deviations—and *structure preservation*—retaining cluster and cross-feature relationships. Existing one-class detectors capture only one of these or balance them via hyperparameters, which is problematic without validation labels. We propose **Discriminant Vector Machine for Anomaly Detection (DVM-AD)**, a closed-form one-class method derived from discriminant analysis that captures both behaviors under a single fixed configuration. From a deterministic reference point built from the training data, DVM-AD derives a bounded ratio whose two ends correspond to compressive and structure-preserving directions. This bound enables selecting directions from both ends simultaneously without dataset-specific tuning. In addition, a Moore-Penrose pseudo-inverse keeps the method well-posed under high-dimensional or rank-deficient settings, and test samples are scored by nearest-neighbor distance in the discriminant space, normalized for thresholding-ready use. Across 47 ADBench tabular datasets and 10 NLP/CV embedding benchmarks against 28 baselines, DVM-AD achieves the best average AUROC (89.65%, average rank 2.98) on tabular datasets and remains top-ranked across four anomaly types and on embedding tasks (average rank 1.60, AUROC 72.68%).

[1]University College Dublin, Dublin City, Ireland [2]University of Information Technology, Ho Chi Minh City, Vietnam [3]Vietnam National University, Ho Chi Minh City, Vietnam. Correspondence to: Xuan-Ha Nguyen <xuan.h.nguyen@ucdconnect.ie>, Nhien-An Le-Khac <an.lekhac@ucd.ie>.

*Proceedings of the 43rd International Conference on Machine Learning*, Seoul, South Korea. PMLR 306, 2026. Copyright 2026 by the author(s).

## 1. Introduction

Anomaly detection (AD) aims to identify observations that deviate from the nominal data distribution (Marques et al., 2023). In many real-world applications, data are represented in tabular form (e.g., financial transactions, network logs, or patient records), where anomalies may indicate fraud, intrusions, or rare clinical conditions. In practice, normal data are abundant, while anomalies are typically scarce, costly to label, and arise in heterogeneous forms: *local anomalies* (samples that deviate from their immediate neighborhood), *global anomalies* (samples lying in low-density regions of the overall distribution), *cluster anomalies* (small coherent groups shifted from normal clusters), and *dependency anomalies* (samples that violate cross-feature relationships while remaining individually plausible) (Thimonier et al., 2024b; Yin et al., 2024; Han et al., 2022); see Appendix D.3. As a result, many deployments operate in the one-class setting, training on normal data but requiring robustness to multiple anomaly types at test time.

Despite decades of progress, one-class AD on tabular data still lacks a single representation strategy that fits all anomaly (Bouman et al., 2024; Han et al., 2022). A central obstacle is a trade-off between compactness and structure-preserving (Cordier et al., 2022; Perera et al., 2021). Compactness-focused methods such as KNFST (Bodesheim et al., 2013) and PMKFN (Arashloo, 2020) learn directions that collapse normal samples toward a single point: this amplifies subtle pointwise deviations and benefits local and global anomalies, but destroys the cluster geometry and cross-feature relationships needed to detect cluster and dependency anomalies. Structure-preserving methods such as PCA (Shyu et al., 2003) retain the principal variance of normal data, which helps detect cluster and dependency anomalies, but they are insensitive to low-variance subspaces where local anomalies typically hide. These two behaviors are not mutually exclusive in principle, but existing one-class formulations optimize only one side of this trade-off or rely on hyperparameters to balance them (Hojjati & Armanfard, 2023; Perera & Patel, 2019; Nguyen & Vien, 2018), which is problematic in the one-class setting where we lack validation labels for tuning (Herurkar et al., 2025; Ding et al., 2022).

The empirical consequence is visible in our benchmark: no

baseline ranks at the top across all four anomaly types, for example, KNFST attains average rank 12.68 on dependency anomalies (Figure 4c), and PCA attains average rank 12.06 on local anomalies (Figure 4d). This leaves an important gap: a detector that can balance compactness and structure under a fixed configuration across various anomaly types and datasets, without data-specific hyperparameter tuning.

A natural way to balance this trade-off is through Fisher-style discriminant learning (Zhao et al., 2024; Ghojogh et al., 2019) to learn projection directions which balance a compactness term (within-class scatter) against a structure/separation term (between/total scatter). For one-class anomaly detection, this idea is particularly appealing: compressive discriminant directions can concentrate on tightening normal samples and amplify subtle deviations, while structure-preserving directions can retain neighborhood relations and cross-feature geometry. However, existing one-class discriminant methods only emphasize one side of this trade-off (Arashloo, 2020; Dufrenois, 2014; Hoffmann, 2007). This motivates a novel discriminant learning view for one-class AD: instead of choosing either compact or structure-preserving modes as in existing methods, we aim to construct a single discriminant space that intentionally spans both behaviors.

In this work, we propose **Discriminant Vector Machine for Anomaly Detection (DVM-AD)**, a closed-form one-class discriminant learning method that addresses the compactness-structure trade-off under a fixed configuration. DVM-AD first introduces a single deterministic proxy reference point constructed from the normal training data, which makes one-class discriminant geometry well-defined without anomaly labels. From this geometry, DVM-AD analyzes an inverse-scatter ratio whose values are provably bounded in $[0, 1]$ (Theorem 4.1). This bound is the key technical enabler. The lower end of the bounded spectrum corresponds to compressive directions that tighten the normal class, while the upper end corresponds to structure-preserving directions that retain cluster and cross-feature geometry. Crucially, the bound makes it possible to select directions from both ends simultaneously, rather than having to trade one for the other.

Beyond the core formulation, two practical properties make DVM-AD deployable as-is. To remain well-posed when scatter matrices are singular or rank-deficient (common for high-dimensional or redundant tabular data), DVM-AD solves the resulting eigenproblem using a Moore-Penrose pseudo-inverse formulation (Ali & Chaudhuri, 2022; Nikovski & Byadarhaly, 2016) rather than tuned regularization. At inference, scores are computed as nearest-neighbor distances in the learned space and normalized using the training geometry, yielding values in $[0, 1]$ that simplify thresholding. DVM-AD requires one pass to accu-

mulate scatter statistics and one eigen-decomposition in $d$ dimensions; no iterative training or validation labels, and no dataset-specific hyperparameter tuning. In summary, the main contributions are listed as follows:

- **DVM-AD algorithm**: We introduce a closed-form one-class discriminant learning method for tabular anomaly detection that addresses the compactness-structure trade-off under a fixed configuration without data-specific hyperparameter tuning.

- **Two-tailed bounded-spectrum representation with rank robustness**: We prove that an inverse-scatter ratio constructed from the proxy-anchored geometry has eigenvalues bounded in $[0, 1]$ (Theorem 4.1), allowing us to select discriminant directions from both ends of the spectrum simultaneously - combining compressive directions that tighten the normal class with structure-preserving directions that retain informative geometry. A Moore-Penrose pseudo-inverse formulation keeps the method well-posed under rank deficiency without data-specific hyperparameter tuning.

- **Extensive evaluation across regimes and domains**: We benchmark DVM-AD against 28 competitive baselines on 47 ADBench tabular datasets and 10 additional natural language processing/computer vision (NLP/CV) embedding datasets, showing strong and consistent performance across diverse anomaly types and real-world contexts. We further compare against 9 recent tabular AD methods, where DVM-AD remains the top-ranked detector.

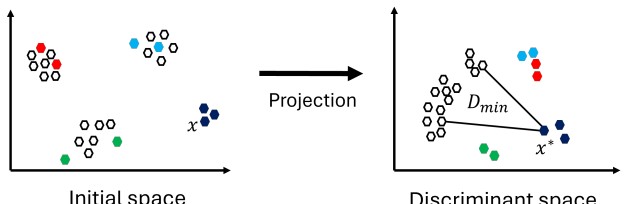

*Figure 1.* An example illustrates the concept of anomaly detection by projecting data into a discriminant space. In the initial space (left), four types of anomaly and normal points are shown: local (red), global (green), cluster (black), dependency (blue), and normal points (white). After projection (right), the separation between the normal data and anomaly becomes more apparent. The black point $x^*$ is marked as an anomaly based on its minimum distance $D_{min}$ to other normal points within the discriminant space.

**Code availability.** Our implementation is publicly available at https://github.com/troisang1/dvmad.

**Conflict of Interest Disclosure.** The authors declare no competing interests.

## 2. Related Work

Anomaly detection is a critical research area with numerous established algorithms. Previous studies (Zhao et al., 2019; Han et al., 2022; Jiang et al., 2023) have facilitated the evaluation of methods, broadly categorized into shallow machine learning and deep learning. This section outlines their key approaches, strengths, and limitations.

### 2.1. Shallow Machine Learning

Shallow machine-learning models are widely adopted for their simplicity, high interpretability, and lower training requirements compared to deep learning, making them suitable for resource-constrained devices (Mageed et al., 2024). Some key approaches are listed below (Zhao et al., 2019):

**Proximity-based methods** identify anomalies by analyzing the spatial relationships among data points. Methods like k-Nearest Neighbors (KNN) (Peterson, 2009) identify anomalies by computing distance to nearest neighbors, while Local Outlier Factor (LOF) (Breunig et al., 2000) compares local data density to neighbors. These excel at local anomaly detection but often require fine-tuning or struggle with density distributions.

**Probabilistic baselines** include ECOD (Li et al., 2022) and COPOD (Li et al., 2020). ECOD aggregates feature-wise empirical cumulative distributions, yielding an efficient detector with no tuned hyperparameters, but it mainly leverages marginal statistics and may underperform when anomalies are expressed through cross-feature dependencies. COPOD partially addresses this by using copula-based modeling to capture dependence structure.

**Linear/discriminant learning methods** provide simple, well-studied baselines. One-Class SVM (OCSVM) (Schölkopf et al., 2001) learns a boundary around normal data (often with kernels for nonlinear structure), while PCA-based detectors (Shyu et al., 2003) score anomalies via reconstruction error in a low-dimensional subspace. A related line is discriminant analysis: one-class kernel Fisher discriminants link one-class classification to Gaussian modeling in an induced feature space (Roth, 2004) and kernel null-space projections map known classes to representative points for distance-based novelty scoring (Bodesheim et al., 2013). Notably, these discriminant methods emphasize only one side of the compactness-structure trade-off.

**Ensemble-based methods** aggregate predictions from multiple base anomaly detectors. Isolation Forest (IForest) (Liu et al., 2008) is a notable example, isolating anomalies through recursive random partitions. This approach is computationally efficient and scalable, but its effectiveness may diminish in datasets with significant overlap between normal and anomalous distributions.

### 2.2. Deep Learning

Deep learning models aim to learn representations that capture complex patterns for anomaly detection (Mageed et al., 2024). A common family is **reconstruction-based methods**, where autoencoders (Bank et al., 2023) and variational autoencoders (Kingma et al., 2013) learn to reconstruct normal data, and anomalies are flagged by high reconstruction error or low likelihood proxies. In tabular settings, their detection performance often depends on architectural, regularization, and optimization details.

Another major family is **deep one-class/boundary learning**, which explicitly encourages normal data to concentrate in representation space. Examples include DeepSVDD (Ruff et al., 2018) and hybrid objectives such as AE1SVM (Nguyen & Vien, 2018), which combine representation learning with one-class boundary criteria. Related approaches introduce implicit negative pressure via adversarial or perturbation-based schemes, for instance DROCC (Goyal et al., 2020), to improve robustness to unseen anomalies.

A third family emphasizes **contrastive/self-supervised and structure-aware deep AD**. Transformation-based objectives and tabular deep detectors such as NeuTraLAD (Qiu et al., 2021) use learned transformations or normalized latent distributions to expose deviations, while graph or neighborhood-based designs such as LUNAR (Goodge et al., 2022) integrate local outlier detection with learned graph representations. These methods can be effective when the local structure is informative, but they introduce additional modeling and training choices.

A complementary line of work develops **generative anomaly detectors** that model the normal data distribution and score deviations from it. Early image-oriented detectors such as GANomaly (Akcay et al., 2018) train GANs on normal data and score by reconstruction or latent residuals, while recent tabular-specific generative detectors include MCM (Yin et al., 2024), diffusion-time estimators like DTE (Livernoche et al., 2024), and continuous-time variants such as TCCM (Li et al., 2025), which target intrinsic feature-correlation structure rather than point compactness.

Despite these advances, comprehensive tabular benchmarks find that deep AD methods often, if not always, fail to surpass simpler classical detectors on tabular AD (Han et al., 2022; Bouman et al., 2024), and their relative advantage varies markedly across datasets and anomaly types (Han et al., 2022; Jiang et al., 2023). Moreover, deep pipelines typically involve many architectural and training hyperparameters that can be highly sensitive (Ding et al., 2022; Zhao & Akoglu, 2024), making dataset-specific tuning difficult to avoid in one-class deployment, where validation labels are limited or unavailable.

**Positioning against recent strong tabular detectors**. Two

of the most competitive recent tabular AD methods are MCM (Yin et al., 2024) and DRL-AD (Ye et al., 2025). Both improve over earlier deep baselines, but differ from DVM-AD along three axes (Appendix Table 4): *(i) anomaly-type coverage* - These baselines primarily target specific structure, whereas DVM-AD spans local, global, cluster, and dependency anomalies; *(ii) hyperparameter requirements* - both baselines expose multiple deep training hyperparameters, while DVM-AD has no dataset-specific tuned hyperparameters; *(iii) data modality* - both target numerical tabular features, while DVM-AD also extends to CV/NLP embeddings under the same recipe (Appendix D.6.2).

# 3. Problem Statement

We now define the one-class anomaly detection problem of interest in this paper. We consider $N_{train}$ normal training samples denoted as $\mathbf{x}_1^{(\text{train})}, \ldots, \mathbf{x}_{N_{train}}^{(\text{train})} \in \mathbb{R}^d$, and $N_{test}$ testing samples represented as $\mathbf{x}_1^{(\text{test})}, \ldots, \mathbf{x}_{N_{test}}^{(\text{test})} \in \mathbb{R}^d$. Each test sample belongs to either the normal or anomalous category. For each $\mathbf{x}_i^{(\text{test})}$, our proposed method should assign an $anomaly\_score$ to specify if it is normal (low score) or an outlier (high score). The problem statement is as follows:

***Problem statement**: How to detect different types of anomalies in diverse real-world contexts by assigning an anomaly score for $x_i^{test}$ based on its representation in the discriminant space.*

In the following section, we present the DVM-AD algorithm that aims to cover diverse real-world data scenarios.

# 4. Our Proposed Approach

## 4.1. Overview

Before diving into the derivation, we sketch the DVM-AD pipeline at a high level. Training proceeds in three steps.

- *Proxy construction:* given normal training data only, we synthesize a single deterministic proxy point to anchor a contrastive geometry that makes one-class discriminant learning well-defined (§4.3).

- *Two-tailed spectrum:* we solve a generalized eigenproblem and prove that the resulting inverse-scatter spectrum is bounded in $[0, 1]$ (Theorem 4.1); we then retain discriminant vectors from both ends of this spectrum (§4.3).

- *Nearest-neighbor scoring:* at inference, a test sample is scored by its nearest-neighbor distance to projected training points and normalized using the training-geometry scale, producing a score in $[0, 1]$ (§4.4).

The remainder of this section formalizes these steps: §4.2 reviews the background, §4.3 presents the derivation and bounded-spectrum analysis, and §4.4 describes the algorithm.

## 4.2. Background on Discriminant Analysis

Given a training set $\{x_i,\}_{i=1}^N \subset \mathbb{R}^d$, which is partitioned into $c$ disjoint classes, Linear Discriminant Analysis (LDA) (Fisher, 1936) aims to find an optimal projection direction $\theta \in \mathbb{R}^d$ that maximizes class separability. This separability is quantified by the Fisher criterion, which balances two competing scatter matrices. The within-class scatter matrix, $S_w = \sum_{j=1}^c \sum_{i=1}^{n_j} (x_j^i - \bar{x}_j)(x_j^i - \bar{x}_j)^\top$, measures the compactness of samples around their respective class means, $\bar{x}_j$. The between-class scatter matrix, $S_b = \sum_{j=1}^c n_j (\bar{x}_j - \bar{x})(\bar{x}_j - \bar{x})^\top$, measures the separation between these class means and the global mean, $\bar{x}$. The Fisher criterion is then defined as:

$$f(\theta) = \frac{\theta^\top S_b \theta}{\theta^\top S_w \theta}. \qquad (1)$$

Setting $\nabla f(\theta) = 0$ yields the generalized eigenproblem $S_b\theta = \mu S_w\theta$, so maximizing $f$ amounts to finding directions where between-class spread dominates within-class spread (Ghojogh et al., 2019). The solutions, known as discriminant vectors (DVs), are the eigenvectors corresponding to the largest eigenvalues. These vectors form the columns of the projection matrix $\Theta$. The resulting transformation $y^i = \Theta^\top x^i$, known as the Foley-Sammon Transform (FST) (Foley & Sammon, 1975), projects the data into the learned discriminant space.

In contrast to FST, the Null Foley-Sammon Transform (NFST) (Guo et al., 2006) identifies specific projection directions where the within-class scatter vanishes while the between-class scatter remains positive. Such a $\theta$ is termed a Null Projection Direction (NPD), collapsing instances of a given class to a single point.

## 4.3. DVM-AD: One-Class Discriminant Vector Machine

Building upon the aforementioned analysis, we propose DVM-AD, a robust one-class projection method specifically designed for anomaly detection. Our approach aims to learn a discriminant space in which normal data becomes compact, while simultaneously preserving the informative global structure of the training set. To achieve this, we analyze the extremal values of the inverse scatter ratio

$$\kappa(\theta) = \frac{\theta^\top S_w \theta}{\theta^\top S_s \theta}. \qquad (2)$$

where $S_s = \sum_{i=1}^n (x^i - \bar{x})(x^i - \bar{x})^\top$ is the total scatter matrix. Since $S_s = S_w + S_b$ under our augmented two-class construction, $\kappa(\theta) = 1/(1 + f(\theta))$: $\kappa$ is a bounded

reparametrization of the Fisher ratio, which is what gives us the $[0, 1]$ spectrum below. As we will demonstrate, the directions that minimize this ratio achieve compactness, while those that maximize it preserve the data's structure.

Since $S_s$ is symmetric, positive semi-definite (PSD), its eigenvectors form an orthonormal basis for $\mathbb{R}^d$, represented by the columns of an orthogonal matrix $Q = [v_1, \ldots, v_d]$ (Anton & Rorres, 2013). Let the corresponding eigenvalues be ordered as $\lambda_1 \geq \lambda_2 \geq \cdots \geq \lambda_k > 0$, where $k = \mathrm{rank}(S_s)$. This allows for the orthogonal diagonalization:

$$Q^\top S_s Q = \Sigma = \mathrm{diag}(\lambda_1, \cdots, \lambda_k, 0, \cdots, 0), \quad (3)$$

where $\mathrm{diag}(\lambda_1, \cdots, \lambda_k, 0, \cdots, 0)$ is the diagonal matrix constructed from $(\lambda_1, \cdots, \lambda_k, 0, \cdots, 0)$.

To find the directions $\theta$ that extremize the ratio $\kappa$, we solve the problem in this new coordinate system via the ansatz $\theta = Q\alpha$. This transforms the problem into finding the extremal values of the generalized Rayleigh quotient for the coefficient vector $\alpha$:

$$\kappa(\alpha) = \frac{\alpha^\top (Q^\top S_w Q)\alpha}{\alpha^\top \Sigma \alpha}. \quad (4)$$

Directions in $\mathrm{Null}(\Sigma)$ are those along which the total scatter vanishes; on these directions $\kappa$ is of the form $0/0$ and carries no discriminant information. Restricting to $\mathrm{Range}(\Sigma)$ therefore loses nothing and yields a well-posed problem under rank deficiency (proof in Appendix B).

**Theorem 4.1** (DVM Solution). *Let $\Sigma^\dagger$ denote the Moore-Penrose pseudo-inverse of $\Sigma$, where $\lambda_i^\dagger = 1/\lambda_i$ for $\lambda_i > 0$ and $0$ otherwise. Define $\mathbf{M} = \Sigma^\dagger (Q^\top S_w Q)$, then*

- *$\mathbf{M}$ possesses $d$ eigenvalues lying within $[0, 1]$.*

- *Let the eigenvalues of $\mathbf{M}$ be sorted as $\lambda_1' \leq \lambda_2' \leq \cdots \leq \lambda_d'$, with corresponding eigenvectors $\alpha_1, \ldots, \alpha_d$. Each eigenvalue can be found sequentially by solving a constrained minimization problem:*

$$\lambda_i' = \min_{\substack{\alpha \in \mathrm{range}(\Sigma), \alpha \neq 0 \\ \alpha^\top \Sigma \alpha_j = 0 \, for \, j=1,\ldots,i-1}} \kappa(\alpha).$$

*This shows that the eigenvectors $\{\alpha_i\}$ are orthogonal with respect to the inner product defined by $\Sigma$.*

*Proof.* See Appendix B for the complete proof. □

**Interpretation.** The bounded spectrum has a clean geometric reading: each $\lambda_i'$ is the fraction of total scatter along $\alpha_i$ attributable to within-class scatter, so $\lambda_i' \approx 0$ marks directions that compress the normal class while $\lambda_i' \approx 1$ marks directions where the proxy-anchored separation vanishes and within-class variance dominates. Selecting from both extremes therefore captures complementary behaviors:

- **Minimal Tail** ($\lambda \approx 0$): $\theta^\top S_w \theta \approx 0$ yields *compressive* directions, akin to NPDs, that enforce compactness of the normal class.

- **Maximal Tail** ($\lambda \approx 1$): $\theta^\top S_b \theta \approx 0$ places $\theta$ approximately in the subspace orthogonal to the proxy–mean direction, where the optimization is dominated by high-variance directions in the normal class (typically aligned with its principal components), thereby preserving cluster and dependency structure.

This combined representation is highly effective: it offers both high sensitivity to deviations (via the collapse) and a stable neighborhood geometry (via the preserved structure), improving robustness across diverse anomaly types.

### 4.4. DVM-AD as an Outlier Detector

This subsection describes the end-to-end training and inference procedure of DVM-AD in the one-class setting. Given normal training samples $X_{\mathrm{train}} = \{x_i^{(\mathrm{train})}\}_{i=1}^N \subset \mathbb{R}^d$, DVM-AD first constructs an auxiliary two-class dataset in order to compute scatter matrices, then learns a projection $\Theta \in \mathbb{R}^{d \times m}, (m \leq d)$ into a discriminant space, and finally assigns an anomaly score to each test sample via nearest-neighbour distance in that space. The Algorithms 1 and 2 present these steps.

**Proxy point.** Because only normal data are available during training, we synthesize a single proxy point by taking the feature-wise maximum over the input training data (which should only include normal data):

$$\tilde{x} = \big( \max_i x_{i,1}^{(\mathrm{train})}, \ \ldots, \ \max_i x_{i,d}^{(\mathrm{train})} \big) \in \mathbb{R}^d. \quad (5)$$

We then form an augmented two-class dataset $X_{\mathrm{train}}^\star = X_{\mathrm{train}} \cup \{\tilde{x}\}$ with labels $y^\star$, where $y^\star = 0$ for normal samples and $y^\star(\tilde{x}) = 1$. The proxy is a geometric anchor that makes the model well-defined and is excluded from test-time scoring. Proposition E.1 (Appendix E.2) theoretically shows $S_w$ is independent of $\tilde{x}$. Consistent with this, three deterministic choices (feature-wise max, min, and farthest-from-mean) yield AUROC with no strategy dominating (Table 11); we adopt the max construction as a stable default.

**Base points in the discriminant space.** We project only the *normal* training samples to obtain base points

$$z_i = \Theta^\top x_i^{(\mathrm{train})}, \quad i = 1, \ldots, N, \quad (6)$$

and define the base-point set $B = \{z_i\}_{i=1}^N$. (We emphasize that the proxy point $\tilde{x}$ is used to compute $\Theta$ but is not included in $B$.)

**Two-tailed selection.** Empirically, the two-tailed selection drives the dominant performance gain and makes DVM-AD

naturally compatible with multimodal normal data: relative to a max-tail-only variant, it improves average AUROC by +4.81 on the 47 ADBench datasets, with the largest gains on local and dependency anomalies (Table 10).

**Anomaly score.** For a test sample $x^{(\text{test})} \in \mathbb{R}^d$, its projected representation is $x^{(\text{test})\star} = \Theta^\top x^{(\text{test})}$. We define the anomaly score as the minimum Euclidean distance to the normal base points:

$$\text{Score}(x^{(\text{test})}) = \min_{z \in B} \|x^{(\text{test})\star} - z\|_2. \qquad (7)$$

**Normalization and prediction.** The raw score in Eq. (7) lives on the natural Euclidean scale of the discriminant space and depends on $d$, $m$, and the data range. We normalize it by $dis_{\max}$, the largest leave-one-out nearest-neighbor gap within the projected training set: a test sample whose distance exceeds this internal gap lies further from the normal class than any training point.

$$dis_i^{(\text{train})} = \min_{j \neq i} \|z_i - z_j\|_2, \qquad dis_{\max} = \max_i dis_i^{(\text{train})}. \qquad (8)$$

The normalized anomaly score is then

$$p(x) = \min\left(1, \frac{\text{Score}(x)}{dis_{\max} + \varepsilon_{\text{tol}}}\right), \qquad (9)$$

where $\varepsilon_{\text{tol}} > 0$ is a small fixed constant used only for numerical stability (to avoid division by zero when $dis_{\max}$ is extremely small), and $p(x) \in [0, 1]$. To reduce the computational complexity, we use FAISS algorithm[1] to compute $dis_{\max}$ once.

---

**Algorithm 1** DVM-AD: Finding DVs

---

**Require:** Data matrix $X$, labels $y$, selection tolerance $\varepsilon_{\text{sel}}$
**Ensure:** Discriminant vector matrix $\Theta$
 1: Compute $S_w$ and $S_s$ from $X$ and $y$
 2: Eigendecompose $S_s = Q\Sigma Q^\top$ (Eq. (3))
 3: Compute $\Sigma^\dagger$ and set $M \leftarrow \Sigma^\dagger(Q^\top S_w Q)$
 4: Solve eigenproblem $MA = A\Lambda'$ where $\Lambda' = \text{diag}(\lambda_1', \ldots, \lambda_d')$ {Solving $M$ is equivalently solve symmetric PSD **S** (App. B).}
 5: Select indices $\mathcal{I} = \{i : \lambda_i' < \varepsilon_{\text{sel}} \text{ or } \lambda_i' > 1 - \varepsilon_{\text{sel}}\}$
 6: **if** $\mathcal{I} = \emptyset$ **then**
 7:    Set $\mathcal{I} = \{1, \ldots, d\}$
 8: **end if**
 9: Set $\Theta \leftarrow QA_{[:,\mathcal{I}]}$
10: **return** $\Theta$

---

**Implementation constants.** By tuning-free, we mean no dataset-specific hyperparameter selection. The only tolerances in Algorithms 1 and 2 are fixed across all datasets.

---

[1] https://faiss.ai/

---

**Algorithm 2** DVM-AD outlier detector

---

**Require:** Normal training set $X_{\text{train}}$, test set $X_{\text{test}}$, selection tolerance $\varepsilon_{\text{sel}}$, stability constant $\varepsilon_{\text{tol}}$
**Ensure:** Scores $\{p(x)\}_{x \in X_{\text{test}}}$ {(Optional) labels require a deployment threshold $t$}
 1: Construct proxy point $\tilde{x}$ and augmented set $(X_{\text{train}}^\star, y^\star)$
 2: Train $\Theta \leftarrow$ Algorithm 1 on $(X_{\text{train}}^\star, y^\star)$ using $\varepsilon_{\text{sel}}$
 3: Compute base points $B = \{\Theta^\top x : x \in X_{\text{train}}\}$
 4: Compute $dis_{\max} = \max_i \min_{j \neq i} \|z_i - z_j\|_2$ on $B$ {one-time normalisation}
 5: **for** each $x \in X_{\text{test}}$ **do**
 6:    $x^\star \leftarrow \Theta^\top x$
 7:    $D_{\min}(x) \leftarrow \min_{z \in B} \|x^\star - z\|_2$
 8:    $p(x) \leftarrow \min(1, D_{\min}(x)/(dis_{\max} + \varepsilon_{\text{tol}}))$
 9:    {Optional: if threshold $t$ is given, predict $\hat{y}(x) \leftarrow \mathbb{I}[p(x) > t]$}
10: **end for**
11: **return** $\{p(x)\}_{x \in X_{\text{test}}}$

---

We observe limited sensitivity to the fixed DV-selection tolerance $\epsilon_{\text{sel}}$; see Appendix E.3 for a sensitivity study. Additionally, the metrics we report (AUROC/AUPRC) are threshold-free and thus do not require selecting any decision threshold for our study. A deployment threshold $t$ is needed only at application time, and it is user- and use-case–dependent.

### 4.5. Computational Complexity Analysis

Let $N$ be the number of normal training samples, $d$ the input feature dimension, and $m$ the number of selected discriminant vectors (DVs) returned by Algorithm 1. Table 1 summarizes the time and memory complexity of DVM-AD for projection learning and scoring.

*Table 1.* Computational complexity of DVM-AD. Here, $N$ is the number of normal training samples, $d$ is the feature dimension, and $m$ is the number of selected DVs.

| Steps | Computation | Memory |
|---|---|---|
| **Projection learning** | | |
| Compute $S_w, S_s$ | $O(Nd^2)$ | $O(d^2)$ |
| $V, \Sigma \leftarrow \text{eig}(S_s)$ (or SVD) | $O(d^3)$ | $O(d^2)$ |
| Compute $\Sigma^\dagger$ (diagonal) | $O(d)$ | $O(d)$ |
| Form $M$ and solve $MA = A\Lambda'$ | $O(d^3)$ | $O(d^2)$ |
| Select DVs and form $\Theta \in \mathbb{R}^{d \times m}$ | $O(dm)$ | $O(dm)$ |
| Project training set to obtain $B = X\Theta$ | $O(Ndm)$ | $O(Nm)$ |
| ***Overall (projection learning)*** | $O(Nd^2 + d^3 + Ndm)$ | $O(d^2 + Nm)$ |
| **Scoring (FAISS ANN)** | | |
| Build FAISS index on $B$ | $\mathcal{T}_{\text{build}}(N, m)$ | $O(Nm) + \mathcal{M}_{\text{index}}$ |
| Compute $dis_{\max}$ (2-NN queries on $B$) | $N\,\mathcal{T}_{\text{qry}}(N, m)$ | $O(Nm) + \mathcal{M}_{\text{index}}$ |
| ***One-time scoring prep*** | $\mathcal{T}_{\text{build}} + N\,\mathcal{T}_{\text{qry}}$ | $O(Nm) + \mathcal{M}_{\text{index}}$ |
| ***Inference per test point*** (project + ANN query) | $O(dm) + \mathcal{T}_{\text{qry}}(N, m)$ | $O(Nm) + \mathcal{M}_{\text{index}}$ |

As shown in Table 1, projection learning is closed-form. Computing the scatter matrices costs $O(Nd^2)$, which is linear in $N$ for fixed $d$, and the subsequent spectral steps depend mainly on $d$ through an $O(d^3)$ eigen-decomposition.

This still remains practical for the moderate-dimensional regime typical of tabular AD ($d < 200$ for most ADBench datasets; under one second training for $d \leq 10^3$, Table 2); PCA pre-reduction is available at extreme $d$ (Appendix F.4). After obtaining $m$ DVs, projecting the normal training set yields the base set $B$ in $O(Ndm)$ time and $O(Nm)$ memory. For scoring, we built $B = \{z_i\}_{i=1}^N \subset \mathbb{R}^m$ after training and estimate $dis_{\max}$ by querying 2-NN for each $z_i$ (excluding itself) once using FAISS algorithm, which is equivalent to the exact method. Then, taking the maximum distance, and clipping the normalized score in (9) to $[0, 1]$. The inference per test point takes only $O(dm) + \mathcal{T}_{\text{qry}}(N, m)$.

# 5. Experiments

To provide a thorough and reliable assessment of DVM-AD's performance, we designed a comprehensive experimental study including diverse datasets, anomaly types, and a wide array of baselines. This section outlines our experiments that address the following key research questions:

- **RQ1 (Detection performance in real-world tabular data)**: How does DVM-AD compare with leading anomaly detection algorithms on tabular data, and to what extent does its performance generalize to the NLP and CV domains?

- **RQ2 (Detection performance on different anomaly types)**: How does the relative performance of DVM-AD vary across different anomaly types (local, global, cluster, and dependency)?

- **RQ3 (Runtime Efficiency and Scalability)**: How does DVM-AD perform when applied to large-scale datasets in terms of runtime and scalability?

## 5.1. Datasets

To address **RQ1**, we use original datasets provided by the **ADBench** (Han et al., 2022), a comprehensive framework designed for assessing anomaly detectors. Specifically, the benchmark comprises 47 tabular datasets and 10 datasets derived from NLP/CV data, presenting various real-world scenarios. For the 10 NLP/CV embedding data sources, we follow the ADBench protocol to construct multiple one-class tasks: each semantic class is treated as normal in turn, and the remaining classes are downsampled to form anomalies (5% contamination), resulting in 74 class-novelty tasks for text and image data. For reporting, we compute metrics per task and present macro-averages grouped by the 10 embedding data sources for readability.

To answer (**RQ2**), we generate four types of controlled synthetic datasets from the original ADBench datasets, based on the definitions, source code, and guidelines provided in

(Han et al., 2022). Detailed descriptions of this process are provided in Appendix D.2 and D.3.

To answer (**RQ3**), we employ randomly generated datasets with sample sizes ranging from $10^3$ to $10^6$ and feature dimensions ranging from $10^1$ to $10^4$. We measure the runtime over 10 times and report the average runtime.

## 5.2. Baseline Algorithms

We leverage the PyOD library (Zhao et al., 2019), a widely adopted benchmarking toolkit for anomaly detection, to implement our baseline algorithms. To ensure a thorough comparison, we select 28 high-detection-performance algorithms based on a comprehensive anomaly detection benchmark study (Han et al., 2022). The detailed baseline algorithms selection are presented in Appendix D.4.

## 5.3. Evaluation Metrics

In our study, we used the Area Under the Receiver Operating Characteristic Curve (AUROC) and the Area Under the Precision-Recall Curve (AUPRC) to assess detection performance. In addition to these metrics, we also record the training and testing speed to assess computational efficiency.

## 5.4. Experiment Setup

**General Settings**: Datasets are split into 7:3 train/test subsets with a fixed seed, via stratified sampling to preserve the anomaly ratio, following ADBench (Han et al., 2022) and recent tabular AD benchmarks (Durani et al., 2025). As per our problem definition, training subsets contain only normal samples. The additional CV and NLP datasets are used for evaluating in real-world contexts but are excluded from synthetic anomaly generation, as their high-dimensional nature poses challenges for generating synthetic anomalies. We follow the data preprocessing approach outlined in ADBench. We also benchmark DVM-AD under the 50/50 normal-only split used by recent works (Livernoche et al., 2024; Yin et al., 2024; Durani et al., 2025); see Appendix F.2.

**Hyperparameter Settings**: All baseline algorithms are configured with their default settings as provided in the PyOD library. DVM-AD has no tuned hyperparameters aside from fixed numerical tolerances.

## 5.5. Experimental Results

To compare algorithms comprehensively, we rank methods within each dataset by AUROC and AUPRC (full AUPRC results are provided in Appendix D.6.3). We then summarize performance using a critical difference diagram (CDD) and boxplots: the CDD reports each method's average rank (and average score), while the boxplots show the score distributions. In the boxplots, shallow ML methods are blue,

deep learning methods are green, and ours is pink.

### 5.5.1. DETECTION PERFORMANCE IN REAL-WORLD TABULAR DATA (RQ1)

The evaluation results summarized in Figures 2, and 3 illustrate the average ranks and value distributions of AUROC across 47 real-world tabular datasets.

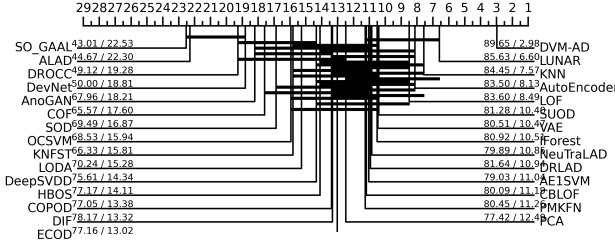

*Figure 2.* The average AUROC value and average rank of algorithms within 47 real-world tabular datasets. The higher average AUROC value is better while lower average rank is better.

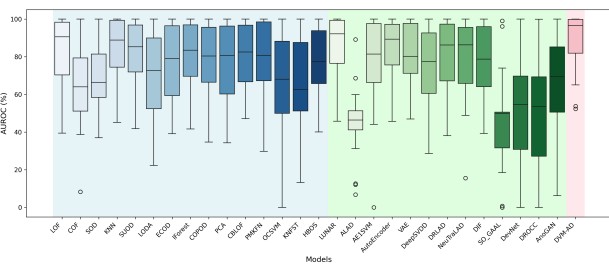

*Figure 3.* The distribution of AUROC value across 47 real-world datasets.

Figures 2 and 3 show that DVM-AD achieves the best AUROC across the 47 real-world datasets, with the highest average AUROC (89.65%) and the lowest average rank (2.98). The boxplots further indicate strong stability, with a tight interquartile range compared to the wider variability of other baselines. DVM-AD also outperforms discriminant-analysis baselines (KNFST, PMKFN, and PCA), highlighting it as a promising method for anomaly detection.

Additionally, we extend our experiment on 10 embedded NLP/CV embedding data sources (Appendix D.2) to evaluate DVM-AD detection performance in a cross-domain context. The detailed results in Appendix D.6.2 demonstrate that DVM-AD consistently outperforms baselines across other critical domains by achieving the best average rank of 1.60 and an average AUROC of 72.68%.

**Robustness to tuning and random splits:** We run an oracle best-of-grid study for KNN/IForest/LUNAR and then evaluate 5 random seeds (see Appendix D.7). Even under this optimistic baseline-tuning advantage, DVM-AD remains best mean AUROC (89.01%), outperform-

ing tuned baselines. Friedman rejects equal performance ($\chi^2 = 53.29, p = 1.59 \times 10^{-11}$), and Holm corrected one-sided Wilcoxon tests show DVM-AD significantly outperforms each baseline.

**Robustness to contamination and noise:** Appendices E.4 and E.5 compare DVM-AD against three strong baselines (KNN, IForest, LUNAR) under training-time annotation contamination and feature-wise Gaussian noise. DVM-AD maintains the highest AUROC at every level, leading the best baseline (LUNAR) by 3.42–4.02% on noise and 2.69-4.02% on contamination.

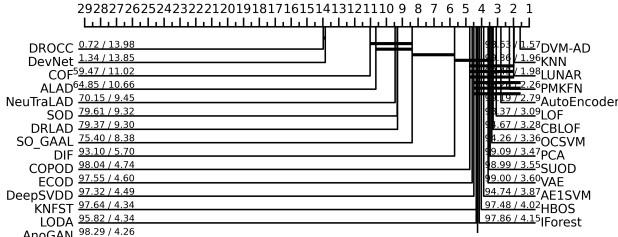

*(a)* Cluster anomaly.

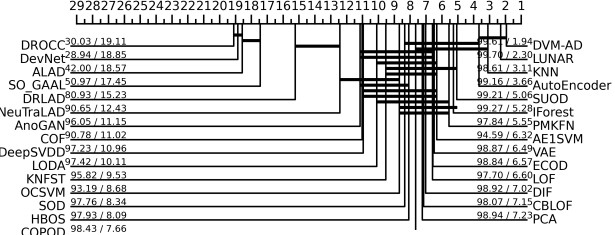

*(b)* Global anomaly.

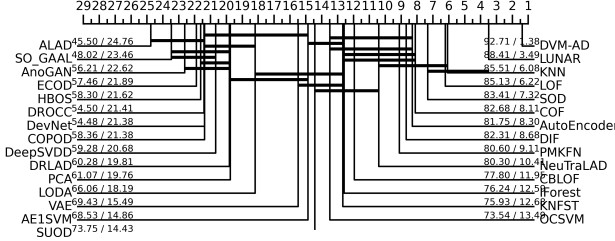

*(c)* Dependency anomaly.

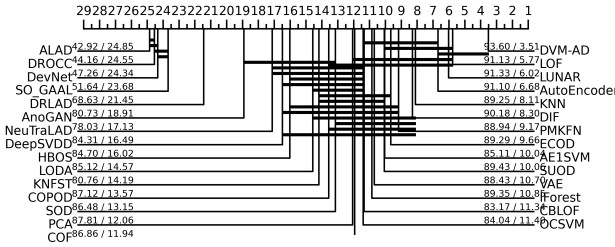

*(d)* Local anomaly.

*Figure 4.* The average AUROC value and average rank of algorithms on detecting different anomaly types.

### 5.5.2. DETECTION PERFORMANCE ON DIFFERENT ANOMALY TYPES (RQ2)

Figure 4 and Figure 5 show the average AUROC ranks across four anomaly types, and the corresponding performance distributions. DVM-AD achieves the best average rank for all anomaly types and the highest mean AUROC in three of them, remaining within 0.09% of LUNAR on global anomalies. The boxplots indicate strong stability, with the highest median AUROC and a compact interquartile range.

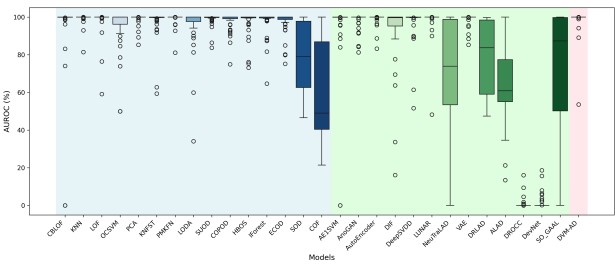

*(a)* Cluster anomaly.

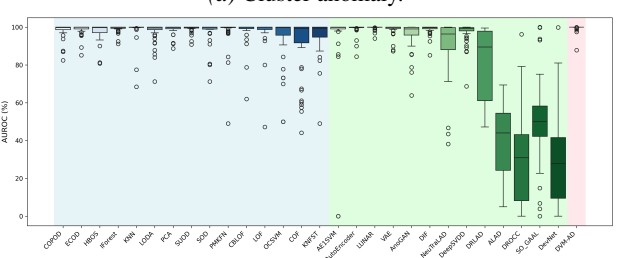

*(b)* Global anomaly.

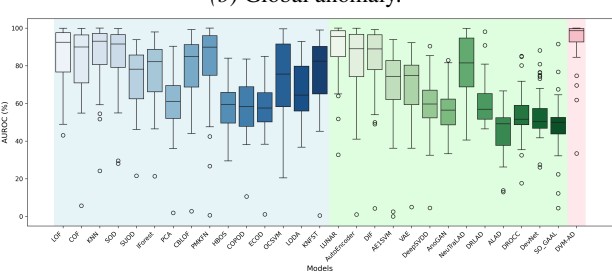

*(c)* Dependency anomaly.

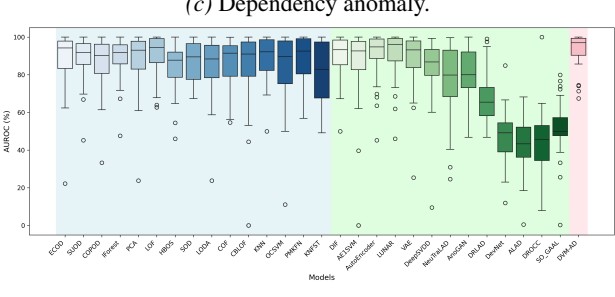

*(d)* Local anomaly.

*Figure 5.* The distribution of AUROC value on detecting four types of anomaly.

### 5.5.3. RUNTIME EFFICIENCY AND SCALABILITY (RQ3)

Within the moderate-dimensional regime typical of tabular AD (Section 4.5), the experimental results highlight DVM-AD's computational efficiency, with competitive training and testing time compared to the baselines. On the benchmark datasets, DVM-AD attains the lowest median training time among the 28 baselines; detailed results are reported in Appendix D.6.4. Table 2 further illustrates the scalability of DVM-AD with respect to the number of samples $n$ and the dimensionality $d$ in terms of training time, showing that the method remains computationally efficient as both $n$ and $d$ increase. We discard the extreme case ($n = 10^6, d = 10^4$) since the data matrix exceeds our 64 GB RAM hardware budget (see Appendix D.1). Beyond the in-paper scalability table, we additionally evaluate DVM-AD against 13 baselines at two extreme-scale regimes (Appendix D.6.4): at $N=10^6, d=10^3$ DVM-AD is the fastest method overall (32.5s), and at $N=10^5, d=10^4$ it remains operable (496s) where 7/13 baselines run out of memory.

*Table 2.* Training time (seconds) on synthetic data for varying $N$ and $d$. Note $d = 10^4$ larger than the maximum feature dimension in our 57-source benchmark.

| $N \backslash d$ | $10$ | $10^2$ | $10^3$ | $10^4$ |
|---|---|---|---|---|
| $10^3$ | $4.00\text{e-}4 \pm 9.80\text{e-}5$ | $9.27\text{e-}3 \pm 1.50\text{e-}2$ | $3.69\text{e-}1 \pm 1.40\text{e-}1$ | $4.07\text{e}2 \pm 4.70\text{e}0$ |
| $10^4$ | $1.44\text{e-}3 \pm 1.40\text{e-}4$ | $1.17\text{e-}2 \pm 6.70\text{e-}4$ | $7.78\text{e-}1 \pm 8.90\text{e-}1$ | $4.20\text{e}2 \pm 2.90\text{e}0$ |
| $10^5$ | $1.34\text{e-}2 \pm 4.90\text{e-}4$ | $2.93\text{e-}1 \pm 2.30\text{e-}1$ | $4.10\text{e}0 \pm 1.70\text{e}0$ | $4.96\text{e}2 \pm 1.90\text{e}0$ |
| $10^6$ | $1.46\text{e-}1 \pm 1.30\text{e-}3$ | $1.83\text{e}0 \pm 4.50\text{e-}2$ | $3.25\text{e}1 \pm 3.10\text{e}0$ | N/A |

## 6. Conclusion

This paper introduced DVM-AD, a closed-form one-class discriminant learning approach for tabular AD that avoids dataset-specific hyperparameter tuning. DVM-AD uses a deterministic proxy point to define one-class discriminant geometry, selects discriminant vectors from both tails of a bounded inverse-scatter spectrum, and uses a pseudo-inverse formulation to stay well-posed in rank-deficient regimes. Across 57 data sources, DVM-AD outperforms competitive baselines and remains robust across global, local, cluster, and dependency anomalies. Limitations and broader impacts are further discussed in Appendix H.

## Impact Statement

This work advances tabular anomaly detection by proposing a tuning-free method. Potential benefits include improved detection of rare events in domains such as fraud monitoring, reliability, and security. Potential risks include harms from false positives (unnecessary interventions) or uneven error rates across subpopulations in high-stakes deployments. We recommend deploying this method with human oversight, careful threshold calibration for the application context, and monitoring of error rates (including subgroup analysis when appropriate and legally/ethically permissible).

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

# A. Notation

Table 3 summarizes the main mathematical symbols used throughout the paper, including the data and indexing conventions, scatter matrices, projection/eigendecomposition quantities, and the scoring/normalization terms used by DVM-AD.

*Table 3.* Summary of mathematical notation used in this paper.

| Symbol | Meaning |
|---|---|
| *Data, sets, and indexing* | |
| $d$ | Feature dimension; samples satisfy $\mathbf{x} \in \mathbb{R}^d$. |
| $N$ or $N_{train}$ | Number of normal training samples. |
| $N_{test}$ | Number of test samples. |
| $i, j$ | Indices over samples (and occasionally features, where clear from context). |
| $c$ | Number of classes in the LDA preliminaries. |
| $n_j$ | Number of samples in class $j$ (LDA preliminaries). |
| $\mathbf{x}_i^{(\text{train})}$ | The $i$-th normal training sample. |
| $\mathbf{x}_i^{(\text{test})}$ | The $i$-th test sample (may be normal or anomalous). |
| $X_{\text{train}}$ | Normal training set, e.g., $X_{\text{train}} = \{\mathbf{x}_i^{(\text{train})}\}_{i=1}^N$. |
| $X_{\text{test}}$ | Test set, e.g., $X_{\text{test}} = \{\mathbf{x}_i^{(\text{test})}\}_{i=1}^M$. |
| $X$ | Data matrix input to Algorithm 1 for computing scatter matrices. |
| $y$ | Class labels corresponding to $X$ (LDA preliminaries / generic supervised notation). |
| $y^\star$ | Labels for the augmented two-class set $X_{\text{train}}^\star$ (normal = 0, proxy = 1). |
| $\bar{\mathbf{x}}$ | Mean of the (normal) training set (and/or global mean in LDA preliminaries, as context requires). |
| $\bar{\mathbf{x}}_j$ | Mean of class $j$ (LDA preliminaries). |
| $\mathbf{y}_i$ | Projected sample in classical discriminant analysis, e.g., $\mathbf{y}_i = \Theta^\top \mathbf{x}_i$ (FST notation). |
| *Scatter matrices and discriminant quantities* | |
| $S_w$ | Within-class scatter matrix. |
| $S_b$ | Between-class scatter matrix. |
| $S_s$ | Total scatter matrix (in DVM-AD, also expressed as $S_s = S_w + S_b$ under the augmented two-class view). |
| $\theta$ | A projection direction / discriminant vector in $\mathbb{R}^d$. |
| $\Theta$ | Projection matrix whose columns are selected discriminant vectors (DVs). |
| $m$ | Number of selected DVs (columns of $\Theta$), typically $m \ll d$. |
| $f(\theta)$ | Fisher criterion in LDA preliminaries, e.g., $f(\theta) = \frac{\theta^\top S_b \theta}{\theta^\top S_w \theta}$. |
| $\mu$ | Generalized eigenvalue in $S_b \theta = \mu S_w \theta$ (LDA preliminaries). |
| $\kappa(\theta)$ | Inverse scatter ratio used by DVM-AD, $\kappa(\theta) = \frac{\theta^\top S_w \theta}{\theta^\top S_s \theta}$. |
| *Spectral decomposition and pseudo-inverse formulation* | |
| $Q$ | Orthogonal eigenvector matrix of $S_s$ (columns are eigenvectors). |
| $\mathbf{v}_i$ | $i$-th eigenvector of $S_s$ (column of $Q$). |
| $\lambda_i$ | Eigenvalues of $S_s$ (typically ordered with the nonzero ones first). |
| $k$ | Rank of $S_s$ (number of nonzero eigenvalues). |
| $\Sigma$ | Diagonal eigenvalue matrix in $Q^\top S_s Q = \Sigma = \text{diag}(\lambda_1, \ldots, \lambda_k, 0, \ldots, 0)$. |
| $\alpha$ | Coefficient vector in eigenbasis via $\theta = Q\alpha$. |
| $\lambda_i^\dagger$ | Pseudo-inverse eigenvalue: $\lambda_i^\dagger = 1/\lambda_i$ if $\lambda_i > 0$, else 0. |
| $\Sigma^\dagger$ | Moore–Penrose pseudo-inverse of $\Sigma$ (invert nonzero eigenvalues, keep zeros). |
| $M$ | Matrix $M = \Sigma^\dagger (Q^\top S_w Q)$ used in Algorithm 1. |
| $A$ | Eigenvector matrix of $M$ (columns are eigenvectors). |
| $\Lambda'$ | Diagonal matrix of eigenvalues of $M$: $\Lambda' = \text{diag}(\lambda_1', \ldots, \lambda_d')$. |
| $\lambda_i'$ | Eigenvalues of $M$; the bounded inverse-scatter spectrum satisfies $\lambda_i' \in [0,1]$. |
| $I$ | Selected DV index set in Algorithm 1 (two-tailed rule based on $\lambda_i'$). |
| $\text{Range}(\cdot)$ | Column space operator (used in Appendix proofs). |
| $\text{Null}(\cdot)$ | Null space operator (used in Appendix proofs). |
| *Proxy point construction and augmented training* | |

Table 3 (continued)

| Symbol | Meaning |
|---|---|
| $\tilde{\mathbf{x}}$ | Proxy (synthetic) anomaly point; default is feature-wise maximum over normal training data. |
| $X_{\text{train}}^{\star}$ | Augmented set $X_{\text{train}}^{\star} = X_{\text{train}} \cup \{\tilde{\mathbf{x}}\}$. |
| $\tilde{x}_j^{\min}$ | $j$-th coordinate of the Min-corner proxy: $\tilde{x}_j^{\min} = \min_i x_{i,j}$. |
| $\tilde{\mathbf{x}}^{\min}$ | Min-corner proxy vector $(\tilde{x}_1^{\min}, \ldots, \tilde{x}_d^{\min})$. |
| $i^{\star}$ | Index of farthest sample from the mean: $i^{\star} = \arg\max_i \|\mathbf{x}_i - \bar{\mathbf{x}}\|_2$. |
| $\tilde{\mathbf{x}}^{\text{far}}$ | Farthest-from-mean proxy: $\tilde{\mathbf{x}}^{\text{far}} = \mathbf{x}_{i^{\star}}$. |
| $\boldsymbol{\mu}^{\star}$ | Global mean of augmented set, e.g., $\boldsymbol{\mu}^{\star} = \frac{n\bar{\mathbf{x}}+\tilde{\mathbf{x}}}{n+1}$. |
| $\boldsymbol{\delta}$ | Proxy displacement: $\boldsymbol{\delta} = \bar{\mathbf{x}} - \tilde{\mathbf{x}}$. |
| $\beta$ | Rank-one scaling factor in $S_b = \beta\,\boldsymbol{\delta}\boldsymbol{\delta}^{\top}$ (Appendix proxy analysis). |

*Projection, base points, and scoring*

| | |
|---|---|
| $z_i$ | Base point in discriminant space: $z_i = \Theta^{\top}\mathbf{x}_i^{(\text{train})}$. |
| $B$ | Base set of projected normal points: $B = \{z_i\}_{i=1}^N$. |
| $\mathbf{x}^{\star}$ | Projected representation of a sample $\mathbf{x}$: $\mathbf{x}^{\star} = \Theta^{\top}\mathbf{x}$. |
| $\text{Score}(\mathbf{x})$ | Raw anomaly score: $\text{Score}(\mathbf{x}) = \min_{z \in B} \|\mathbf{x}^{\star} - z\|_2$. |
| $D_{\min}(\mathbf{x})$ | Minimum distance to base set in discriminant space (used interchangeably with $\text{Score}(\mathbf{x})$ in the paper). |
| $dis_i^{(\text{train})}$ | Leave-one-out nearest-neighbour distance for base point $z_i$: $dis_i^{(\text{train})} = \min_{j \neq i} \|z_i - z_j\|_2$. |
| $dis_{\max}$ | Training-geometry scale: $dis_{\max} = \max_i dis_i^{(\text{train})}$. |
| $p(\mathbf{x})$ | Normalized anomaly score: $p(\mathbf{x}) = \min\left(1, \frac{\text{Score}(\mathbf{x})}{dis_{\max}+\epsilon_{\text{tol}}}\right) \in [0, 1]$. |
| $\epsilon_{\text{sel}}$ | Fixed DV-selection tolerance (two-tailed selection in Algorithm 1). |
| $\epsilon_{\text{tol}}$ | Small stability constant in normalization (avoid division by zero). |
| $t$ | Deployment threshold for converting scores to binary predictions (application-dependent). |
| $\hat{y}(\mathbf{x})$ | Optional predicted label, e.g., $\hat{y}(\mathbf{x}) = \mathbb{I}[p(\mathbf{x}) > t]$. |
| $\mathbb{I}[\cdot]$ | Indicator function (1 if condition holds, else 0). |

*Complexity notation (FAISS ANN)*

| | |
|---|---|
| $\mathcal{T}_{\text{build}}(N, m)$ | Time to build the ANN index on $B$. |
| $\mathcal{T}_{\text{qry}}(N, m)$ | Time per ANN query on the index. |
| $\mathcal{M}_{\text{index}}$ | Memory overhead of the ANN index (in addition to storing $B$). |

*Proof-only auxiliary variables (Appendix)*

| | |
|---|---|
| $\alpha_r, \alpha_0$ | Decomposition $\alpha = \alpha_r + \alpha_0$ with $\alpha_r \in \text{Range}(\Sigma)$ and $\alpha_0 \in \text{Null}(\Sigma)$. |
| $u, u_r, u_0$ | Decomposition $u = u_r + u_0$ used in Appendix proofs. |
| $S$ | Auxiliary symmetric matrix in Appendix A: $S = (\Sigma^{\dagger})^{1/2}(Q^{\top}S_wQ)(\Sigma^{\dagger})^{1/2}$. |
| $f_S(u)$ | Rayleigh quotient $f_S(u) = \frac{u^{\top}Su}{u^{\top}u}$. |

## B. The Proof of Theorem (4.1)

Recall

$$\kappa(\alpha) = \frac{\alpha^{\top}(Q^{\top}S_wQ)\alpha}{\alpha^T\Sigma\alpha}. \tag{10}$$

Analysis of the null space of $\Sigma$ (detailed in Appendix B) shows that the achievable values of $\kappa$ over its entire domain are identical to those achieved on $\text{Range}(\Sigma)$. Indeed, if $\alpha \in \text{Null}(\Sigma)$, then $0 = \alpha^{\top}\Sigma\alpha = \alpha^{\top}(Q^{\top}S_wQ)\alpha + \alpha^{\top}(Q^{\top}S_bQ)\alpha$. Both terms are nonnegative, so $\alpha^{\top}(Q^{\top}S_wQ)\alpha = 0$ and hence $(Q^{\top}S_wQ)\alpha = 0$ due to PSD of $(Q^{\top}S_wQ)$. Thus, on $\text{Null}(\Sigma)$, $\kappa(\alpha)$ is indeterminate (0/0). Now, for any $\alpha \in \mathbb{R}^d$, it decomposes $\alpha = \alpha_r + \alpha_0$ with $\alpha_r \in \text{Range}(\Sigma)$ and $\alpha_0 \in \text{Null}(\Sigma)$. It claims that $\alpha^{\top}(Q^{\top}S_wQ)\alpha = \alpha_r^{\top}(Q^{\top}S_wQ)\alpha_r$ and $\alpha^{\top}\Sigma\alpha = \alpha_r^{\top}\Sigma\alpha_r$, so $\kappa(\alpha) = \kappa(\alpha_r)$. This implies that the set of all achievable values of $\kappa(\alpha)$ over its entire domain is identical to the set of values achieved on $\text{Range}(\Sigma)$.

**Proposition B.1.** *Let* $\mathbf{S} = (\Sigma^{\dagger})^{1/2}(Q^{\top}S_wQ)(\Sigma^{\dagger})^{1/2}$. *Any eigenvector of* $\mathbf{S}$ *corresponding to a non-zero eigenvalue must lie within the range of* $\Sigma$.

*Proof.* We first note two key properties: (1) since $(\Sigma^{\dagger})^{1/2}$ is diagonal with non-zero entries only for non-zero eigenvalues of $\Sigma$, $\text{Range}((\Sigma^{\dagger})^{1/2}) = \text{Range}(\Sigma)$; (2) consequently, $\text{Range}(\mathbf{S}) \subseteq \text{Range}(\Sigma)$.

Let $(\lambda', u)$ be an eigenpair of $\mathbf{S}$ with $\lambda' \neq 0$, i.e., $\mathbf{S}u = \lambda' u$. Decompose $u$ as $u = u_r + u_0$, where $u_r \in \text{Range}(\Sigma)$ and $u_0 \in \text{Null}(\Sigma)$. Substituting this into the eigen-equation yields:

$$\mathbf{S}u = \lambda' u_r + \lambda' u_0$$

Since $\mathbf{S}u \in \text{Range}(\mathbf{S}) \subseteq \text{Range}(\Sigma)$ and $\lambda' u_r \in \text{Range}(\Sigma)$, the term $\lambda' u_0$ must also belong to $\text{Range}(\Sigma)$. However, as $u_0 \in \text{Null}(\Sigma)$ and $\lambda' \neq 0$, this is only possible if $u_0 = 0$. Thus, $u = u_r \in \text{Range}(\Sigma)$. $\qquad \square$

*The proof of Theorem* (4.1). The proof relies on establishing the equivalence between $\kappa(\alpha)$ and the standard Rayleigh quotient for the similar symmetric matrix $\mathbf{S} = (\Sigma^\dagger)^{1/2}(Q^\top S_w Q)(\Sigma^\dagger)^{1/2}$:

Step 1: Let $(\lambda', u)$ be an eigenpair of the symmetric matrix $\mathbf{S}$ i.e., $\mathbf{S}u = \lambda' u$. Pre-multiplying both sides by $(\Sigma^\dagger)^{1/2}$ yields $\left(\Sigma^\dagger(Q^\top S_w Q)\right)\left((\Sigma^\dagger)^{1/2}u\right) = \lambda' \left((\Sigma^\dagger)^{1/2}u\right)$ leading to $(\lambda', (\Sigma^\dagger)^{1/2}u)$ is a eigenpair of $\mathbf{M}$. Furthermore, since $\mathbf{S}$ is a real, symmetric, and positive semi-definite matrix, it possesses $d$ real, non-negative eigenvalues. As $\mathbf{M}$ shares the same eigenvalues as $\mathbf{S}$, $\mathbf{M}$ is also guaranteed to have $d$ real, non-negative eigenvalues.

Step 2: By the Courant-Fischer theorem, the eigenvalues of the symmetric matrix $\mathbf{S}$, sorted as $\lambda'_1 \leq \lambda'_2 \leq \cdots \leq \lambda'_d$, can be characterized through its Rayleigh quotient:

$$f_{\mathbf{S}}(u) = \frac{u^\top \mathbf{S} u}{u^\top u}.$$

- The smallest and largest eigenvalues are the extremal values of $f_{\mathbf{S}}(u)$ over the entire space:

$$\lambda'_1 = \min_{u \neq 0} f_{\mathbf{S}}(u) \quad \text{and} \quad \lambda'_d = \max_{u \neq 0} f_{\mathbf{S}}(u).$$

- The $i$-th smallest eigenvalue is obtained sequentially by minimizing the Rayleigh quotient over the subspace orthogonal to the first $i-1$ eigenvectors $(u_1, \ldots, u_{i-1})$:

$$\lambda'_i = \min_{u \neq 0,\, u \perp u_1, \ldots, u_{i-1}} f_{\mathbf{S}}(u).$$

Step 3: Any eigenvector $u$ of $\mathbf{S}$ corresponding to an eigenvalue $\lambda' > 0$ must lie within $\text{Range}(\Sigma)$ (see (B.1)). Hence, $\lambda'_i > 0$, can be characterized as

$$\lambda'_i = \min_{u \in \text{Range}(\Sigma), u \neq 0,\, u \perp u_1, \ldots, u_{i-1}} f_{\mathbf{S}}(u).$$

Step 4: We introduce the change of variables that for $u \in \text{Range}(\Sigma)$, we set $\alpha = (\Sigma^\dagger)^{1/2}u$. The inverse is $u = \Sigma^{1/2}\alpha$. Substituting $\alpha = (\Sigma^\dagger)^{1/2}u$ into $\kappa(\alpha)$ gives

$$\kappa(\alpha) = \frac{u^\top (\Sigma^\dagger)^{1/2}(Q^\top S_w Q)(\Sigma^\dagger)^{1/2}u}{u^\top (\Sigma^\dagger)^{1/2}\Sigma(\Sigma^\dagger)^{1/2}u} = \frac{u^\top \mathbf{S} u}{u^\top u} = f_{\mathbf{S}}(u).$$

Here we used the fact that as $u \in \text{Range}(\Sigma)$, by straightforward computing gives $u^\top (\Sigma^\dagger)^{1/2}\Sigma(\Sigma^\dagger)^{1/2}u = u^\top u$. Furthermore, the orthogonality condition $u_i^\top u_j = 0$ transforms into $(\Sigma^{1/2}\alpha_i)^\top(\Sigma^{1/2}\alpha_j) = \alpha_i^\top \Sigma \alpha_j = 0$. Then, $\lambda'_i$ is obtained by solving a constrained minimization problem:

$$\lambda'_i = \min_{\substack{\alpha \in \text{Range}(\Sigma), \alpha \neq 0 \\ \alpha^\top \Sigma \alpha_j = 0 \text{ for } j=1, \ldots, i-1}} \kappa(\alpha).$$

Step 5: Using the identity $\Sigma = Q^\top S_w Q + Q^\top S_b Q$, the quotient becomes:

$$\kappa(\alpha) = \frac{\alpha^\top Q^\top S_w Q \alpha}{\alpha^\top Q^\top S_w Q \alpha + \alpha^\top Q^\top S_b Q \alpha}$$

Since both $S_w$ and $S_b$ are positive semi-definite, the numerator is always less than or equal to the denominator, which implies $\lambda'_i \leq 1$. The lower bound is trivially $\lambda'_i \geq 0$. Thus, all eigenvalues lie in $[0, 1]$. $\qquad \square$

**Numerical note.** Although Algorithm 1 uses $M = \Sigma^\dagger(Q^\top S_w Q)$ (not necessarily symmetric), this section shows it is similar on $\text{Range}(\Sigma)$ to the symmetric PSD matrix $S = (\Sigma^\dagger)^{1/2}(Q^\top S_w Q)(\Sigma^\dagger)^{1/2}$, hence they share the same real nonnegative spectrum $\{\lambda_i'\}$. For numerical stability, one may solve $SU = U\Lambda'$ with a symmetric eigensolver and recover $A = (\Sigma^\dagger)^{1/2}U$, $\Theta = QA_{[:,I]}$.

**Implementation note.** In all experiments, we solve the eigenproblem $MA = A\Lambda'$ directly as in Algorithm 1 and we did not observe numerical instabilities (e.g., complex eigenvalues beyond machine precision). If a numerical package returns complex values due to round-off, one can safely take the real part when the imaginary component is negligible; alternatively, one can solve the symmetric eigenproblem $SU = U\Lambda'$ with a symmetric eigensolver and recover $A = PU$ (and thus $\Theta = QA_{:,I}$). Both routes can yield the same $\Lambda'$ and an equivalent selected subspace.

In practice, one may equivalently restrict computations to the $k = \text{rank}(S_s)$ nonzero-eigenvalue subspace of $S_s$ by using $Q_k \in \mathbb{R}^{d \times k}$ and $\Sigma_k = \text{diag}(\lambda_1, \ldots, \lambda_k)$. Then $M_k = \Sigma_k^{-1}(Q_k^\top S_w Q_k)$ and $S_k = \Sigma_k^{-1/2}(Q_k^\top S_w Q_k)\Sigma_k^{-1/2}$ are $k \times k$ matrices, and the resulting $\Theta$ is unchanged after lifting back to $\mathbb{R}^d$.

## C. Related Works With More Details

In this section, we conduct a survey and present Table 4, which supports the limitations discussed in Section 1. The table compares our proposed DVM-AD with various baseline models across four aspects: data assumptions, handling of different types of anomalies, the need for hyperparameter tuning, and model performance on real-world data.

The table shows that our model outperforms others in real-world data detection by handling the widest variety of data types, including tabular, computer vision, and natural language processing datasets. Besides, DVM-AD also analyzes a broader range of anomaly types (global, dependency, local and cluster), while baseline models typically excel with only one or two anomaly types. Overall, our proposed DVM-AD gets a better performance across the evaluated aspects.

*Table 4.* Comparative overview of various anomaly detection algorithms.

| Algorithm | Anomaly type | Types of real-world data | Main hyperparameters | Data Assumption |
|---|---|---|---|---|
| LOF | Local Anomaly | Tabular | No. neighbors Leaf size | Local density variation between normal and anomalous data points |
| PCA | Global Anomaly | Tabular | No. components | Anomalies deviate from the global variance structure of the data |
| SOD | Global Anomaly Local Anomaly | Tabular | Window size Learning rate | The data follows certain patterns or trends; can detect both local and global anomalies based on behaviour |
| AutoEncoder | Global Anomaly Local Anomaly | Tabular Computer vision | Latent space size Learning rate No. epochs | Reconstruction error can be used as an anomaly detection criterion |
| LUNAR | Local Anomaly | Tabular Computer vision | Learning rate No. layers Latent size No. neighbors | Anomalies manifest as deviations from learned patterns or clusters |
| DeepSVDD | Global Anomaly | Tabular | Learning rate No. epochs Regularization | The data follows some distribution and anomalies lie outside of it |
| DevNet | Local Anomaly | Tabular | No. epochs Learning rate Layers | Anomalies are deviations from a learned manifold |
| LODA | Local Anomaly | Tabular | No. bins Window size | The majority of data lies within a normal range, and deviations indicate anomalies |
| KNN | Local Anomaly | Tabular | No. neighbors Distance metric | Local density and distance properties for anomaly detection |

*Continued on next page*

Table 4 (continued)

| Algorithm | Anomaly type | Types of real-world data | Main hyperparameters | Data Assumption |
|---|---|---|---|---|
| IForest | Global Anomaly | Tabular | No. trees
Sample size | Anomalies are few and different from the majority of data |
| OCSVM | Global Anomaly | Tabular | Kernel type
C (regularization)
nu (outlier fraction) | A hyperplane can separate normal and anomalous points |
| HBOS | Global Anomaly | Tabular | Bin width
No. bins | Normal data follows a certain distribution (e.g., Gaussian, Poisson) |
| COF | Local Anomaly | Tabular | k (neighbors)
Distance metric | Data points that are close together should have similar densities |
| CBLOF | Cluster Anomaly | Tabular | Cluster size
Distance metric | Data clusters that may contain anomalies separate from normal clusters |
| COPOD | Global Anomaly | Tabular | None | Anomalies are significantly different from the overall distribution |
| ECOD | Global Anomaly | Tabular | None | Normal data follows a specific multivariate distribution |
| KNFST | Global Anomaly | Computer Vision | Kernel type
Kernel bandwidth | Collapses normal data in kernel null space; deviations indicate anomalies |
| PMKFN | Global Anomaly | Tabular | Number of kernels | Learns multi-kernel null-space projection for anomaly detection |
| AE1SVM | Global Anomaly
Local Anomaly | Tabular | Latent size
C (SVM parameter)
gamma | Reconstruction from autoencoders combined with SVM |
| ALAD | Local Anomaly | Tabular | Learning rate
No. epochs | Anomalies are deviations in the latent space learned by deep models |
| SUOD | Global Anomaly
Local Anomaly | Tabular | Base detector set
Number of estimators | Suitable for heterogeneous anomalies detectable via diverse ensemble models |
| DROCC | Local Anomaly | Tabular | Learning rate
Radius (r)
Epochs | Detects anomalies lying outside compact normal regions in latent space |
| NeuTraLAD | Local Anomaly | Tabular | Learning rate
Batch size
Epochs | Sensitive to deviations in normalized latent distributions |
| DIF | Global Anomaly | Tabular | Learning rate
Latent dimension | Focuses on instances far from denoised representations in latent space |
| AnoGAN | Local Anomaly | Computer vision | Learning rate
Epochs | Targets inputs that cannot be realistically reconstructed by the generator |
| SO_GAAL | Global Anomaly | Tabular | Learning rate
Epochs | Detects outliers that differ significantly from synthetic pseudo-anomalies |
| DRL-AD | Global Anomaly | Tabular | Learning rate
Epochs | Detects anomalies via latent decomposition reconstruction error |
| VAE | Global Anomaly | Tabular | Latent dimension
Learning rate
Epochs | Flags instances with high reconstruction error in latent space |
| MCM | Global Anomaly
Dependency Anomaly | Tabular | Number of masks
Mask diversity weight
Learning rate
Epochs | Self-supervised masked-cell reconstruction; captures feature correlations among normal data |

Table 4 (continued)

| Algorithm | Anomaly type | Types of real-world data | Main hyperparameters | Data Assumption |
|---|---|---|---|---|
| NPT-AD | Global Anomaly Dependency Anomaly | Tabular | Learning rate Hidden dimension Stacking depth Training steps | Combines feature–feature and sample–sample dependencies via a Non-Parametric Transformer reconstructing masked features |
| RAAD | Global Anomaly Dependency Anomaly | Tabular | Learning rate Number of retrieved neighbours Hidden dimension Epochs | Reconstruction-based detector augmented with retrieval over normal training samples to inject sample–sample dependencies |
| DTE | Global Anomaly Local Anomaly | Tabular | Learning rate Number of diffusion steps Hidden dimension Epochs | Estimates the diffusion-time distribution of an input; mode/mean of the estimated time is used as the anomaly score |
| PTAD | Global Anomaly Local Anomaly Cluster Anomaly Dependency Anomaly | Tabular | Learning rate Number of basis vectors Number of prototypes Epochs | Combines data-space and projection-space mask modeling with optimal-transport-based prototype learning over orthogonal basis vectors |
| GANomaly | Global Anomaly Local Anomaly | Computer Vision | Learning rate Latent dimension Adversarial loss weights Epochs | Encoder–decoder–encoder GAN trained on normal data; scores combine reconstruction and latent-residual terms |
| TCCM | Global Anomaly Dependency Anomaly | Tabular | Learning rate Hidden dimension Number of time samples Epochs | One-step flow-matching detector: learns a time-conditioned contraction velocity field; scores by deviation from expected contraction |
| ADERH | Global Anomaly Local Anomaly | Tabular | Number of hyperspheres Pair-sampling size | Ensemble of multi-scale hyperspheres on random point pairs; refined by Pitch and NDensity signals (isolation-based) |
| ICL | Global Anomaly Dependency Anomaly | Tabular | Learning rate Hidden dimension Number of ensemble heads Epochs | Internal contrastive learning between feature subsets; trained to maximise mutual information; contrastive loss serves as the anomaly score |
| Ours | Global Anomaly Cluster Anomaly Dependency Anomaly Local Anomaly | Tabular Computer vision NLP | None (no tuned hyperparameters; fixed numerical tolerances only) | None |

# D. Extended Experiments With More Details

### D.1. Software and Hardware Environment Configuration

All experiments were conducted using Python 3.10.13 and widely-used open-source libraries. The experiments were performed on a server running Ubuntu 22.04, equipped with a 13th Gen Intel Core i5-13400F processor and 64GB of RAM.

For anomaly detection, we utilized the PyOD library (Zhao et al., 2019; Han et al., 2022) (version 1.0.8), an open-source Python framework that provides a wide range of algorithms and comprehensive documentation. Additionally, handling data and evaluation metrics was handled using the Scikit-learn library version 1.3.2.

## D.2. Datasets

In this section, we provide additional details regarding the datasets discussed in Section 5.1. ADBench (Han et al., 2022) includes 47 core benchmark datasets for model evaluation, covering a variety of application areas such as healthcare, audio, and image processing. This diverse set of datasets enables a thorough assessment of various anomaly detection methods. Additionally, 10 more complex datasets with larger sample sizes from the computer vision (CV) and natural language processing (NLP) domains have been included, as shown in Table 5.

Following the approach in the ADBench paper (Han et al., 2022), the number of samples per dataset (including both tabular and NLP/CV datasets) is limited to 10,000 to manage computational costs. It's worth noting that this downsample is intended to ensure that some baseline models (such as KNFST) can run the experiments and only affect a few of the huge datasets. DVM-AD is not limited by the scale of samples (see 5.5.3). For NLP datasets, we use the ADBench-provided BERT (Devlin et al., 2019) pretrained on the BookCorpus embeddings. For CV datasets, we use the ADBench-provided ResNet18 (He et al., 2016) pretrained on ImageNet (Deng et al., 2009) and extract embeddings from the final average pooling layer.

Furthermore, since the 10 external datasets include multiple classes, We follow a standard one-vs-rest class-novelty protocol: one semantic class is treated as normal, and the remaining classes are downsampled to 5% of the total instances to form anomalies, yielding 74 class-novelty tasks in total. This setting is closer to novelty/OOD detection on fixed embeddings than to classical tabular AD; we include it as a stress test of DVM-AD's applicability beyond tabular features.

Additionally, we use the standard ADbench (Han et al., 2022) preprocessing and standardize only the training data to match their original distributions.

## D.3. Additional Demonstration of Synthetic Anomalies

To evaluate the performance of our approach across different anomaly types, we test DVM-AD on 47 synthetic benchmark datasets generated from the 47 tabular datasets used in the original ADBench paper (Han et al., 2022). This allows us to assess how well the proposed methods work with different anomaly types. Below, we describe the generation process for four common types of anomalies.

**Definition and Generation Process of Four Common Types of Anomalies** (Han et al., 2022):

- **Local Anomalies:** Local anomalies are points that deviate significantly from their local neighborhoods. To generate local anomalies, we first create synthetic normal samples using the GMM (Reynolds et al., 2009). Then, we scale the covariance matrix by a scaling parameter $\alpha = 5$ to generate local anomalies, which alters the covariance structure of the normal data.

- **Global Anomalies:** Global anomalies are data points that are significantly different from the normal data. These anomalies are generated using a uniform distribution $\text{Unif}(min(X_k), max(X_k))$, where the boundaries are defined by the minimum and maximum values of an input feature $X_k$. The scaling parameter $\alpha = 1.1$ controls how extreme these anomalies are.

- **Dependency Anomalies:** Dependency anomalies refer to samples that do not follow the dependency structure of the normal data. In this case, the input features of the anomalies are assumed to be independent of each other. We apply the Vine Copula method (Aas et al., 2009) to model the dependency structure of the original normal data. The probability density function (PDF) for generating anomalies is set to complete independence by removing the modeled dependency. Notably, because the Vine Copula (Aas et al., 2009) method can be computationally expensive when working with large datasets, we only use 2,000 normal samples to fit this model. After fitting the model, we use Kernel Density Estimation (KDE) (Hastie et al., 2009) to estimate the PDF of the features and generate independent anomaly samples.

- **Cluster Anomalies:** Cluster anomalies, also known as group anomalies, exhibit similar characteristics to clusters. To generate cluster anomalies, we adjust the mean feature vector of the normal samples by a scaling parameter $\alpha = 5$, i.e., shifting the mean of the GMM (Reynolds et al., 2009). This parameter controls the distance between anomaly clusters and the normal data. Then, the anomalies are generated by applying the scaled GMM (Reynolds et al., 2009).

Building on the approach from ADBench, we generate "realistic" synthetic data using a generative model like Gaussian Mixture Model (GMM) (Reynolds et al., 2009), trained on the normal samples of a benchmark dataset. The original anomalies are discarded since their exact types are unknown. By adjusting the parameters of the generative model, we can

*Table 5.* Data description of the 57 datasets included in ADBench original paper.

| No. | Data | # Samples | # Features | # Anomaly | % Anomaly | Category |
|---|---|---|---|---|---|---|
| 1 | ALOI | 49534 | 27 | 1508 | 3.04 | Image |
| 2 | annthyroid | 7200 | 6 | 534 | 7.42 | Healthcare |
| 3 | backdoor | 95329 | 196 | 2329 | 2.44 | Network |
| 4 | breastw | 683 | 9 | 239 | 34.99 | Healthcare |
| 5 | campaign | 41188 | 62 | 4640 | 11.27 | Finance |
| 6 | cardio | 1831 | 21 | 176 | 9.61 | Healthcare |
| 7 | Cardiotocography | 2114 | 21 | 466 | 22.04 | Healthcare |
| 8 | celeba | 202599 | 39 | 4547 | 2.24 | Image |
| 9 | census | 299285 | 500 | 18568 | 6.20 | Sociology |
| 10 | cover | 286048 | 10 | 2747 | 0.96 | Botany |
| 11 | donors | 619326 | 10 | 36710 | 5.93 | Sociology |
| 12 | fault | 1941 | 27 | 673 | 34.67 | Physical |
| 13 | fraud | 284807 | 29 | 492 | 0.17 | Finance |
| 14 | glass | 214 | 7 | 9 | 4.21 | Forensic |
| 15 | Hepatitis | 80 | 19 | 13 | 16.25 | Healthcare |
| 16 | http | 567498 | 3 | 2211 | 0.39 | Web |
| 17 | InternetAds | 1966 | 1555 | 368 | 18.72 | Image |
| 18 | Ionosphere | 351 | 32 | 126 | 35.90 | Oryctognosy |
| 19 | landsat | 6435 | 36 | 1333 | 20.71 | Astronautics |
| 20 | letter | 1600 | 32 | 100 | 6.25 | Image |
| 21 | Lymphography | 148 | 18 | 6 | 4.05 | Healthcare |
| 22 | magic.gamma | 19020 | 10 | 6688 | 35.16 | Physical |
| 23 | mammography | 11183 | 6 | 260 | 2.32 | Healthcare |
| 24 | mnist | 7603 | 100 | 700 | 9.21 | Image |
| 25 | musk | 3062 | 166 | 97 | 3.17 | Chemistry |
| 26 | optdigits | 5216 | 64 | 150 | 2.88 | Image |
| 27 | PageBlocks | 5393 | 10 | 510 | 9.46 | Document |
| 28 | pendigits | 6870 | 16 | 156 | 2.27 | Image |
| 29 | Pima | 768 | 8 | 268 | 34.90 | Healthcare |
| 30 | satellite | 6435 | 36 | 2036 | 31.64 | Astronautics |
| 31 | satimage-2 | 5803 | 36 | 71 | 1.22 | Astronautics |
| 32 | shuttle | 49097 | 9 | 3511 | 7.15 | Astronautics |
| 33 | skin | 245057 | 3 | 50859 | 20.75 | Image |
| 34 | smtp | 95156 | 3 | 30 | 0.03 | Web |
| 35 | SpamBase | 4207 | 57 | 1679 | 39.91 | Document |
| 36 | speech | 3686 | 400 | 61 | 1.65 | Linguistics |
| 37 | Stamps | 340 | 9 | 31 | 9.12 | Document |
| 38 | thyroid | 3772 | 6 | 93 | 2.47 | Healthcare |
| 39 | vertebral | 240 | 6 | 30 | 12.50 | Biology |
| 40 | vowels | 1456 | 12 | 50 | 3.43 | Linguistics |
| 41 | Waveform | 3443 | 21 | 100 | 2.90 | Physics |
| 42 | WBC | 223 | 9 | 10 | 4.48 | Healthcare |
| 43 | WDBC | 367 | 30 | 10 | 2.72 | Healthcare |
| 44 | Wilt | 4819 | 5 | 257 | 5.33 | Botany |
| 45 | wine | 129 | 13 | 10 | 7.75 | Chemistry |
| 46 | WPBC | 198 | 33 | 47 | 23.74 | Healthcare |
| 47 | yeast | 1484 | 8 | 507 | 34.16 | Biology |
| 48 | CIFAR10 | 5263 | 512 | 263 | 5.00 | Image |
| 49 | FashionMNIST | 6315 | 512 | 315 | 5.00 | Image |
| 50 | MNIST-C | 10000 | 512 | 500 | 5.00 | Image |
| 51 | MVTec-AD | | See Table 6 | | | Image |
| 52 | SVHN | 5208 | 512 | 260 | 5.00 | Image |
| 53 | Agnews | 10000 | 768 | 500 | 5.00 | NLP |
| 54 | Amazon | 10000 | 768 | 500 | 5.00 | NLP |
| 55 | Imdb | 10000 | 768 | 500 | 5.00 | NLP |
| 56 | Yelp | 10000 | 768 | 500 | 5.00 | NLP |
| 57 | 20newsgroups | | See Table 7 | | | NLP |

*Table 6.* Detailed description of the MVTec-AD dataset; see the full dataset list in Table 5.

| Class | # Samples | # Features | # Anomaly | % Anomaly |
|---|---|---|---|---|
| Carpet | 397 | 512 | 89 | 22.42 |
| Grid | 342 | 512 | 57 | 16.67 |
| Leather | 369 | 512 | 92 | 24.93 |
| Tile | 347 | 512 | 84 | 24.21 |
| Wood | 326 | 512 | 60 | 18.40 |
| Bottle | 292 | 512 | 63 | 21.58 |
| Cable | 374 | 512 | 92 | 24.60 |
| Capsule | 351 | 512 | 109 | 31.05 |
| Hazelnut | 501 | 512 | 70 | 13.97 |
| Metal Nut | 335 | 512 | 93 | 27.76 |
| Pill | 434 | 512 | 141 | 32.49 |
| Screw | 480 | 512 | 119 | 24.79 |
| Toothbrush | 102 | 512 | 30 | 29.41 |
| Transistor | 313 | 512 | 40 | 12.78 |
| Zipper | 391 | 512 | 119 | 30.43 |
| Total | 5354 | 512 | 1258 | 23.50 |

*Table 7.* Detailed description of the 20newsgroups dataset; see the full dataset list in Table 5.

| Class | # Samples | # Features | # Anomaly | % Anomaly |
|---|---|---|---|---|
| Computer | 3090 | 768 | 154 | 4.98 |
| Recreation | 2514 | 768 | 125 | 4.97 |
| Science | 2497 | 768 | 124 | 4.97 |
| Miscellaneous | 615 | 768 | 30 | 4.88 |
| Politics | 1657 | 768 | 82 | 4.95 |
| Religion | 1532 | 768 | 76 | 4.96 |
| Total | 11905 | 768 | 591 | 4.96 |

create both normal samples and different types of anomalies. This method adds complexity to the generated anomalies, making the synthetic data better reflect the challenges of real-world anomaly detection tasks.

To better highlight the differences between these anomaly types, a visual comparison using t-SNE (Van der Maaten & Hinton, 2008) with the 'letter' dataset is provided in Figure 6. The figure displays the four types of anomalies( local, global, dependency, and cluster) applied to this dataset, helping to understand the data distribution for each anomaly type.

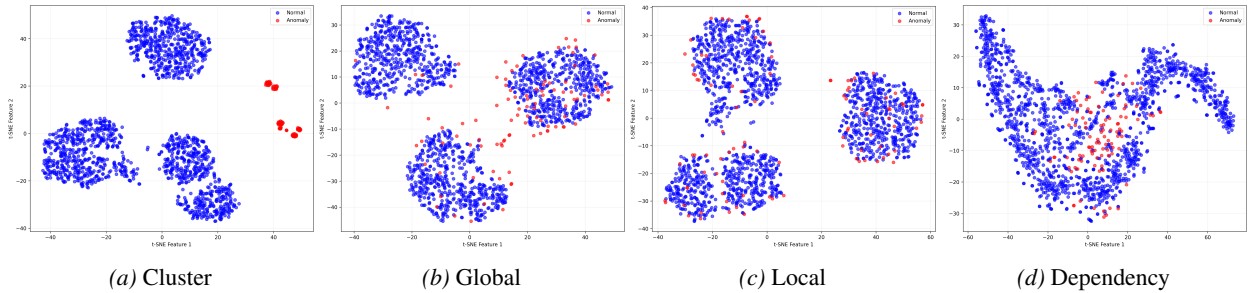

|  |  |  |  |
|---|---|---|---|
| *(a)* Cluster | *(b)* Global | *(c)* Local | *(d)* Dependency |

*Figure 6.* Data distribution visualization of four types of synthetic anomaly shown on the 20_letter dataset

## D.4. Baseline Algorithms

In this section, we present the baseline algorithms employed for anomaly detection. While DVM-AD is related to discriminant/null-space novelty detection, those methods typically assume multi-class supervision and are not standard baselines in tabular one-class AD benchmarks (e.g., PyOD/ADBench). A direct comparison may therefore be non-trivial and could require nonstandard adaptations. Therefore, we chose to benchmark DVM-AD against a large set of established one-class AD baselines and only three related discriminant analysis methods (PCA, KNFST, and PMKFN), which are coded

according to the algorithm presented in their publications. We chose the baseline models based on their top performance on previous benchmarks such as (Han et al., 2022) and some of the latest deep learning research, such as DIF (Xu et al., 2023) or DRLAD (Ye et al., 2025). We categorize these algorithms into two groups: Shallow Machine Learning-based and Deep Learning-based.

### D.4.1. MACHINE LEARNING-BASED ANOMALY DETECTION ALGORITHMS

- **Clustering-Based Local Outlier Factor (CBLOF)** (He et al., 2003): CBLOF identifies anomalies by first partitioning the data into clusters and then evaluating the distance between the data points and the cluster centroids. Instances that exhibit significant distance from their respective clusters are considered outliers.

- **K-Nearest Neighbors (KNN)** (Peterson, 2009): KNN calculates anomaly scores based on the distance to the $k$-th nearest neighbor. Data points with larger distances from their nearest neighbors are assigned higher anomaly scores, suggesting that they are potential outliers.

- **Isolation Forest (IForest)** (Liu et al., 2008): Isolation Forest isolates observations by recursively partitioning the data using random features and split values. The anomalies, which are few and distinct, are isolated more efficiently, resulting in shorter path lengths.

- **One-Class Support Vector Machine (OCSVM)** (Schölkopf et al., 2001): OCSVM constructs a decision boundary in a high-dimensional space, separating normal data points from outliers by maximizing the margin between the origin and the normal data points. The decision boundary thus defines the region where anomalies are unlikely to reside.

- **Local Outlier Factor (LOF)** (Breunig et al., 2000): LOF measures the local density deviation of a data point in relation to its neighbors. Points exhibiting substantially lower density compared to their neighbors are considered anomalies.

- **Histogram-Based Outlier Detection (HBOS)** (Goldstein & Dengel, 2012): HBOS assumes the independence of the features and constructs histograms for each feature. Anomalies are identified based on the deviation of a data point from the expected distribution represented by these histograms.

- **Principal Component Analysis (PCA)** (Shyu et al., 2003): PCA is employed to reduce the dimensionality of the data by projecting it onto a lower-dimensional space. Anomalies are detected by evaluating the reconstruction error, with larger errors indicating a poor fit to the reduced space and suggesting outlier behavior.

- **Subspace Outlier Detection (SOD)** (Kriegel et al., 2009): SOD detects outliers by examining various subspaces of the feature space. Anomalies are identified based on their deviation from expected behavior in one or more of these subspaces.

- **Connectivity-Based Outlier Factor (COF)** (Tang et al., 2002): COF measures the chaining distance of a data point and its k-th nearest neighbors. Data points with unusually large or small chaining distances relative to their neighbors are classified as anomalies.

- **Lightweight Online Detector of Anomalies (LODA)** (Pevnỳ, 2016): LODA is an ensemble-based anomaly detection algorithm particularly suitable for real-time anomaly detection in streaming data. It is capable of adapting to concept drift and large-scale data environments.

- **Empirical Cumulative Distribution-Based Outlier Detection (ECOD)** (Li et al., 2022): ECOD uses the empirical cumulative distribution function (ECDF) to model the distribution of each feature. Anomalies are detected by identifying data points that lie in the extreme tails of the distribution.

- **Copula-Based Outlier Detector (COPOD)** (Li et al., 2020): COPOD is a model-free, highly interpretable anomaly detection technique that uses copula-based models to capture the dependence structure between features. Detects anomalies in multivariate data without requiring hyperparameter tuning.

- **Accelerating Large-scale Unsupervised Heterogeneous Outlier Detection (SUOD)** (Zhao et al., 2021): SUOD is a scalable and efficient framework designed to handle large-scale unsupervised outlier detection. It leverages a combination of diverse base detectors (e.g., kNN, LOF, PCA) and accelerates computation through model approximation, parallel execution, and task pruning. SUOD achieves a good balance between detection accuracy and computational efficiency, making it suitable for large and high-dimensional datasets.

- **$\ell_p$-Norm Multiple Kernel One-Class Fisher Null-Space (PMKFN)** (Arashloo, 2020): PMKFN is a multiple-kernel extension of the one-class Fisher null-space classifier. It learns a nonnegative weighted combination of several base kernels under an $\ell_p$-norm constraint, and optimizes the resulting one-class objective via a saddle-point (min–max) formulation. In the induced RKHS, the learned projection collapses normal training samples to a constant target value (with the origin acting as a pseudo negative), so test points that deviate from this target in the projected space receive higher anomaly scores.

- **Kernel Null Foley–Sammon Transform (KNFST)** (Ali & Chaudhuri, 2022): KNFST is a kernelized null-space discriminant projection method that constructs projection directions with (near-)zero within-class scatter in RKHS, effectively collapsing training samples of a class to a single point in the null space. This makes novelty/anomaly detection simple: a test sample is scored by its distance (or absolute deviation) from the normal-class target point in the null space, without requiring explicit density estimation in the projected space. The projection is obtained in closed form (no iterative end-to-end training of the subspace itself), with the main modeling choice being the kernel.

### D.4.2. DEEP LEARNING-BASED ANOMALY DETECTION ALGORITHMS

- **Deep Support Vector Data Description (DeepSVDD)** (Ruff et al., 2018): DeepSVDD uses a deep neural network to learn representations of the data while minimizing the volume of a hypersphere that encloses these representations. Data points that lie far from the center of this hypersphere are identified as anomalies.

- **AutoEncoder** (Bank et al., 2023): AutoEncoders are neural networks designed to learn efficient representations of data through a process of encoding and decoding. Anomalies are detected based on reconstruction errors, with higher errors indicating instances that do not conform to the learned data distribution.

- **Deviation Networks (DevNet)** (Pang et al., 2019): DevNet is a deep learning model that enforces a deviation-based loss function to penalize large deviations in the learned representations. This method is particularly effective in identifying outliers by ensuring that anomalous instances exhibit substantial deviations from the learned data distribution.

- **Unifying Local Outlier Detection Methods via Graph Neural Networks** (Goodge et al., 2022): LUNAR is a deep learning-based, neural network approach designed for unsupervised anomaly detection. It learns a data distribution through representation learning and detects anomalies by identifying instances that deviate significantly from the normal patterns learned. This method is particularly effective in high-dimensional, complex data environments where labeled data is unavailable.

- **Autoencoder-based One-Class Support Vector Machine (AE1SVM)** (Nguyen & Vien, 2018): AE1SVM is a hybrid method combining AutoEncoder-based feature learning with Support Vector Machines for enhanced anomaly detection. This approach leverages the reconstruction errors from the AutoEncoder in conjunction with SVM's classification capabilities to detect outliers.

- **Adversarial Learned Anomaly Detection (ALAD)** (Zenati et al., 2018): ALAD is a deep learning model that utilizes adversarial learning techniques for anomaly detection. The architecture is designed to capture the subtle differences between normal and anomalous instances by learning the underlying data distribution.

- **Anomaly Detection with Generative Adversarial Networks (AnoGAN)** (Schlegl et al., 2017): AnoGAN is one of the first approaches to apply GANs for unsupervised anomaly detection. It learns a generator to model the distribution of normal data and identifies anomalies by searching for a latent vector whose generated sample best matches a test sample, using reconstruction and discriminator scores.

- **Variational AutoEncoder (VAE)** (Kingma et al., 2013): VAE is a probabilistic generative model that learns a latent representation of data by maximizing a variational lower bound on the data likelihood. Anomalies are detected by measuring reconstruction error or likelihood under the learned latent distribution.

- **Single-Objective Generative Adversarial Active Learning (SO-GAAL)** (Liu et al., 2019): SO-GAAL introduces a generative adversarial framework to generate informative potential outliers from a single objective function. These synthetic outliers help train a discriminator that distinguishes between real (normal) and generated (outlier-like) samples.

- **Deep Isolation Forest for Anomaly Detection (DIF)** (Xu et al., 2023): DIF integrates the Isolation Forest principle with deep neural representations. It learns hierarchical feature spaces where instances are separated by randomly partitioning the transformed space, enhancing the separability between normal and anomalous instances.

- **Deep Robust One-Class Classification (DROCC)** (Goyal et al., 2020): DROCC is a deep learning-based one-class classification method that employs adversarial perturbations around the normal data manifold to generate synthetic negatives. The model is trained to discriminate between true normal samples and perturbed outliers, enhancing robustness to unseen anomalies.

- **Neural Transformation Learning for Anomaly Detection (NeuTraLAD)** (Qiu et al., 2021): NeuTraLAD learns a nonlinear transformation that maps the normal data distribution into a simple, modelable form (e.g., Gaussian). Anomalies are then detected based on their deviation from the transformed normal distribution using likelihood-based scoring.

- **Decomposed Representation Learning Anomaly Detection (DRLAD)** (Ye et al., 2025): DRLAD is a deep tabular anomaly detection method that remaps inputs into a constrained latent space to reduce representation entanglement between normal and anomalous patterns. It enforces each normal latent representation to be expressible as a sample-specific weighted linear combination of fixed, randomly generated orthogonal basis vectors, and further strengthens separation between normal and anomalous patterns through an additional discrepancy-amplifying constraint (plus an alignment term that injects feature-correlation information). In inference, DRLAD uses the latent-space decomposition error (distance between the learned representation and its basis reconstruction) as the anomaly score.

- **Masked Cell Modeling (MCM)** (Yin et al., 2024): MCM is a self-supervised tabular anomaly detector that learns feature correlations of normal data through a learnable mask-generator module. The model is trained to reconstruct randomly masked cells of normal samples; a diversity loss encourages the generator to produce complementary masks that cover different feature subsets. At inference, anomalies are scored by the reconstruction error aggregated over multiple masks. The method targets dependency-style anomalies where the correlation structure among features is informative.

- **Non-Parametric Transformer for Anomaly Detection (NPT-AD)** (Thimonier et al., 2024a): NPT-AD is a reconstruction-based deep tabular detector built on Non-Parametric Transformers, which model both feature–feature and sample–sample interactions in a single architecture. It is trained to reconstruct masked features of normal samples and leverages the full training set non-parametrically during inference. The anomaly score is derived from the per-feature reconstruction error aggregated across mask patterns, allowing the detector to combine within-sample feature dependencies with cross-sample retrieval-style cues.

- **Retrieval Augmented Anomaly Detection (RAAD)** (Thimonier et al., 2024b): RAAD augments a reconstruction-based deep tabular detector with a retrieval module that conditions reconstruction on nearest-neighbour training samples. The model learns to reconstruct masked or perturbed features of a query while attending to retrieved normal samples; the anomaly score aggregates the reconstruction discrepancy. The retrieval component is intended to introduce sample-sample dependencies on top of feature-feature reconstruction signals.

- **Diffusion Time Estimation (DTE)** (Livernoche et al., 2024): DTE is a diffusion-based anomaly detector that, rather than running a full denoising chain, directly estimates the distribution over diffusion time corresponding to a given input. A neural network predicts an analytical form of this distribution, and the mode (or mean) of the predicted diffusion time is used as the anomaly score: inputs that "look like" they came from a heavily noised stage of the forward process receive high scores. The approach is reported to be substantially faster than full DDPM-based anomaly detection while remaining competitive on the ADBench benchmark.

- **Prototype-oriented Tabular Anomaly Detection (PTAD)** (Lu et al., 2025): PTAD combines mask modeling and prototype learning within a reconstruction-based one-class framework. The encoder applies masking in both the data space and a projection space spanned by orthogonal basis vectors; the decoder reconstructs the masked representations in parallel and learns a set of association prototypes that capture global normal correlations. Both projection-space learning and prototype matching are formulated as optimal-transport problems, and the resulting calibration distances are used to refine the anomaly score.

- **GANomaly** (Akcay et al., 2018): GANomaly is an adversarially trained one-class anomaly detector originally proposed for images. It uses an encoder–decoder–encoder generator paired with a discriminator and is trained on normal data only. At inference, the anomaly score combines pixel/feature reconstruction error with the latent residual between the two encoders. We include GANomaly as a representative GAN-based generative anomaly detector applied to tabular features under a default-setting protocol.

- **Time-Conditioned Contraction Matching (TCCM)** (Li et al., 2025): TCCM is a flow-matching-inspired semi-supervised tabular anomaly detector. It learns a time-conditioned velocity field that, for normal data, contracts samples toward a fixed target (the origin) over a continuous time variable. At inference, an input is scored by how much its predicted velocity field deviates from the expected one-step contraction behaviour at a sampled time, requiring only a single forward pass rather than a full ODE solve. The score function operates in the input space, which the authors leverage for feature-wise attribution.

- **Anomaly Detection by an Ensemble of Random Pairs of Hyperspheres (ADERH)** (Durani et al., 2025): ADERH is an isolation-based detector that builds an ensemble of hyperspheres of varying radii from randomly paired data points. Each hypersphere yields a local anomaly score that is refined by two complementary signals: *Pitch*, which emphasises points lying near hypersphere boundaries, and *NDensity*, which down-weights hyperspheres centred in sparse regions. Final anomaly scores are obtained by averaging across the ensemble, which is intended to mitigate variance and to remain effective in higher-dimensional regimes than classical isolation methods.

- **Internal Contrastive Learning (ICL)** (Shenkar & Wolf, 2022): ICL is a deep one-class anomaly detector for tabular data. For each sample, ICL splits the features into two disjoint parts and learns mappings that maximise the mutual information between the two parts via a contrastive objective evaluated on a single sample at a time. The detector is trained on normal data only; at test time, a sample is scored by the resulting contrastive loss between its halves, with high loss indicating an anomaly. The method is reported to give state-of-the-art results across multiple tabular benchmarks using a fixed default hyperparameter setting.

## D.5. Hyperparameter Setting

**Out-of-the-box evaluation protocol:** Our goal is to assess out-of-the-box performance in the one-class setting, where labeled validation data for tuning is typically unavailable. Therefore, we run all baselines using their standard library defaults (PyOD) without performing any dataset-specific hyperparameter tuning. For DVM-AD, we fix the selection tolerance to a single value across all datasets and settings, $\epsilon_{\text{sel}} = 10^{-1}$ (i.e., 0.1). The other tolerances are fix to $10^{-8}$ to avoid division by zero.

## D.6. Extended Experimental Results

### D.6.1. DETECTION PERFORMANCE IN REAL-WORLD CONTEXTS

In our study, we not only conducted experiments to evaluate the models using the AUROC metric, as discussed in Section 5.5.1, but also performed evaluations using the AUPRC metric. This subsection presents the performance of DVM-AD according to the AUPRC metric.

From the rank plot (Figure 7), it is evident that our proposed DVM-AD achieves the highest detection performance with an average AUPRC value of 67.45%, outperforming other algorithms such as LUNAR (60.14%) by 7.31% and KNN (56.72%) by 10.73%. Additionally, in the ranking statistics, our model holds the top position with a score of 3.38, followed by the second-place algorithm LUNAR at 6.79 (a difference of 3.41). The next best competitor, KNN, ranks at 7.91, demonstrating a substantial lead of 4.53 in rank for DVM-AD. These results demonstrate that the proposed approach consistently excels in terms of ranking performance.

Furthermore, the boxplot (Figure 8) demonstrates the outstanding detection performance of DVM-AD compared to other models. Our algorithm achieves the highest median AUPRC among all benchmarked models, highlighting its strong ability to detect anomalies. Furthermore, the AUPRC range shown in the box plot spans from around 50% (the lowest point) to approximately 99%, which represents the highest lower bound among all models, emphasizing the stability of DVM-AD. This means that even in the worst-case scenario, DVM-AD performs better than the other models, further demonstrating its reliability across different datasets.

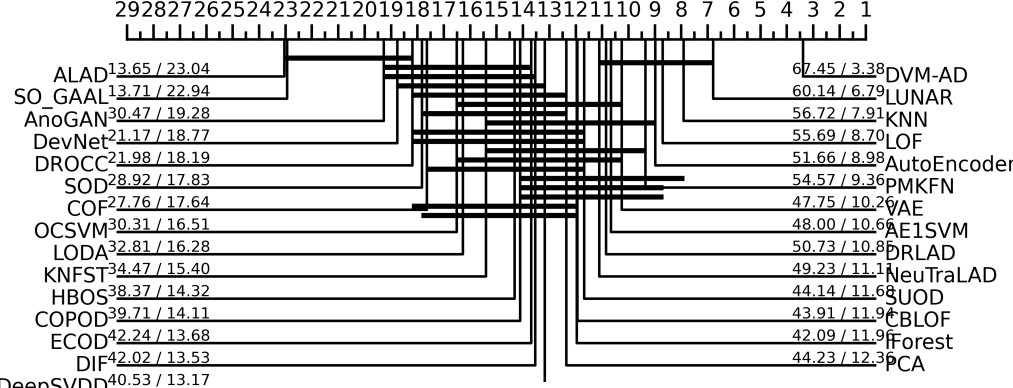

*Figure 7.* The average rank of various algorithms based on their AUPRC values, evaluating across 47 real-world tabular datasets. The results are consistent with the results shown in Section 5.5 that DVM-AD outperforms other algorithms.

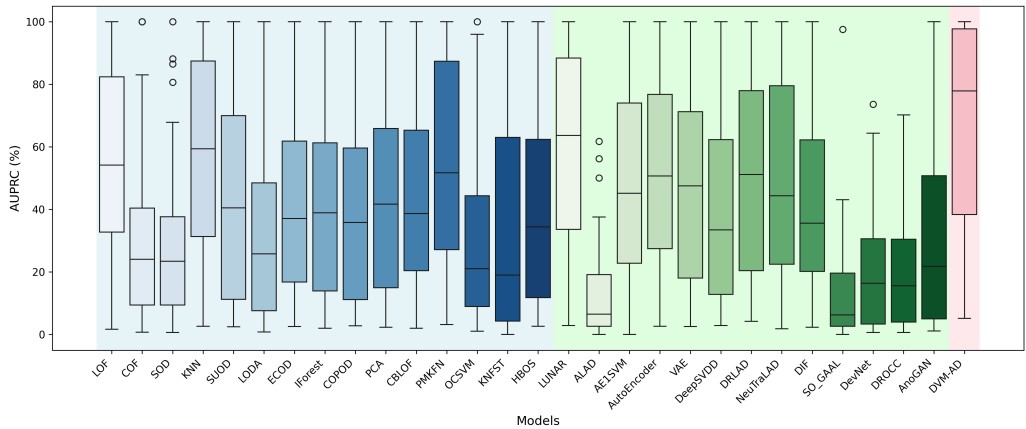

*Figure 8.* The distribution of AUPRC values across 47 real-world tabular datasets. The results consistently show that DVM-AD outperforms other algorithms.

### D.6.2. DETECTION PERFORMANCE IN CROSS-DOMAIN EXTENSION

Besides benchmarking DVM-AD on tabular data, we also extend our experiment to assert its detection performance on other critical domains such as text and image.

**Reporting protocol for embedding benchmarks.** Each embedding dataset is decomposed into multiple one-vs-rest class-novelty tasks. We compute AUROC/AUPRC on each task and report macro-averages grouped by the 10 embedding data sources.

Figure 9 shows DVM-AD's outstanding performance on NLP and CV domains when achieving the lowest average rank of 1.6 and the highest average AUROC of 72.68%. Figure 10 shows that DVM-AD achieves the highest detection performance with an average AUPRC value of 26.92%, outperforming other algorithms such as LOF (25.67%) and LUNAR (23.31%). Additionally, in the ranking statistics, our model holds the top position with a lowest average rank of 2.10, followed by the second-place algorithm LOF at 4.80.

Furthermore, the boxplots (Figure 11 and 12 ) demonstrate the outstanding detection performance of DVM-AD compared to other models. The AUPRC range shown in the box plot also shows that DVM-AD achieves a higher performance than other baselines. This means that even in the worst-case or best-case scenario, DVM-AD performs better than the other models. This indicates the application potential of DVM-AD not only in processing tabular data but also in other critical domains.

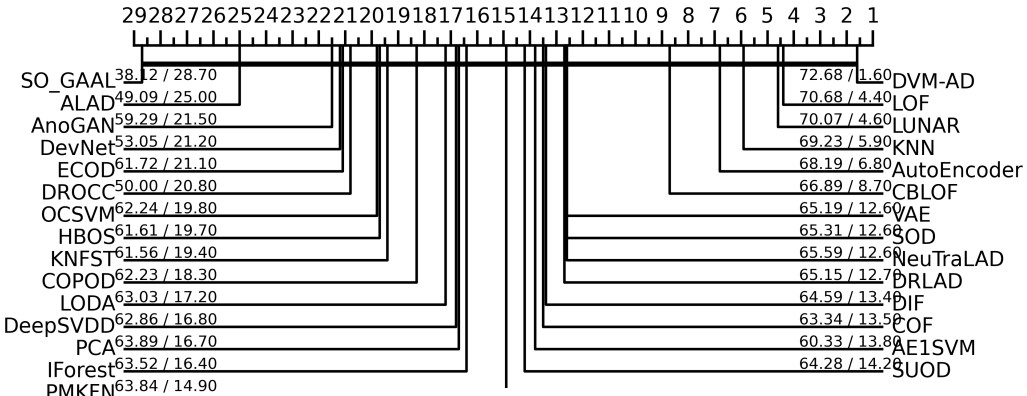

*Figure 9.* The average AUROC value and average rank of algorithms within 10 real-world NLP and CV datasets.

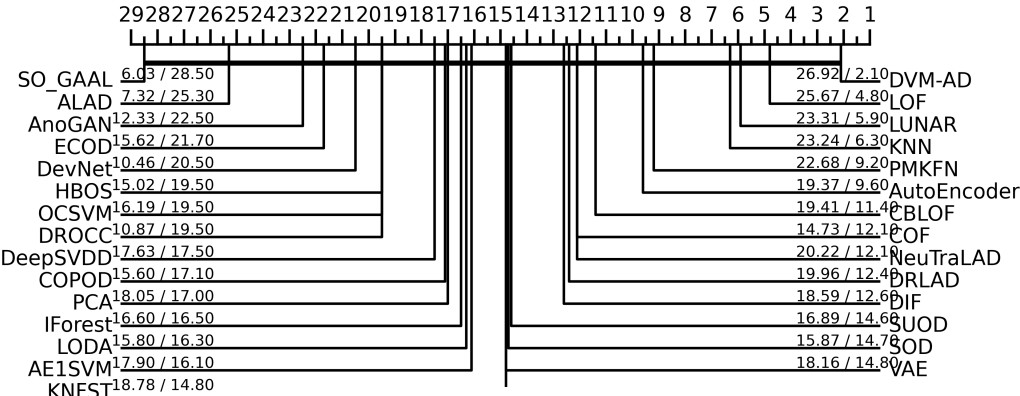

*Figure 10.* The average rank of various algorithms based on their AUPRC values, evaluating across 10 real-world text and image datasets.

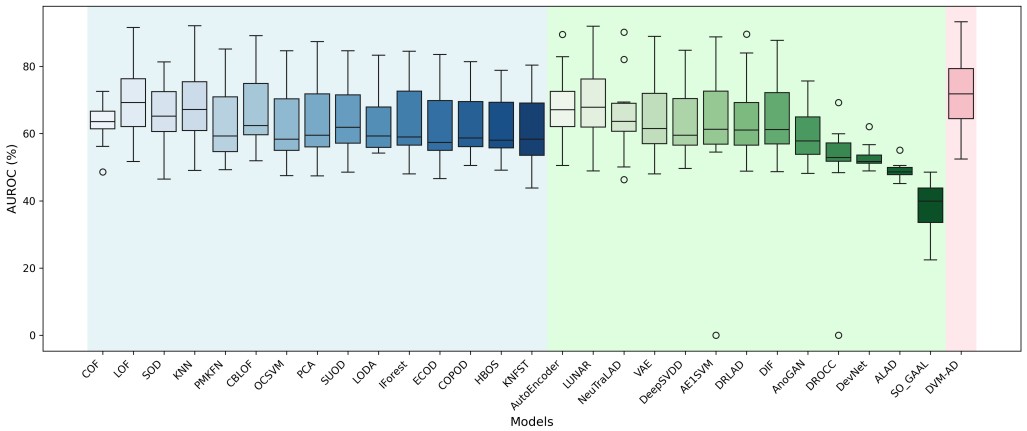

*Figure 11.* The distribution of AUROC values across 10 real-world text and image datasets.

### D.6.3. Experimental Results on Detecting Different Anomaly Types

In this section, we present the experimental results of DVM-AD and baseline methods for four types of anomalies.

The experiment results show that DVM-AD outperforms other baseline algorithms by achieving the best average ranking AUPRC across all types of anomalies. In more detail, DVM-AD significantly outperforms other baselines in terms of

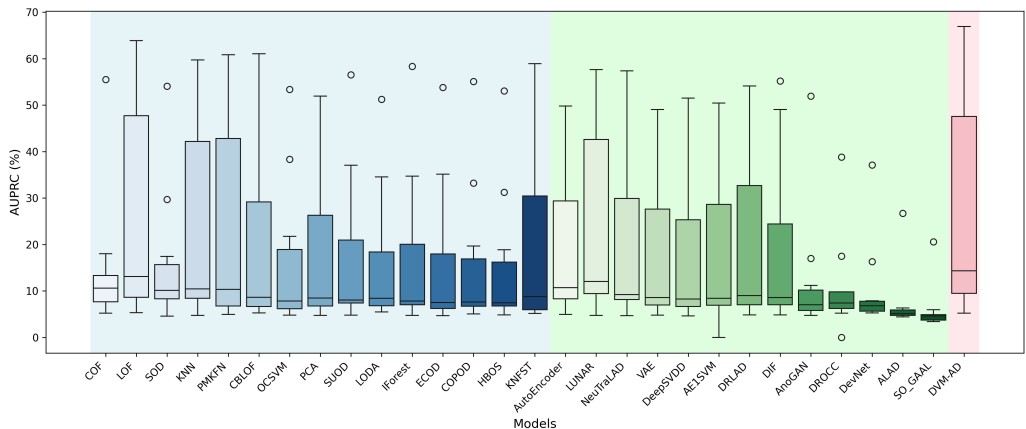

*Figure 12.* The distribution of AUPRC values across 10 real-world text and image datasets.

dependency and local anomaly, achieving the corresponding best average AUPRC of 68.73% and 73.60%, respectively, and presenting a gap of over 4% compared to baseline models. DVM-AD also achieves the best average AUPRC in global anomaly, but is slightly inferior (by 0.34%) to the best model (LUNAR) on cluster anomaly. However, DVM-AD still achieves the best average ranking across all types of anomalies.

Besides achieving high accuracy, the boxplots in Figures 14 highlight the stability of DVM-AD (Ours) and demonstrate its outstanding detection performance compared to all baselines. It achieves the highest median score in the AUROC, while significantly reducing variability, as indicated by the compact interquartile range.

### D.6.4. EXPERIMENTAL RESULTS ON RUNNING TIME

In this section, we provide further details on the experimental results regarding the running time, supporting the claims made in Section 1 and Section 5.5.3.

The experimental results, as shown in the boxplots in Figure 15, highlight the significant computational efficiency of the proposed model compared to other methods. While many machine learning models, such as KNN and OCSVM, exhibit lower training times compared to deep learning models, DVM-AD consistently achieves significantly shorter training times. The training time range spans from approximately $10^{-3}$ to nearly 1 seconds, while deep learning models range from about $10^{-1}$ to roughly $10^3$ seconds. Besides, our algorithm's median testing time is positioned around $0.5 \times 10^{-2}$, with the overall range spanning from $10^{-2}$ to $10^0$. Although the median testing time is still higher than some models, it remains competitive.

**Extreme-scale stress tests (against 13 baselines).**  To probe behaviour outside the ADBench size range, we ran an additional benchmark on two synthetic regimes: Setting A ($N{=}10^6, d{=}10^3$, sample-stress) and Setting B ($N{=}10^5, d{=}10^4$, feature-stress), against 13 representative baselines (7 ML on CPU, 6 DL on GPU). Table 8 reports mean training time over 5 seeds. DVM-AD is the fastest method on Setting A ($32.5$ s, against the second-fastest HBOS at $43.7$ s). On Setting B the $O(d^3)$ eigendecomposition cost becomes visible ($496$ s), but DVM-AD remains operable while 7/13 baselines fail with out-of-memory errors. The result is consistent with the complexity analysis in Section 4.5 and supports the PCA pre-reduction option (Appendix F.4) as the recommended remedy in the extreme-$d$ regime.

### D.7. Oracle Best-of-Grid Analysis for Strong Baseline Families

Most of our experiments follow a strictly no-tuning protocol, evaluating all baselines with their library default configurations to reflect practical one-class AD, where labeled validation data for tuning is typically unavailable.

To address the concern that some strong but hyperparameter-sensitive baselines may be underestimated under default settings, we include an additional oracle-best-of-grid analysis of the three best baselines (KNN, IForest, and LUNAR). In this analysis, we evaluate a modest hyperparameter grid and report the best AUROC achieved on the evaluation split for each dataset. Because selecting the best configuration in this manner uses label information from the evaluation split, this is not a deployable one-class model-selection protocol; rather, it should be interpreted as an optimistic upper bound on how

*Table 8.* Training time (seconds) on two extreme-scale synthetic regimes with 13 baselines (7 ML on CPU, 6 DL on GPU); DVM-AD runs on CPU. Mean $\pm$ std over 5 seeds. OoM = out-of-memory. Methods are sorted by Setting A time, with OoM-on-A rows last.

| Method | Type | Setting A: $N{=}10^6, d{=}10^3$ (s) | Setting B: $N{=}10^5, d{=}10^4$ (s) |
|---|---|---|---|
| **DVM-AD** | ML/CPU | **32.5** $\pm$ 3.1 | **496.0** $\pm$ 1.9 |
| HBOS | ML/CPU | 43.7 $\pm$ 0.8 | 47.0 $\pm$ 0.7 |
| KNFST | ML/CPU | 48.2 $\pm$ 2.3 | 38.0 $\pm$ 0.7 |
| ECOD | ML/CPU | 142.5 $\pm$ 0.3 | 147.3 $\pm$ 4.1 |
| COPOD | ML/CPU | 156.0 $\pm$ 11.4 | 145.6 $\pm$ 0.2 |
| PCA | ML/CPU | 397.3 $\pm$ 18.2 | OoM |
| ContrastiveAD (ICL) | DL/GPU | 680.8 $\pm$ 1.4 | OoM |
| ADERH | ML/CPU | 845.9 $\pm$ 14.9 | 573.9 $\pm$ 1.8 |
| DTE | DL/GPU | 1856.3 $\pm$ 714.2 | 140.6 $\pm$ 5.6 |
| MCM | DL/GPU | 3403.4 $\pm$ 238.1 | OoM |
| AutoEncoder | DL/GPU | OoM | OoM |
| LUNAR | DL/GPU | OoM | OoM |
| ALAD | DL/GPU | OoM | OoM |
| OCSVM | ML/CPU | OoM | OoM |

much these baselines could improve with careful hyperparameter choice. Importantly, even under this optimistic baseline advantage, DVM-AD remains the top method on average (Table 9).

### D.7.1. STRONG BASELINES, TUNING, AND MULTI-SEED EVALUATION

Our main comparison emphasizes an out-of-the-box setting. On the real-world tabular benchmark (without tuning), we observe that KNN and LUNAR are the closest competitors to DVM-AD, i.e., they rank immediately behind DVM-AD among the considered methods. Motivated by this observation, we select KNN (a proximity-based detector) and LUNAR (a deep detector) for hyperparameter tuning, since they represent the strongest baselines from the proximity and deep branches under the default setting.

In addition, we tune Isolation Forest (IForest) as a canonical tree-based baseline to cover the third major branch of AD methods and to test whether tuning can substantially improve tree-based approaches under the same one-class protocol.

For each dataset, we follow the same one-class training pipeline as in our main experiments: we train only on normal samples from the training split and evaluate on the held-out evaluation split, which contains both normal and anomalous samples. We then run a modest grid of hyperparameters for each baseline family and record the best-of-grid AUROC on that evaluation split. Finally, we compute per-dataset ranks (rank 1 is best; ties use the minimum rank) and report the average rank across datasets. We emphasise again that this selection is oracle (it uses labels from the evaluation split) and is included only to quantify an optimistic upper bound for these hyperparameter-sensitive baselines; it is not intended as a recommended one-class tuning recipe.

**Hyperparameter tuning setup.** To assess whether careful tuning can overturn the real-world out-of-the-box ranking, we tune three strong baselines representing the major AD branches: KNN (proximity-based), IForest (tree-based), and LUNAR (deep). For each dataset, we follow the same one-class protocol as in the main experiments: we create a stratified 70/30 train/test split, train the detector using normal-only samples from the training split, and evaluate anomaly scores on the test split containing both normal and anomalous instances. To keep the computational budget bounded and comparable across datasets, we cap each dataset to at most 10,000 samples (with stratified subsampling when necessary).

**Search space.** We use a moderate grid search with three hyperparameters per model. For KNN, we tune $(i)$ `n_neighbors` $\in \{5, 10, 20\}$, $(ii)$ `metric` $\in \{$`minkowski`, `euclidean`, `manhattan`$\}$, and $(iii)$ the Minkowski order `p` $\in \{1, 2, 3\}$, while keeping `method=largest` fixed. This yields $3 \times 3 \times 3 = 27$ configurations. For IForest, we tune `n_estimators` $\in \{100, 200, 400\}$ and `max_samples` $\in \{$`auto`, 128, 256$\}$, (resulting in 9 configurations). For LUNAR, we tune `n_neighbours` $\in \{10, 50\}$, `epsilon` $\in \{0.05, 0.2\}$, and `proportion` $\in \{1.0, 2.0\}$, resulting in $2 \times 2 \times 2 = 8$ configurations, while fixing other training-related settings to standard values. The grid sizes are intentionally modest to

bound compute and should be interpreted as a controlled stress test; because selection is oracle, these results are an optimistic estimate of what these baselines could achieve under per-dataset hyperparameter choice.

### D.7.2. Multi-Seed Robustness With Frozen Oracle-Selected Configurations

**Why multi-seed is non-trivial at our benchmark scale.**   Our overall evaluation spans 57 data sources (47 tabular datasets and 10 embedding sources), which are decomposed into 121 evaluation tasks. Combined with 28 baselines, running a full multi-seed evaluation would require on the order of $121 \times 28 \times 5 \approx 16,940$ train–test runs even without hyperparameter search, and would be substantially larger if we re-tuned hyperparameters per seed.

**Protocol (5 seeds).**   To address concerns about sensitivity to random splits while keeping the study feasible, we perform a targeted multi-seed robustness study on the 47 real-world tabular datasets for DVM-AD and the three tuned strong baseline families (KNN, IForest, and LUNAR). For each dataset and each seed in $\{0, 1, 2, 3, 4\}$, we: (i) cap the dataset to 10,000 instances using stratified subsampling with a fixed subsampling seed; (ii) generate a stratified 70/30 train/test split using the given seed; (iii) train in the one-class setting (training uses only normal samples from the training split); (iv) evaluate AUROC on the held-out test split containing both normal and anomalous instances. We then report mean±std across the 5 seeds. The mean rank is computed from per-dataset means (averaged over seeds), where rank 1 is best and ties take the minimum rank.

**Frozen oracle-selected configurations (no re-tuning per seed).**   For tuned baselines, we freeze the best hyperparameter configuration found in the tuning phase for each dataset and reuse the same configuration across all 5 seeds. This avoids the prohibitively expensive practice of re-tuning for every split and prevents per-seed hyperparameter selection from inflating the reported robustness results. DVM-AD is evaluated with the same default setting as in our main experiments (no tuning), now repeated over 5 seeds.

Table 9 summarizes AUROC on 47 tabular datasets, reported as mean±std over 5 random seeds. For the overall summary, we compute the mean AUROC by averaging the per-dataset means, and the mean rank by ranking methods within each dataset by mean AUROC (higher is better; ties receive average ranks) and then averaging ranks across datasets. To validate that performance differences are significant across datasets, we apply a Friedman test on the 47 per-dataset mean AUROCs over the four methods, which rejects the null hypothesis of equal performance ($\chi^2(3) = 53.29$, $p = 1.59 \times 10^{-11}$). We then perform post-hoc one-sided Wilcoxon signed-rank tests comparing DVM-AD against each baseline and apply Holm correction over the three comparisons; DVM-AD significantly outperforms IForest, KNN, and LUNAR (Holm-adjusted $p = 1.72 \times 10^{-8}$, $1.51 \times 10^{-3}$, and $1.58 \times 10^{-5}$, respectively), consistent with its best overall mean AUROC (89.01%) and lowest mean rank (1.63).

*Table 9.* AUROC(%) on 47 tabular datasets (mean±std over 5 seeds). Best per dataset is in bold. For baselines, hyperparameters are frozen per dataset based on the oracle best-of-grid procedure and then evaluated across seeds. DVM-AD uses a single fixed configuration across all datasets.

| Dataset | IForest | KNN | LUNAR | DVM-AD (ours) |
|---|---|---|---|---|
| 1_ALOI | 54.83±2.03 | 61.18±2.30 | 68.78±2.70 | **73.05±1.67** |
| 2_annthyroid | 91.23±0.78 | **93.89±0.54** | 88.00±1.06 | 84.41±3.53 |
| 3_backdoor | 77.87±2.31 | 92.60±0.53 | 96.31±0.56 | **98.48±0.45** |
| 4_breastw | **99.46±0.34** | 99.39±0.33 | 98.98±0.51 | 98.94±0.60 |
| 5_campaign | 74.38±1.01 | 79.26±0.47 | 72.28±1.67 | **80.29±1.02** |
| 6_cardio | 95.48±1.59 | 94.39±1.50 | 96.71±1.36 | **97.03±1.09** |
| 7_Cardiotocography | 81.44±1.40 | 74.63±1.86 | 81.99±2.29 | **85.51±1.24** |
| 8_celeba | 71.12±2.90 | **79.51±2.71** | 65.01±2.57 | 68.48±3.30 |
| 9_census | 58.74±3.99 | **71.09±1.09** | 66.29±1.26 | 68.01±1.61 |
| 10_cover | 84.37±5.79 | 95.92±1.33 | **97.45±0.86** | 97.39±0.69 |
| 11_donors | 88.75±2.09 | 98.98±0.12 | 99.96±0.05 | **99.99±0.01** |
| 12_fault | 66.00±3.08 | 81.71±1.90 | 81.50±3.24 | **82.70±2.13** |
| 13_fraud | 89.85±8.04 | **91.19±7.49** | 90.15±8.86 | 90.87±8.25 |
| 14_glass | 80.54±3.21 | 87.20±3.99 | 88.06±4.66 | **92.69±2.45** |
| 15_Hepatitis | 79.25±7.79 | **84.50±5.05** | 71.75±13.77 | 83.50±10.13 |
| 16_http | 99.42±0.22 | 99.95±0.03 | **100.00±0.00** | **100.00±0.00** |
| 17_InternetAds | 41.52±2.83 | 81.25±3.07 | 83.30±4.67 | **91.65±1.44** |
| 18_Ionosphere | 89.74±2.18 | **97.27±0.62** | 96.90±1.27 | 95.51±1.42 |
| 19_landsat | 60.26±3.19 | 78.04±0.84 | 78.19±1.06 | **79.90±0.98** |
| 20_letter | 63.38±2.83 | 88.32±2.51 | 91.60±2.79 | **93.54±2.07** |
| 21_Lymphography | 99.30±1.04 | 99.30±0.64 | 99.30±0.64 | **100.00±0.00** |
| 22_magic.gamma | 76.97±1.01 | 83.82±0.53 | 84.77±0.30 | **85.01±0.60** |
| 23_mammography | 86.38±1.36 | 86.05±1.56 | 85.33±2.22 | **89.26±2.03** |
| 24_mnist | 86.91±1.42 | 94.48±0.48 | 93.11±0.36 | **97.62±0.21** |
| 25_musk | 90.32±5.52 | **100.00±0.00** | **100.00±0.00** | **100.00±0.00** |
| 26_optdigits | 84.60±3.56 | 97.77±0.35 | 99.82±0.09 | **99.88±0.07** |
| 27_PageBlocks | 92.58±0.67 | **96.53±0.66** | 94.61±0.75 | 95.22±0.72 |
| 28_pendigits | 96.91±0.66 | 99.89±0.05 | 99.92±0.03 | **99.95±0.01** |
| 29_Pima | 72.84±2.70 | **74.53±3.06** | 72.87±3.43 | 70.98±4.46 |
| 30_satellite | 79.61±2.15 | 87.72±0.52 | 87.98±0.30 | **88.16±0.55** |
| 31_satimage-2 | 99.09±0.55 | **99.90±0.06** | 99.68±0.29 | 99.83±0.13 |
| 32_shuttle | 99.65±0.17 | 99.93±0.05 | 99.91±0.07 | **99.97±0.02** |
| 33_skin | 89.04±0.52 | 98.85±0.12 | 98.27±0.41 | **99.58±0.12** |
| 34_smtp | 86.71±15.43 | 91.36±11.86 | 91.40±11.78 | **93.05±10.46** |
| 35_SpamBase | 84.07±1.13 | **85.99±1.18** | 82.86±1.92 | 81.78±1.20 |
| 36_speech | 51.41±4.63 | 53.33±2.53 | 60.74±3.39 | **82.26±0.60** |
| 37_Stamps | 93.95±1.91 | 94.53±1.52 | 95.27±2.27 | **97.72±0.99** |
| 38_thyroid | 99.13±0.27 | **99.14±0.21** | 98.45±0.80 | 98.01±0.75 |
| 39_vertebral | 37.43±4.93 | 39.29±8.22 | 40.46±10.45 | **57.39±6.11** |
| 40_vowels | 78.29±4.35 | 97.73±0.81 | **99.44±0.47** | 98.78±1.13 |
| 41_Waveform | 75.14±1.42 | 76.82±5.17 | 76.29±4.69 | **80.27±7.13** |
| 42_WBC | 99.17±0.95 | 99.17±1.20 | 98.23±2.07 | **98.28±1.70** |
| 43_WDBC | 99.26±0.41 | 99.26±0.64 | 99.75±0.34 | **99.85±0.22** |
| 44_Wilt | 47.33±3.93 | 64.37±3.31 | 59.51±8.49 | **87.18±1.12** |
| 45_wine | 86.30±6.23 | 95.56±4.87 | 96.30±3.98 | **99.26±0.77** |
| 46_WPBC | 56.49±5.82 | 58.17±9.08 | 59.50±9.83 | **61.35±8.51** |
| 47_yeast | 40.69±3.26 | 44.25±2.86 | 45.33±2.70 | **60.76±2.52** |
| **mean** | 79.51±2.84 | 86.13±2.12 | 85.79±2.71 | 89.01±2.05 |
| **mean rank** | 3.53 | 2.36 | 2.48 | 1.63 |
| **Friedman ($\chi^2$, $p$)** | | 53.29, $1.59 \times 10^{-11}$ | | |
| **Wilcoxon $p$ (Holm, DVM-AD>baseline)** | $1.72 \times 10^{-8}$ | $1.51 \times 10^{-3}$ | $1.58 \times 10^{-5}$ | – |

# E. Ablation Study

## E.1. Ablation on Discriminant-Vector Tail Selection

### E.1.1. DISCRIMINANT-VECTOR SELECTION STRATEGY

Algorithm 1 selects discriminant vectors (DVs) by solving the eigenproblem $MA = A\Lambda'$ and thresholding the eigenvalues $\{\lambda'_i\}_{i=1}^d$ (Step 5). In our formulation, $\lambda'_i \in [0, 1]$, where the minimal-tail eigenvalues ($\lambda'_i \approx 0$) correspond to highly compressive directions, while the maximal-tail eigenvalues ($\lambda'_i \approx 1$) correspond to more structure-preserving directions. Our default design (2-tail) selects DVs from both ends of the bounded spectrum to form a complementary projection. This design is motivated by the representation tradeoff: compressive directions tend to amplify point-wise deviations, whereas structure-preserving directions can be critical for cluster- and dependency-type anomalies.

To isolate the effect of DV tail selection, we compare the default 2-tailed mode with two single-tailed variants. All other components are kept identical, including the proxy point construction and the scoring/normalization procedure.

Let $\epsilon_{\text{sel}}$ be the (fixed, non-tuned) selection tolerance used in Algorithm 1. We define the DV index sets as follows:

**Min-tail only.** We keep only minimal-tail eigenvectors:

$$\mathcal{I}_{\min} = \{i \ : \ \lambda_i' < \epsilon_{\text{sel}}\}. \tag{11}$$

This variant emphasizes compactness of the normal class, which can benefit point-wise deviations, but it may under-represent broader structure that matters for cluster or dependency anomalies.

**Max-tail only.** We keep only maximal-tail eigenvectors:

$$\mathcal{I}_{\max} = \{i \ : \ \lambda_i' > 1 - \epsilon_{\text{sel}}\}. \tag{12}$$

This variant emphasizes preserving intrinsic structure, which can help for structure-related anomalies, but it may be less sensitive to subtle point-wise deviations that benefit from compactness.

**Two-tailed selection (default).** Our default configuration uses both tails:

$$\mathcal{I}_{2\text{tail}} = \mathcal{I}_{\min} \cup \mathcal{I}_{\max}. \tag{13}$$

Following Algorithm 1, if the selected set is empty, we fall back to using all directions.

### E.1.2. EXPERIMENT RESULTS

Table 10 reports the average AUROC/AUPRC (%) over the 47 real-world tabular datasets and the four synthetic anomaly settings (cluster/global/local/dependency). Overall, the default 2-tail mode consistently achieves the best performance across all groups and both metrics. Compared to Max-tail, 2-tail yields clear gains on real-world data (+4.806 AUROC and +7.536 AUPRC) and particularly large improvements on local (+4.754 AUROC, +10.213 AUPRC) and dependency anomalies (+4.954 AUROC, +12.898 AUPRC). On cluster and global settings, 2-tail also improves over Max-tail (+1.135 AUROC, +3.380 AUPRC for cluster; +1.458 AUROC, +4.890 AUPRC for global).

These results indicate that selecting from only one tail is insufficient on this benchmark: the two tails capture complementary information, and combining them provides a substantially more robust projection under diverse anomaly regimes.

We note that this study reports averaged results; in Section 5.5.2 we further analyze performance under different anomaly types, where the compactness and structure tradeoff becomes more apparent.

*Table 10.* Average results (% AUROC/AUPRC) of different DV tail-selection modes. Here, **Min-tail/Max-tail/2-tail** refer to DV selection (i.e., which eigen-directions are retained), not to the proxy anomaly construction. The last row reports differences between **2-tail** and **Max-tail**.

| Mode | Real-world | | Cluster | | Global | | Local | | Dependency | |
|---|---|---|---|---|---|---|---|---|---|---|
| | AUROC | AUPRC | AUROC | AUPRC | AUROC | AUPRC | AUROC | AUPRC | AUROC | AUPRC |
| **Min-tail** | 82.936 | 57.929 | 96.776 | 92.605 | 97.656 | 91.489 | 85.882 | 59.384 | 87.552 | 55.917 |
| **Max-tail** | 84.841 | 59.916 | 97.491 | 93.849 | 98.154 | 92.092 | 88.842 | 63.385 | 87.756 | 55.835 |
| **2-tail (default)** | 89.647 | 67.452 | 98.626 | 97.229 | 99.612 | 96.982 | 93.596 | 73.597 | 92.709 | 68.733 |
| $\Delta$ (**2-tail** $-$ **Max-tail**) | 4.8065 | 7.5358 | 1.1346 | 3.3800 | 1.4579 | 4.8900 | 4.7539 | 10.2126 | 4.9537 | 12.8980 |

**Qualitative visualization.** To provide an intuitive view of how discriminant-vector (DV) tail selection affects the learned representation, we visualize the projected space produced by DVM-AD under three DV selection modes: min-tail, max-tail, and the default 2-tail (both). For each mode, we first project samples using the corresponding DVM-AD projection and then apply t-SNE to obtain a 2D visualization. Normal instances are shown in blue and anomalies in orange.

As shown in Fig. 16, the min-tail projection (left) emphasizes highly compressive directions: the normal data tend to concentrate along a dominant low-dimensional structure (a clear curved manifold), while anomalies appear more dispersed

around it. In contrast, the max-tail projection (middle) preserves broader data geometry, resulting in a representation where the normal structure is less compact and anomalies overlap more noticeably with dense normal regions. The default 2-tail (both) mode (right) combines the two ends of the bounded spectrum, producing a representation that simultaneously maintains the intrinsic structure of the normal data while improving the contrast for anomalous samples (e.g., several anomaly groups become more peripheral/isolated relative to the main normal mass). Overall, this visualization supports the intuition that the two tails capture complementary information and that combining them yields a more robust embedding for diverse anomaly patterns.

### E.2. Alternative Single-Point Constructions of the Proxy Anomaly

E.2.1. SYNTHETIC ANOMALY GENERATION STRATEGY

Our default max-strategy constructs a single proxy anomaly point by taking the feature-wise maximum over the normal training set. To test whether performance is tied to this particular "max-corner" choice, we consider alternative constructions of the representative anomaly point $\tilde{\mathbf{x}}$, while keeping the rest of the training pipeline unchanged on 47 original real-world datasets and its four synthetic anomaly datasets.

Let $X_{\text{train}} = \{\mathbf{x}_i\}_{i=1}^n$ denote the normal training set with $\mathbf{x}_i \in \mathbb{R}^d$, and let $x_{i,j}$ be the $j$-th feature of $\mathbf{x}_i$.

**Feature-wise minimum (Min-corner).** As an "opposite corner" counterpart to the max construction, we define

$$\tilde{x}_j^{\min} = \min_{1 \le i \le n} x_{i,j}, \qquad \tilde{\mathbf{x}}^{\min} = (\tilde{x}_1^{\min}, \dots, \tilde{x}_d^{\min}) \in \mathbb{R}^d. \tag{14}$$

This choice probes whether the method benefits specifically from the *direction* of the proxy anomaly (max versus min), or more generally from placing a point at an axis-aligned "corner" of the normal data cloud.

**Farthest actual training sample from the normal mean.** Finally, we consider a data-driven, no data-specific tuning parameter construction that selects a real training example rather than synthesizing a coordinate-wise combination:

$$i^\star = \arg \max_{1 \le i \le n} \|\mathbf{x}_i - \bar{\mathbf{x}}\|_2, \qquad \tilde{\mathbf{x}}^{\text{far}} = \mathbf{x}_{i^\star}. \tag{15}$$

This strategy chooses the most "typical" normal instance under Euclidean distance to the normal mean. In contrast to the max/min corners-which may combine feature extremes that never co-occur in a single sample-$\tilde{\mathbf{x}}^{\text{far}}$ lies on the empirical data manifold by construction, while still being maximally distant from the average normal profile (especially meaningful after standardization).

**Augmented two-class training set.** For each construction above (max, min, or farthest), we form the same augmented two-class dataset as in the max strategy, $X_{\text{train}}^\star = X_{\text{train}} \cup \{\tilde{\mathbf{x}}\}$, assigning label $y^\star = 0$ to all normal samples and $y^\star(\tilde{\mathbf{x}}) = 1$ to the constructed proxy anomaly point.

E.2.2. WHY THE PROXY-POINT CONSTRUCTION HAS LIMITED IMPACT (THEORY)

Although we instantiate the proxy anomaly $\tilde{\mathbf{x}}$ via a deterministic rule (Max by default), Table 11 shows that alternative single-point constructions (Min/Max/Farthest) yield near-tied and non-monotonic performance. This empirical stability is expected from the geometry of our one-proxy discriminant formulation: the proxy is introduced to make one-class discriminant learning well-defined, but it enters the learning problem only through a rank-one update and is excluded from test-time scoring.

**Setup.** Let $X_{\text{train}} = \{\mathbf{x}_i\}_{i=1}^n \subset \mathbb{R}^d$ be the normal training set with mean $\bar{\mathbf{x}}$. We form the augmented two-class set $X_{\text{train}}^\star = X_{\text{train}} \cup \{\tilde{\mathbf{x}}\}$ by assigning label 0 to all normal samples and label 1 to the proxy sample. Let $S_w$ and $S_s$ denote the within-class and total scatter matrices computed on $(X_{\text{train}}^\star, y^\star)$ as in Algorithm 1.

**Proposition E.1** (Rank-one role of a single proxy point). *In the augmented two-class construction with one proxy point $\tilde{\mathbf{x}}$, the within-class scatter does not depend on $\tilde{\mathbf{x}}$ and the total scatter admits the decomposition*

$$S_s = S_w + S_b, \qquad S_b = \beta \, \boldsymbol{\delta}\boldsymbol{\delta}^\top, \qquad \beta = \frac{n}{n+1}, \quad \boldsymbol{\delta} = \bar{\mathbf{x}} - \tilde{\mathbf{x}}.$$

*Consequently, for any direction $\boldsymbol{\theta} \in \mathbb{R}^d$ the inverse-scatter ratio used by DVM-AD satisfies*

$$\kappa_{\tilde{\mathbf{x}}}(\boldsymbol{\theta}) \;=\; \frac{\boldsymbol{\theta}^\top S_w \boldsymbol{\theta}}{\boldsymbol{\theta}^\top S_s \boldsymbol{\theta}} \;=\; \frac{\boldsymbol{\theta}^\top S_w \boldsymbol{\theta}}{\boldsymbol{\theta}^\top S_w \boldsymbol{\theta} + \beta (\boldsymbol{\delta}^\top \boldsymbol{\theta})^2}.$$

*In particular: (i) if $\boldsymbol{\delta}^\top \boldsymbol{\theta} = 0$ and $\boldsymbol{\theta}^\top S_w \boldsymbol{\theta} > 0$, then $\kappa_{\tilde{\mathbf{x}}}(\boldsymbol{\theta}) = 1$; (ii) if $\boldsymbol{\theta} \in \mathrm{Null}(S_w)$ and $\boldsymbol{\delta}^\top \boldsymbol{\theta} \neq 0$, then $\kappa_{\tilde{\mathbf{x}}}(\boldsymbol{\theta}) = 0$.*

*Proof.* Since the proxy class contains a single sample, its within-class scatter is zero; hence $S_w$ is exactly the within-class scatter of the normal class and is independent of $\tilde{\mathbf{x}}$.

Let $\boldsymbol{\mu}^\star = \frac{n\bar{\mathbf{x}} + \tilde{\mathbf{x}}}{n+1}$ be the global mean of $X^\star_{\text{train}}$. The between-class scatter for two classes (normal vs. proxy) is

$$S_b = n(\bar{\mathbf{x}} - \boldsymbol{\mu}^\star)(\bar{\mathbf{x}} - \boldsymbol{\mu}^\star)^\top + (\tilde{\mathbf{x}} - \boldsymbol{\mu}^\star)(\tilde{\mathbf{x}} - \boldsymbol{\mu}^\star)^\top.$$

Using $\bar{\mathbf{x}} - \boldsymbol{\mu}^\star = \frac{\bar{\mathbf{x}} - \tilde{\mathbf{x}}}{n+1} = \frac{\boldsymbol{\delta}}{n+1}$ and $\tilde{\mathbf{x}} - \boldsymbol{\mu}^\star = -\frac{n}{n+1}\boldsymbol{\delta}$, we obtain

$$S_b = n\frac{\boldsymbol{\delta}\boldsymbol{\delta}^\top}{(n+1)^2} + \frac{n^2}{(n+1)^2}\boldsymbol{\delta}\boldsymbol{\delta}^\top = \frac{n}{n+1}\boldsymbol{\delta}\boldsymbol{\delta}^\top.$$

Finally, $S_s = S_w + S_b$ implies $\boldsymbol{\theta}^\top S_s \boldsymbol{\theta} = \boldsymbol{\theta}^\top S_w \boldsymbol{\theta} + \beta(\boldsymbol{\delta}^\top \boldsymbol{\theta})^2$, which yields the stated form of $\kappa_{\tilde{\mathbf{x}}}(\boldsymbol{\theta})$. The two extremal claims follow immediately by substituting $\boldsymbol{\delta}^\top \boldsymbol{\theta} = 0$ (giving $\kappa = 1$) or $\boldsymbol{\theta}^\top S_w \boldsymbol{\theta} = 0$ with $\boldsymbol{\delta}^\top \boldsymbol{\theta} \neq 0$ (giving $\kappa = 0$). $\qquad\square$

**Implication for proxy insensitivity.** Proposition E.1 shows that changing the proxy point only changes the single vector $\boldsymbol{\delta} = \bar{\mathbf{x}} - \tilde{\mathbf{x}}$ and affects $S_s$ through a *rank-one* update. Therefore, the set of tail directions targeted by DVM-AD is largely stable:

- The maximal-tail solutions ($\kappa \approx 1$) include all directions orthogonal to $\boldsymbol{\delta}$ (up to degeneracies), forming a high-dimensional subspace that exists for *any* proxy choice.

- The minimal-tail solutions ($\kappa \approx 0$) are primarily governed by $\mathrm{Null}(S_w)$ (when present), and depend on $\tilde{\mathbf{x}}$ only through the single constraint $\boldsymbol{\delta}^\top \boldsymbol{\theta} \neq 0$, which can exclude at most a low-dimensional slice.

Moreover, the proxy point is not used at test time; the anomaly score is computed solely from nearest-neighbor distances to projected normal training samples. Hence, the proxy mainly serves to make the one-class discriminant geometry well-defined, while its specific single-point construction is structurally constrained to have only a limited effect on the learned representation and detection performance, consistent with the near-tie behavior observed in Table 11.

E.2.3. EXPERIMENTAL RESULTS

Across three synthetic-anomaly selection strategies (Min, Max, and Farthest), the results in Table 11 show that performance differences are small and non-monotonic, with no single strategy dominating across real-world and anomaly-type settings. Min is marginally best on the real-world split (AUROC/AUPRC 90.02/68.04) and slightly higher on the cluster setting, while Farthest attains the top average scores on the global and dependency AUPRC (and global AUROC), and Max is competitive throughout, matching the best methods within a few tenths of a point and achieving the highest average AUROC on local and dependency anomalies. Given this near-tie behavior, we adopt the Max strategy as the default because it provides a stable, deterministic, and no data-specific tuning method for constructing a representative synthetic anomaly directly from observed data, thereby preserving the method's no data-specific tuning (up to fixed numerical tolerances) design while maintaining consistently strong performance across all anomaly categories. Importantly, the Max strategy never exhibits a catastrophic drop relative to alternatives, and its results are effectively on par with the best-performing choice in each column, making it a sensible default that avoids introducing extra selection complexity without sacrificing accuracy.

**E.3. Ablation on the Selection Tolerance $\epsilon_{\text{sel}}$**

Algorithm 1 selects discriminant vectors (DVs) by thresholding eigenvalues $\{\lambda'_i\}_{i=1}^d$ using the tolerance $\epsilon_{\text{sel}}$ (Step 5). Specifically, DVM-AD retains eigen-directions from the two tails of the bounded spectrum $\lambda'_i \in [0, 1]$: (i) *compressive* directions with $\lambda'_i < \epsilon_{\text{sel}}$, and (ii) more *structure-preserving* directions with $\lambda'_i > 1 - \epsilon_{\text{sel}}$ (see Eq. (13)). While $\epsilon_{\text{sel}}$ is intended to be a fixed, non-tuned numerical tolerance shared across datasets, it controls how aggressively tail directions are mined and may affect the resulting projection dimension and detection performance.

*Table 11.* The average results of different generated synthetic anomaly strategies.

| Strategy | Real-world | | Cluster | | Global | | Local | | Dependency | |
|---|---|---|---|---|---|---|---|---|---|---|
| | **AUROC** | **AUPRC** | **AUROC** | **AUPRC** | **AUROC** | **AUPRC** | **AUROC** | **AUPRC** | **AUROC** | **AUPRC** |
| **Min** | 90.02 | 68.04 | 98.65 | 97.28 | 99.90 | 97.83 | 93.52 | 73.64 | 92.66 | 69.15 |
| **Max** | 89.65 | 67.45 | 98.63 | 97.23 | 99.61 | 96.98 | 93.60 | 73.60 | 92.71 | 68.73 |
| **Farthest** | 89.44 | 66.35 | 98.32 | 97.22 | 99.92 | 98.05 | 93.23 | 73.87 | 92.51 | 69.17 |

**Experimental setup.** To assess sensitivity to $\epsilon_{\text{sel}}$, we run DVM-AD on the same 47 real-world tabular datasets under the same out-of-the-box protocol described in Section 5.4 (70/30 split with a fixed seed; normal-only training; identical proxy-point construction; identical scoring and normalization). The only change is the value of $\epsilon_{\text{sel}}$. We evaluate the averaged AUROC/AUPRC (%) over the 47 datasets for

$$\epsilon_{\text{sel}} \in \{10^{-1}, 10^{-2}, 10^{-3}, 10^{-4}, 10^{-5}\}. \tag{16}$$

**Results and discussion.** Table 12 reports the average AUROC/AUPRC (%) across the 47 datasets. Overall, performance is broadly stable across a wide range of tolerances, supporting the use of a single fixed $\epsilon_{\text{sel}}$ without dataset-specific tuning. We observe the best average results at $\epsilon_{\text{sel}} = 10^{-1}$ (AUROC $= 89.65\%$, AUPRC $= 67.45\%$). As $\epsilon_{\text{sel}}$ decreases, performance tends to degrade, consistent with the intuition that overly strict tail thresholds may select too few discriminant directions and reduce representational capacity. In this study, we fix $\epsilon_{\text{sel}=0.1}$ throughout all other experiments.

*Table 12.* Average results (% AUROC/AUPRC) on 47 real-world tabular datasets for different fixed selection tolerances $\epsilon_{\text{sel}}$.

| $\epsilon_{\text{sel}}$ | **AUROC(%)** | **AUPRC(%)** |
|---|---|---|
| $10^{-1}$ | 89.65 | 67.45 |
| $10^{-2}$ | 88.79 | 66.21 |
| $10^{-3}$ | 87.41 | 62.64 |
| $10^{-4}$ | 88.44 | 64.86 |
| $10^{-5}$ | 86.66 | 62.30 |

### E.4. Ablation Study on Contaminated Training Data

**Contamination Injection Protocol.** To evaluate the robustness of DVM-AD under contaminated supervision, we consider *annotation noise* in the training data, which commonly arises in real-world anomaly detection scenarios due to missing, ambiguous, or incorrect labels.

Let $(\mathbf{X}, \mathbf{y})$ denote the training set, where $\mathbf{X} \in \mathbb{R}^{n \times d}$ is the feature matrix and $\mathbf{y} \in \{0,1\}^n$ is the label vector, with 1 indicating anomalies and 0 indicating normal samples. For a given contamination level $\eta \in \{0.01, 0.02, 0.05\}$, we inject *asymmetric annotation noise* by randomly flipping a fraction $\eta$ of anomalous labels in the training data from anomaly to normal.

Formally, let $\mathcal{I}_1 = \{i \mid y_i = 1\}$ denote the index set of anomalous samples. A subset $\mathcal{I}_\eta \subset \mathcal{I}_1$ with cardinality $|\mathcal{I}_\eta| = \lfloor \eta |\mathcal{I}_1| \rfloor$ is selected uniformly at random, and the contaminated labels $\tilde{\mathbf{y}}$ are defined as

$$\tilde{y}_i = \begin{cases} 0, & \text{if } i \in \mathcal{I}_\eta, \\ y_i, & \text{otherwise.} \end{cases}$$

This asymmetric contamination protocol reflects a realistic setting in anomaly detection, where anomalous instances are more likely to be incorrectly labeled as normal rather than the reverse. Importantly, label contamination is applied *only to the training data*, while the test data remains clean, ensuring an unbiased evaluation of robustness to training-time contamination.

**Results and Discussion.** Table 13 reports the average detection performance under different levels of annotation noise. As the contamination level increases, a consistent degradation in performance is observed across both evaluation metrics.

At a low contamination level of $\eta = 0.01$, the mean AUCROC and AUCPR are 89.19 and 65.08, respectively. When the noise level increases to $\eta = 0.02$, the mean AUCROC decreases to 88.81, while the mean AUCPR drops to 62.95, indicating a more pronounced sensitivity in precision–recall performance. At the highest contamination level of $\eta = 0.05$, the mean AUCROC further declines to 87.11, accompanied by a substantial reduction in AUCPR to 55.59.

Overall, the results demonstrate that annotation noise has a measurable impact on detection performance, particularly on AUCPR, which is more sensitive to mislabeled anomalies. Nevertheless, DVM-AD maintains a relatively high AUCROC even under moderate levels of contamination, suggesting a degree of robustness to noisy supervision.

*Table 13.* Average detection performance across 47 tabular datasets under different contamination levels $\eta$, comparing DVM-AD against three strong baselines (KNN, IForest, LUNAR). The DVM-AD row matches Table 13 of the original submission; baseline rows are added in the camera-ready revision.

**(a) Mean AUROC (%)**

| Method | $\eta = 0.00$ | $\eta = 0.01$ | $\eta = 0.02$ | $\eta = 0.05$ |
|--------|------|------|------|------|
| **DVM-AD** | **89.65** | **89.19** | **88.81** | **87.11** |
| LUNAR | 85.63 | 85.71 | 86.01 | 84.42 |
| KNN | 84.45 | 84.41 | 84.33 | 84.04 |
| IForest | 80.92 | 80.23 | 79.91 | 80.30 |

**(b) Mean AUPRC (%)**

| Method | $\eta = 0.00$ | $\eta = 0.01$ | $\eta = 0.02$ | $\eta = 0.05$ |
|--------|------|------|------|------|
| **DVM-AD** | **67.45** | **65.08** | **62.95** | **55.59** |
| LUNAR | 60.14 | 57.30 | 57.95 | 53.30 |
| KNN | 56.72 | 56.56 | 56.08 | 54.66 |
| IForest | 42.09 | 41.92 | 41.92 | 42.26 |

### E.5. Ablation Study on Noisy Training Data

**Noise Injection Protocol.**  To evaluate the robustness of DVM-AD under data perturbations, we inject controlled Gaussian noise into the training data. Specifically, given a training feature matrix $\mathbf{X} \in \mathbb{R}^{n \times d}$, noise is added feature-wise and proportionally to the empirical standard deviation of each feature.

Let $\sigma_j$ denote the standard deviation of the $j$-th feature computed over the training set. For a given noise level $\eta \in \{0.01, 0.02, 0.05\}$, the noisy training data $\tilde{\mathbf{X}}$ is constructed as:

$$\tilde{\mathbf{X}}_{ij} = \mathbf{X}_{ij} + \epsilon_{ij}, \quad \epsilon_{ij} \sim \mathcal{N}(0, \eta^2 \sigma_j^2),$$

where the noise is independently sampled for each sample and each feature. To avoid degenerate cases, features with zero variance are assigned $\sigma_j = 1$.

Importantly, noise is injected only into the training data, while the test data remains clean. This protocol isolates the effect of noise on model learning and prevents information leakage from corrupted test samples. All noise realizations are generated using a fixed random seed to ensure reproducibility.

**Results and Discussion.**  Table 14 reports the average detection performance across all datasets at different noise levels. In the noise-free setting ($\eta = 0.00$), the mean AUCROC and AUCPR are 89.65 and 67.45, respectively. When the noise level increases to $\eta = 0.01$ and $\eta = 0.02$, the corresponding mean AUCROC values are 89.55 and 89.52, while the mean AUCPR values are 66.54 and 66.41. At $\eta = 0.05$, the mean AUCROC is 89.31, representing a relative decrease of approximately 0.34% compared to the noise-free case. These results indicate that the average performance metrics remain stable as the noise level increases. However, we suggest to make futher experiment with high noise level and with different types of noise in future work.

*Table 14.* Average detection performance across 47 tabular datasets under different feature-noise levels $\eta$, comparing DVM-AD against three strong baselines (KNN, IForest, LUNAR). The DVM-AD row matches Table 14 of the original submission; baseline rows are added in the camera-ready revision.

**(a) Mean AUROC (%)**

| Method | $\eta = 0.00$ | $\eta = 0.01$ | $\eta = 0.02$ | $\eta = 0.05$ |
|---|---|---|---|---|
| **DVM-AD** | **89.65** | **89.55** | **89.52** | **89.31** |
| LUNAR | 85.63 | 85.95 | 85.95 | 85.89 |
| KNN | 84.45 | 84.49 | 84.47 | 84.44 |
| IForest | 80.92 | 79.05 | 80.08 | 80.19 |

**(b) Mean AUPRC (%)**

| Method | $\eta = 0.00$ | $\eta = 0.01$ | $\eta = 0.02$ | $\eta = 0.05$ |
|---|---|---|---|---|
| **DVM-AD** | **67.45** | **66.54** | **66.41** | **65.64** |
| LUNAR | 60.14 | 58.25 | 58.61 | 61.25 |
| KNN | 56.72 | 56.42 | 56.32 | 56.27 |
| IForest | 42.09 | 41.13 | 42.69 | 42.37 |

## F. Extended Experiments

### F.1. Full Benchmark Comparison Across 38 Methods

Table 15 reports the full per-method averages on the 47 ADBench tabular datasets under the 7:3 protocol. The pool consists of 38 methods in three groups: DVM-AD (**Proposed**); 28 **Original** baselines from the main results pool (Section 5.5.1, also visible in the critical-difference diagram of Figure 2); and 9 **Extended** baselines newly run in response to reviewer requests. Within each group, methods are sorted by Avg Rank AUROC (lower is better); ranks are computed across the full 38-method pool, so they are directly comparable across groups. DRL-AD (Ye et al., 2025) is part of the Original group; it was included in the main baseline pool before the rebuttal round. Note: minor discrepancies (typically $< 1\%$) may appear between baseline values in this table and the corresponding rows of Tables 13-14 or the main results in Section 5.5.1, as the experiments were run with different random seeds.

DVM-AD attains rank 3.53 (AUROC) and 4.15 (AUPRC) overall, leading the strongest baseline in each group: LUNAR (Original, rank 8.12) by a margin of 4.59, and TCCM (Extended, rank 11.62) by a margin of 8.09. The advantage extends across both the established main-pool baselines and the most recent generative and reconstruction-based tabular AD methods. Per-method positioning along anomaly-type coverage, hyperparameter requirements, and data modality is summarized in Appendix Table 4.

### F.2. Alternative Evaluation Protocol (50/50 Normal-Only Split)

The main experiments use a stratified 7:3 train/test split following ADBench (Han et al., 2022). Several recent works (Livernoche et al., 2024; Yin et al., 2024; Durani et al., 2025; Zong et al., 2018; Zhang et al., 2024) instead use a 50/50 protocol where 50% of normal samples form the training set and the remaining 50% of normals are combined with all anomalies for the test set. We verify DVM-AD's leading performance under this alternative protocol by re-running the experiment on all 47 ADBench tabular datasets against 31 baselines: 27 of the 28 original PyOD methods (PMKFN was not re-run under the 50/50 protocol due to computational constraints) and 4 reviewer-requested extended baselines (MCM (Yin et al., 2024), ContrastiveAD (ICL) (Shenkar & Wolf, 2022), ADERH (Durani et al., 2025), DTE (Livernoche et al., 2024)).

Table 16 reports the full ranking. DVM-AD achieves average rank 3.30 and average AUROC 88.81%, leading the second-best method (LUNAR, rank 6.66) by a margin of 3.36. The drop relative to the 7:3 protocol ($-0.84$ AUROC) is consistent with using less normal data for training, and DVM-AD's rank-1 position is preserved.

**Scope note.** Of the 10 reviewer-touched extended baselines compared under the 7:3 protocol (Appendix F.1; 9 newly run plus DRL-AD, which was already in the main pool), only 4 (MCM, ContrastiveAD (ICL), ADERH, DTE) were re-run under the 50/50 protocol; the remaining 6 (NPT-AD, RAAD, PTAD, GANomaly, TCCM, DRL-AD) appear only under 7:3 in the main results. PMKFN, a baseline in the main 28-method pool, was likewise not re-run under 50/50. We caution that direct

*Table 15.* Full benchmark comparison on 47 ADBench tabular datasets, 7:3 protocol. Average AUROC, AUPRC, and ranks (lower is better) across all 38 methods. DVM-AD is in **bold**. Within each group, methods are sorted by Avg Rank AUROC.

| Group | Model | Avg AUROC (%) | Avg AUPRC (%) | Avg Rank AUROC | Avg Rank AUPRC |
|---|---|---|---|---|---|
| Proposed | **DVM-AD (ours)** | **89.65** | **67.45** | **3.53** | **4.15** |
| Original | LUNAR | 85.10 | 59.13 | 8.12 | 8.65 |
| Original | KNN | 83.91 | 55.74 | 9.38 | 10.10 |
| Original | AutoEncoder | 83.05 | 50.80 | 10.19 | 11.88 |
| Original | LOF | 83.13 | 54.73 | 10.96 | 11.58 |
| Original | VAE | 80.00 | 46.92 | 13.69 | 13.69 |
| Original | SUOD | 80.74 | 43.38 | 13.75 | 15.46 |
| Original | NeuTraLAD | 79.47 | 48.39 | 13.85 | 14.56 |
| Original | IForest | 80.38 | 41.38 | 14.06 | 15.85 |
| Original | DRL-AD | 81.64 | 50.73 | 14.36 | 14.38 |
| Original | AE1SVM | 78.56 | 47.17 | 14.48 | 14.15 |
| Original | PMKFN | 80.45 | 54.57 | 14.72 | 12.51 |
| Original | CBLOF | 79.58 | 43.15 | 14.75 | 16.25 |
| Original | PCA | 76.97 | 43.48 | 16.10 | 16.35 |
| Original | ECOD | 76.69 | 41.51 | 17.10 | 18.08 |
| Original | DIF | 77.69 | 41.32 | 17.33 | 17.73 |
| Original | COPOD | 76.58 | 39.04 | 17.65 | 18.77 |
| Original | DeepSVDD | 75.21 | 39.86 | 18.35 | 17.21 |
| Original | HBOS | 76.69 | 37.73 | 18.42 | 18.69 |
| Original | LODA | 69.93 | 32.31 | 19.94 | 21.35 |
| Original | KNFST | 66.33 | 34.47 | 20.57 | 20.36 |
| Original | OCSVM | 68.26 | 29.82 | 20.58 | 21.83 |
| Original | SOD | 69.30 | 28.53 | 21.96 | 23.50 |
| Original | COF | 65.48 | 27.39 | 22.62 | 23.23 |
| Original | AnoGAN | 67.69 | 29.98 | 23.65 | 25.31 |
| Original | DevNet | 50.02 | 20.89 | 24.42 | 24.52 |
| Original | DROCC | 49.19 | 21.68 | 24.92 | 23.81 |
| Original | ALAD | 44.79 | 13.49 | 28.71 | 29.98 |
| Original | SO_GAAL | 43.13 | 13.55 | 28.94 | 29.75 |
| Extended | TCCM | 81.76 | 55.32 | 11.62 | 11.72 |
| Extended | MCM | 82.38 | 52.40 | 11.79 | 13.83 |
| Extended | ContrastiveAD (ICL) | 82.41 | 55.08 | 12.00 | 11.74 |
| Extended | PTAD | 80.56 | 42.46 | 14.28 | 15.79 |
| Extended | ADERH | 79.73 | 48.52 | 14.85 | 15.28 |
| Extended | GANomaly | 78.95 | 49.56 | 15.45 | 14.23 |
| Extended | RAAD | 79.31 | 46.81 | 16.26 | 15.23 |
| Extended | NPT-AD | 78.10 | 45.04 | 18.23 | 17.96 |
| Extended | DTE | 67.78 | 33.82 | 20.62 | 20.55 |

7:3-vs-50/50 numerical comparisons should focus on methods evaluated under both protocols.

## F.3. Distribution of Selected Discriminant Directions

This subsection documents how often each tail of the bounded spectrum contributes selected discriminant vectors (DVs) across the 47 ADBench tabular datasets, under our default tolerance $\epsilon_{\text{sel}} = 10^{-1}$. For each dataset we record (i) whether the fallback rule fired (i.e. no eigenvalues satisfy $\lambda'_i < \epsilon_{\text{sel}}$ or $\lambda'_i > 1 - \epsilon_{\text{sel}}$, in which case all $d$ DVs are kept; (ii) whether at least one max-tail DV was selected; (iii) whether at least one min-tail DV was selected; and (iv) the ratio $m/d$ of selected DVs to the input dimension. Table 17 reports the aggregate statistics.

Three observations follow. First, the fallback rule never triggers on this benchmark, so the bounded-spectrum selection always

*Table 16.* 50/50 protocol, all methods combined ($n_{\text{methods}}=32$). Average across 47 ADBench tabular datasets; ranks computed within each dataset (lower is better).

| Method | AUROC (%) | AUPRC (%) | Rank (AUROC) | Rank (AUPRC) |
|---|---|---|---|---|
| **DVM-AD** | **88.81** | **71.87** | **3.30** | **3.23** |
| LUNAR | 86.11 | 65.64 | 6.66 | 6.74 |
| KNN | 83.85 | 63.06 | 8.13 | 8.49 |
| MCM | 81.00 | 57.94 | 10.43 | 11.45 |
| AutoEncoder | 82.37 | 57.62 | 10.64 | 11.15 |
| DRL-AD | 82.52 | 58.75 | 11.02 | 11.00 |
| LOF | 81.52 | 61.81 | 11.11 | 11.19 |
| ContrastiveAD (ICL) | 81.55 | 60.07 | 11.63 | 10.22 |
| VAE | 80.68 | 55.25 | 11.96 | 12.96 |
| SUOD | 80.47 | 52.51 | 12.30 | 13.81 |
| CBLOF | 79.51 | 52.21 | 12.45 | 13.15 |
| IForest | 79.89 | 49.68 | 12.68 | 14.30 |
| NeuTraLAD | 77.62 | 52.49 | 13.09 | 13.02 |
| AE1SVM | 79.67 | 53.41 | 13.20 | 13.35 |
| ADERH | 79.11 | 55.39 | 13.30 | 12.66 |
| PCA | 77.06 | 50.65 | 14.91 | 15.79 |
| HBOS | 77.03 | 47.37 | 15.26 | 15.66 |
| DIF | 76.34 | 48.23 | 15.36 | 15.34 |
| COPOD | 74.46 | 43.86 | 17.23 | 18.34 |
| DeepSVDD | 74.74 | 47.98 | 17.51 | 17.13 |
| DTE | 68.15 | 41.11 | 17.83 | 17.26 |
| ECOD | 74.03 | 44.86 | 18.19 | 18.23 |
| KNFST | 64.72 | 41.77 | 18.77 | 18.28 |
| OCSVM | 68.26 | 37.64 | 19.26 | 19.70 |
| LODA | 67.03 | 38.75 | 19.66 | 19.77 |
| AnoGAN | 69.58 | 39.62 | 20.11 | 20.81 |
| SOD | 63.70 | 33.20 | 20.60 | 21.30 |
| COF | 60.99 | 32.32 | 21.98 | 22.06 |
| DROCC | 49.19 | 29.39 | 22.81 | 21.53 |
| DevNet | 48.28 | 26.59 | 22.89 | 22.57 |
| SO_GAAL | 46.63 | 23.41 | 26.21 | 26.53 |
| ALAD | 44.94 | 19.38 | 27.09 | 27.47 |

*Table 17.* Distribution of selected discriminant directions across 47 ADBench tabular datasets at default tolerance $\epsilon_{\text{sel}} = 10^{-1}$.

| Quantity | Value |
|---|---|
| Datasets triggering the all-DV fallback | 0/47 |
| Datasets selecting $\geq 1$ max-tail DV | 47/47 |
| Datasets selecting $\geq 1$ min-tail DV | 7/47 |
| Median ratio of selected DVs to feature dimension ($m/d$) | 0.926 |

returns a non-empty set under our fixed tolerance. Second, the max-tail directions are universally selected ($47/47$): every dataset contains directions where the proxy-anchored separation effectively vanishes, supporting the structure-preserving role discussed in Section 4. Third, the min-tail (compressive, NPD-like) directions are also present on a non-trivial fraction of datasets ($7/47$), consistent with the two-tail mechanism analyzed in Appendix E.1: when normal data admit truly compressible directions, DVM-AD picks them up; when they do not, the bounded geometry pushes mass toward the max-tail end. The median $m/d = 0.926$ indicates that most of the spectrum is informative under the default tolerance; tightening $\epsilon_{\text{sel}}$ shrinks $m/d$, as analyzed in Appendix E.3.

## F.4. PCA Pre-Reduction at Extreme Feature Dimension

Section 4.5 notes that the dominant training cost in $d$ for DVM-AD is the $O(d^3)$ eigendecomposition step, which becomes the bottleneck only at extreme feature dimension ($d \gtrsim 10^3$). For such regimes we provide a simple drop-in option: project the input to a lower-dimensional subspace via PCA before running DVM-AD. The projection is fitted on normal training data only, and the projection axes are fixed across train and test. We report the trade-off between speed and detection performance when reducing to $\lfloor \rho \cdot d \rfloor$ components for $\rho \in \{0.50, 0.25, 0.10\}$, averaged across the 47 ADBench tabular datasets.

*Table 18.* PCA pre-reduction trade-off, averaged over 47 ADBench tabular datasets. $\Delta$AUROC is the (signed) change relative to the no-pre-reduction baseline (Section 5.5.1, AUROC $= 89.65\%$). Training-time speedup is the ratio of full-$d$ wall-clock time to the pre-reduced wall-clock time.

| PCA ratio $\rho$ | $\Delta$AUROC (%) | Training-time speedup |
|---|---|---|
| 0.50 | $-0.40$ | $2.5\times$ |
| 0.25 | $-0.61$ | $6.3\times$ |
| 0.10 | $-1.38$ | $15.5\times$ |

Detection performance degrades gracefully with aggressive reduction. Retaining half the principal components ($\rho = 0.50$) gives a $2.5\times$ speedup at a $0.40\%$ AUROC cost; retaining 10% ($\rho = 0.10$) still keeps the loss within $1.38\%$ while delivering a $15.5\times$ speedup. The result confirms two design properties of DVM-AD: (i) the scatter geometry remains informative even after substantial linear compression, and (ii) the practical bottleneck at extreme $d$ can be alleviated without sacrificing the closed-form, tuning-free recipe. We recommend this as an opt-in for $d \gtrsim 10^3$ regimes; in the moderate-$d$ regime ($d < 200$, which covers most ADBench datasets) the speedup is marginal and the small detection cost is not justified.

# G. Additional Visualizations

This section provides intuition for when DVM-AD succeeds and where it struggles, beyond the aggregated metrics in Section 5.5.1. We present two complementary views: (i) t-SNE embeddings of the DVM-AD discriminant subspace on a success case and a failure case, and (ii) histograms of nearest-neighbour distances in the discriminant space on two representative datasets, showing how DVM-AD's bounded score $p(x) \in [0, 1]$ separates normal and anomalous samples.

## G.1. t-SNE on a Success Case (3_backdoor) and a Failure Case (39_vertebral)

Figure 17 contrasts a high-AUROC success (3_backdoor, AUROC $= 98.48\%$) with a low-AUROC failure (39_vertebral, AUROC $= 57.39\%$). On 3_backdoor, the DVM-AD subspace places the normal cluster compactly while anomalies (orange) sit on the periphery, consistent with the two-tail mechanism. On 39_vertebral, anomalies and normals are heavily interleaved in the projected space, indicating that the bounded inverse-scatter spectrum cannot find directions that separate them with the available training signal. This dataset is small (240 samples, 6 features) with a 12.5% anomaly proportion (Table 5), which limits the resolution of any one-class projection learner.

## G.2. Nearest-Neighbour Distance Histograms (10_cover and 30_satellite)

Figure 18 shows the distribution of DVM-AD anomaly scores $p(x)$ (Eq. 9) on two datasets where the bounded normalization makes the threshold-readiness of DVM-AD visible: 10_cover and 30_satellite. Normal samples concentrate near $p(x) \approx 0$ on both datasets, while anomalies are pushed toward larger values, producing the wide separation that drives high AUROC. The bounded range $[0, 1]$ stabilizes thresholding in practice: a single deployment threshold $t$ in the high-density gap (e.g. $t \approx 0.5$ for both datasets shown) recovers near-optimal precision-recall tradeoffs.

# H. Limitations and Broader Impacts

DVM-AD is designed as a robust, closed-form one-class detector under a fixed and simple configuration. Nevertheless, some limitations arise either from (i) general constraints shared by anomaly detection methods or (ii) practical/experimental choices made by experimental constraints. Importantly, these limitations should be viewed as opportunities for further research, especially for extending DVM-AD to broader data types and for strengthening deployment practices in high-stakes settings.

- **Linear projection**: DVM-AD learns a linear discriminant projection, which provides a closed-form solution with stability and interpretability benefits. A linear projection, however, may miss higher-order nonlinear interactions in the raw feature space for some problems. That said, our evaluation spans 47 real-world tabular datasets and 10 NLP/CV embedding datasets, in which the upstream representations can contain substantial nonlinear structure. Across these settings, including data that is not strictly linear, DVM-AD still achieves top-tier detection performance, suggesting that a linear detector can remain highly competitive in practice. We nonetheless agree that deployment on inherently nonlinear data should be explored more carefully in future work, e.g., via principled nonlinear feature learning, kernelized or piecewise-linear variants, or tighter guidance on when linear projections are sufficient. A complementary direction is extending the proxy-anchored construction to multi-class Fisher discriminant analysis when sub-class labels for the normal data are available, which would let DVM-AD exploit known multimodal structure rather than relying on the two-tailed mechanism alone.

- **Anomaly threshold**: Threshold selection remains application-dependent and is a general limitation of anomaly detection models whenever discrete decisions are required. DVM-AD outputs a bounded anomaly score $p(x) \in [0, 1]$, which can simplify score calibration and comparison across datasets. However, producing binary decisions still requires choosing a deployment threshold $t$ (Algorithm 2), and the appropriate value depends on operational constraints (e.g., tolerance for false positives/negatives) and the relative costs of errors. This issue is not unique to DVM-AD: many anomaly detectors require thresholding to produce label outputs. In our experiments, we therefore report threshold-free metrics (AUROC/AUPRC), so no threshold tuning is needed for evaluation.

- **Scaling in very high dimension**: Projection learning in DVM-AD involves scatter computation and a spectral decomposition step, which can be computationally heavy when $d$ is extremely large (e.g., $d = 10^4$, where full eigen-decomposition scales as $O(d^3)$). In practice, such extremely high-dimensional feature vectors are relatively rare in real-world tabular scenarios; moreover, even in that case, DVM-AD can still operate (it remains well-posed under rank deficiency via the pseudo-inverse formulation), and inference remains efficient after training (projection + ANN query). Still, we agree that further work can improve this limitation, for example through randomized/iterative eigensolvers, low-rank approximations, feature screening, or structured projections that reduce the training-time spectral cost.

- **Potential negative impacts in high-stakes deployments**: As with any anomaly detector, false positives can trigger unnecessary interventions (e.g., incorrectly flagging legitimate users/transactions), while false negatives can miss genuinely harmful events. These risks are amplified in high-stakes settings (e.g., fraud screening, security monitoring, healthcare triage) where the cost of errors is uneven and may affect different groups unequally. We therefore recommend: (i) validating thresholds and downstream actions with domain experts, (ii) monitoring error rates by subgroup when demographic attributes are available and appropriate to use, and (iii) using DVM-AD outputs as decision support with human oversight rather than as the sole automated decision-maker in high-stakes deployments.

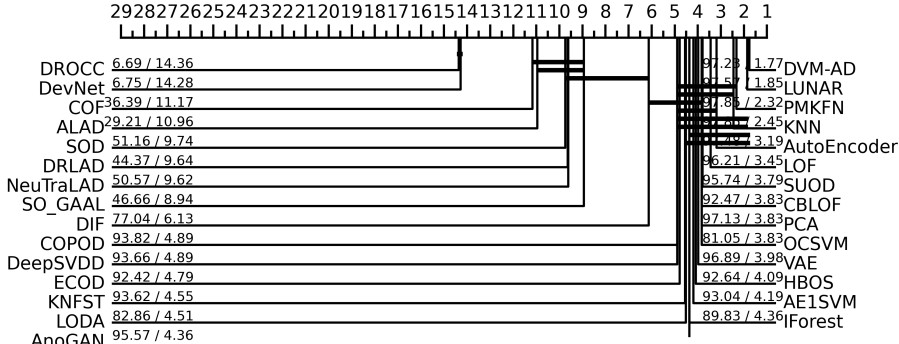

*(a)* Cluster anomaly.

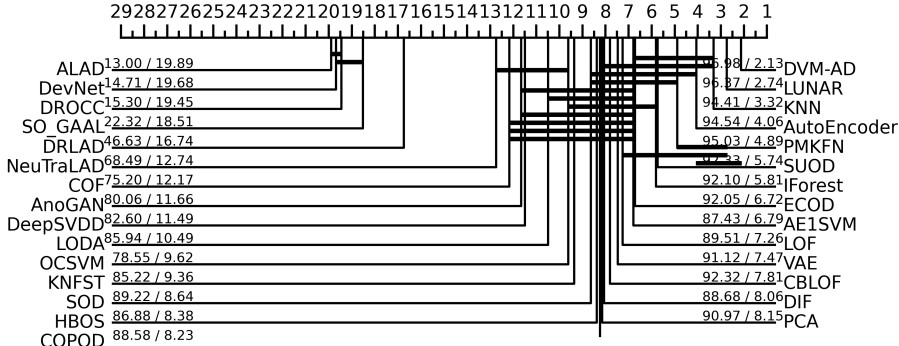

*(b)* Global anomaly.

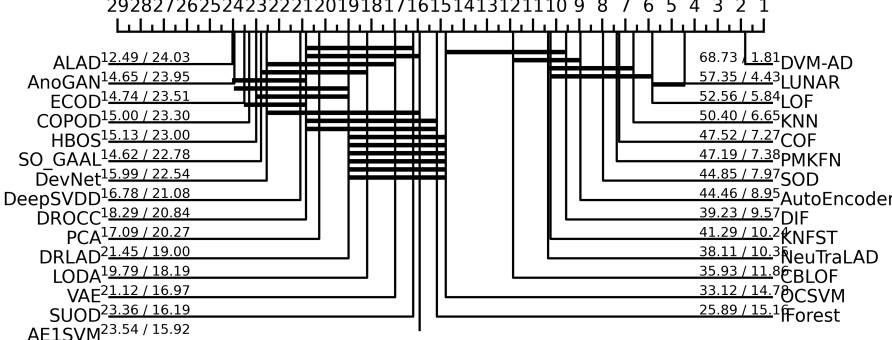

*(c)* Dependency anomaly.

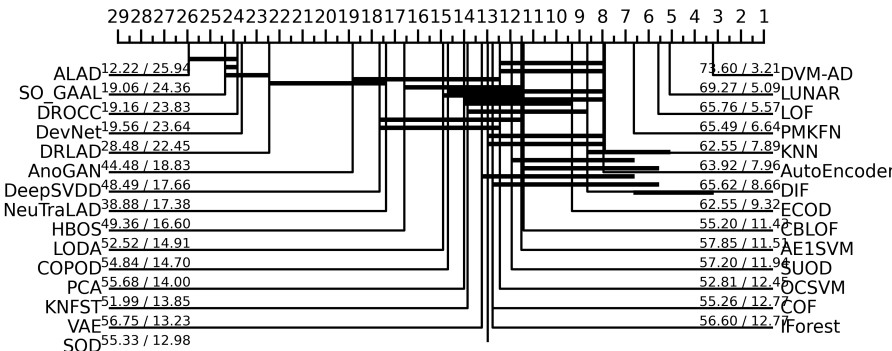

*(d)* Local anomaly.

*Figure 13.* The average AUPRC value and average rank of algorithms on detecting different anomaly types. The results consistently show that DVM-AD outperforms other algorithms.

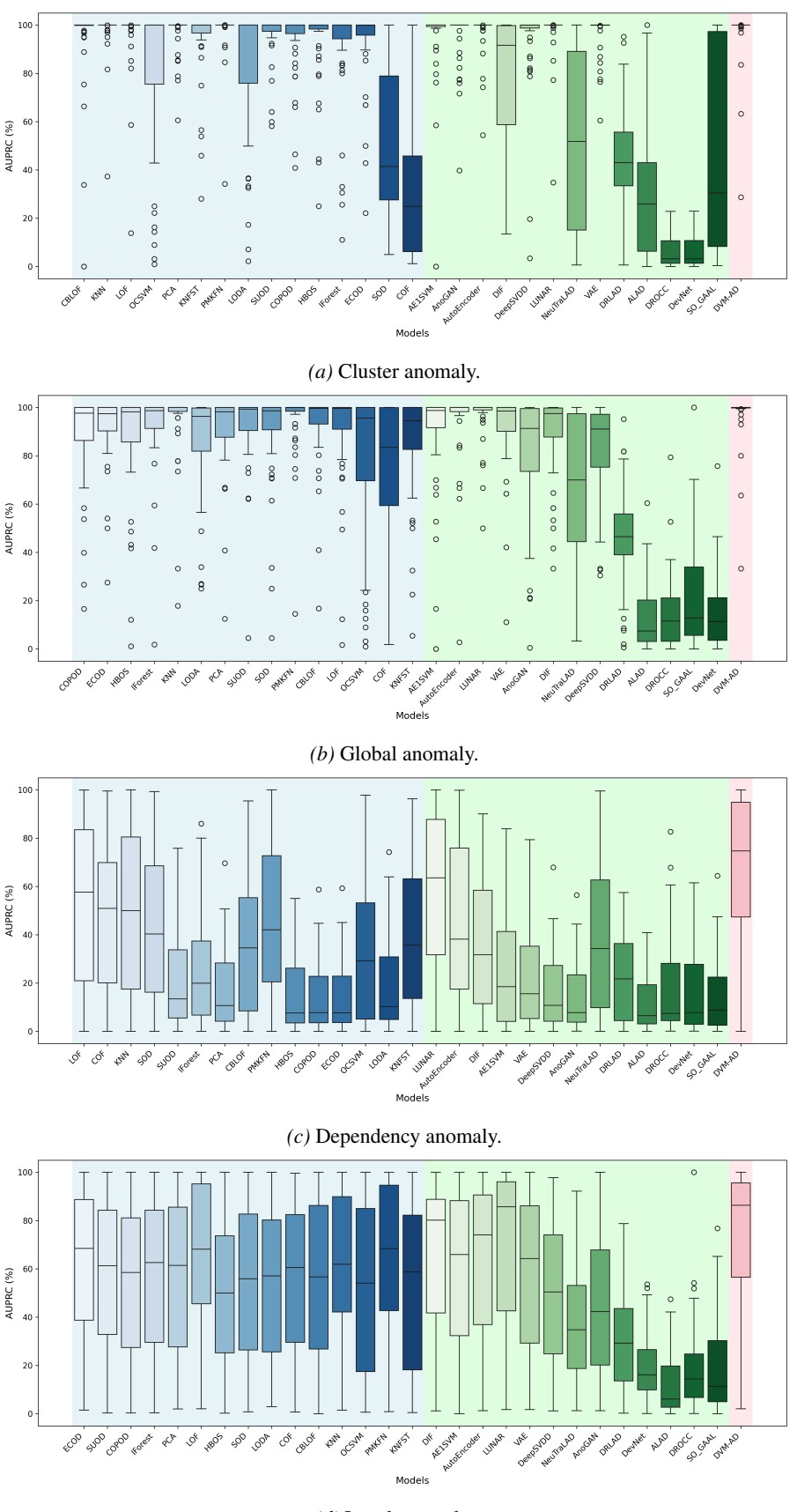

*(a)* Cluster anomaly.

*(b)* Global anomaly.

*(c)* Dependency anomaly.

*(d)* Local anomaly.

*Figure 14.* The distribution of AUROC values on detecting different anomaly types. The results consistently show that DVM-AD outperforms other algorithms.

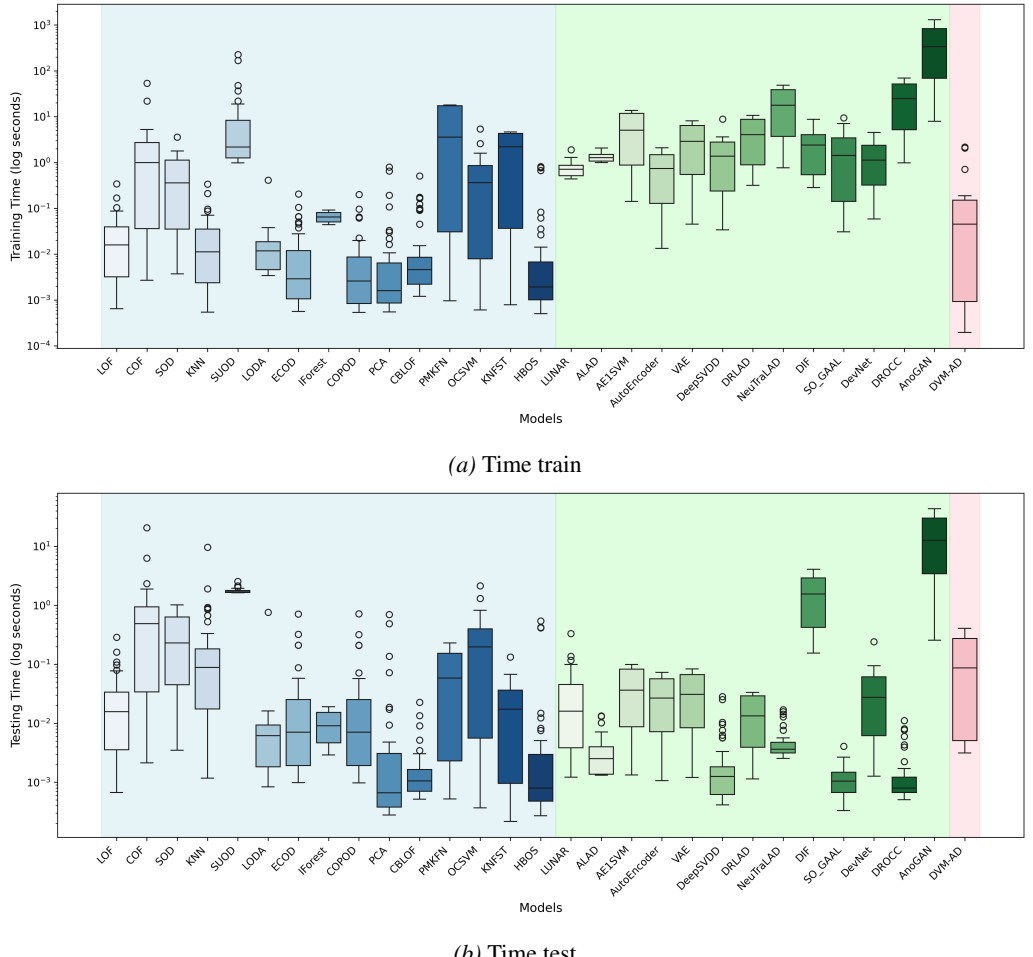

*(a)* Time train

*(b)* Time test

*Figure 15.* The summary of runtime performance (in seconds) of the train/test process of algorithms. The lower value is better. The results show that DVM-AD achieves the lowest median training time (together with HBOS and COPOD). However, in terms of testing time, while DVM-AD is not the fastest algorithm, it still achieves a comparable testing time.

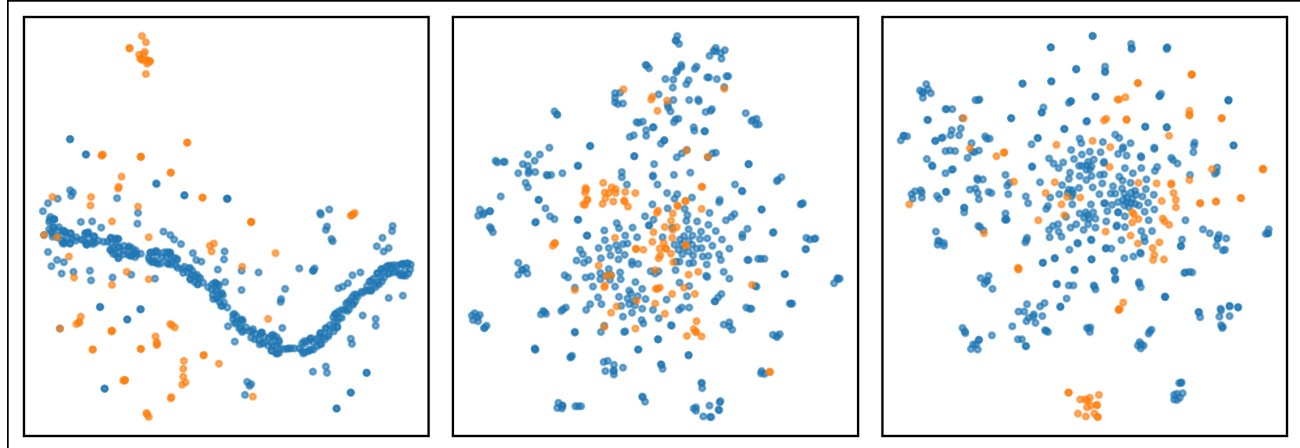

*Figure 16.* **t-SNE visualization of DVM-AD representations under different DV tail-selection modes on 17_InternetAds.** From left to right: min-tail, max-tail, and 2-tail (both). Samples are first projected by the corresponding DVM-AD discriminant subspace and then embedded into 2D by t-SNE. Normal instances are shown in blue and anomalies in orange.

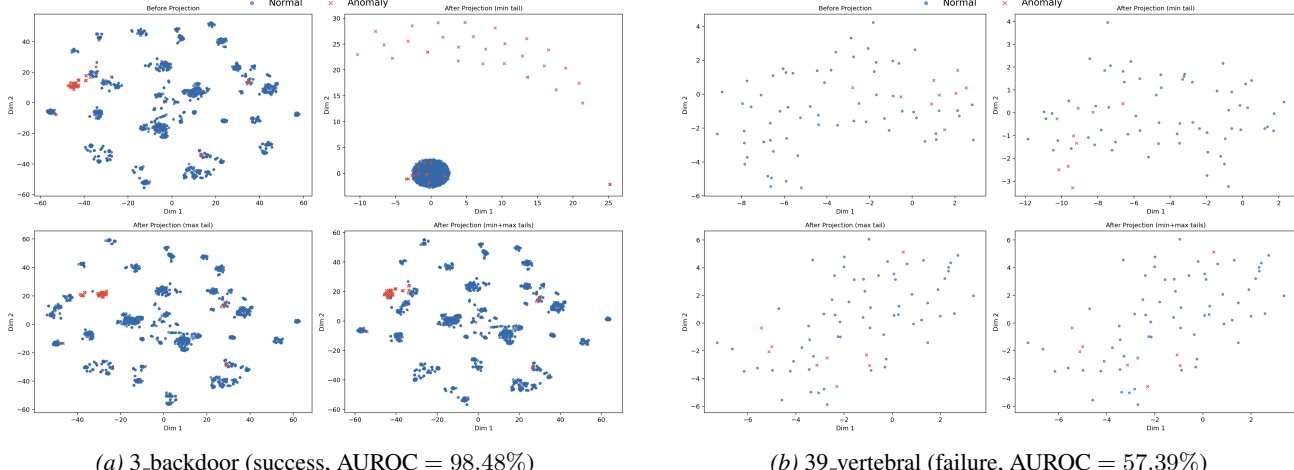

*(a)* 3_backdoor (success, AUROC = 98.48%)        *(b)* 39_vertebral (failure, AUROC = 57.39%)

*Figure 17.* t-SNE visualization of DVM-AD discriminant-space representations on (a) a success case and (b) a failure case from the 47 ADBench tabular datasets. Normal instances are shown in blue and anomalies in orange. The success case shows compact normals with peripheral anomalies; the failure case shows substantial overlap between the two classes, reflecting limited separability under the available training signal.

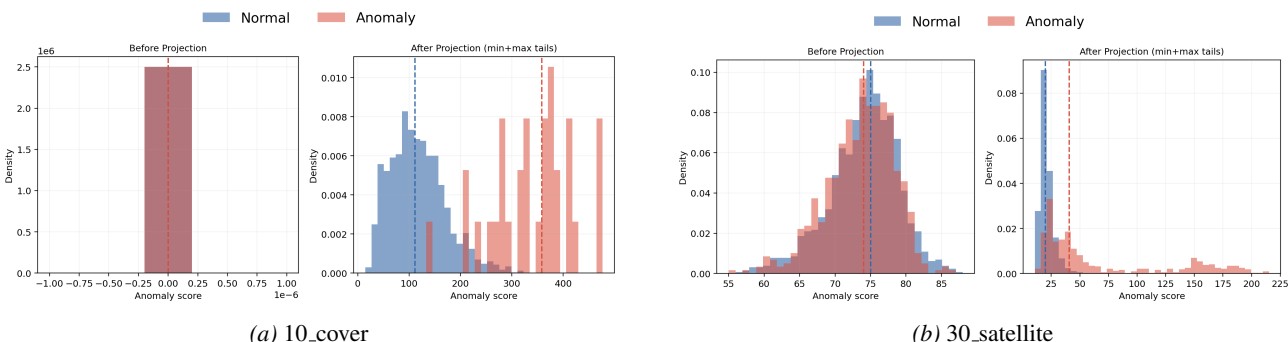

*(a)* 10_cover            *(b)* 30_satellite

*Figure 18.* Histograms of DVM-AD normalized anomaly scores $p(x) \in [0, 1]$ on (a) 10_cover and (b) 30_satellite. Normal samples (blue) concentrate near zero; anomalies (orange) are pushed toward the upper end of the range. The bounded $[0, 1]$ scale simplifies deployment-time threshold selection.

