# OpenReview forum: "Tuning-Free One-Class Discriminant Learning for Tabular Anomaly Detection"
_ICML.cc/2026/Conference — ICML 2026 regular_

### Official Review · Reviewer_VZkh · 2026-03-06

**Soundness:** 2
**Presentation:** 3
**Significance:** 2
**Originality:** 3
**Overall Recommendation:** 4
**Confidence:** 4

**Summary:**

This paper proposes a novel traditional anomaly detection method called DVM-AD. The method constructs a single discriminative space from the training data that is both compact and capable of preserving the structural properties of the data. The proposed approach is evaluated on 57 datasets with comparisons against 28 baseline methods. The experimental results demonstrate the effectiveness of the proposed approach.

**Compliance With Llm Reviewing Policy:**

Affirmed.

**Final Justification:**

Although the proposed method is inefficient on high-dimensional data, which is a shortcoming of most method that does not apply
neural networks, and the original experimental setting of dropping anomalies is somewhat confusing, I decide to raise my score since the proposed method is effective in 50/50 training/testing setting.

**Key Questions For Authors:**

Please refer to the concerns raised in the Weaknesses section. I would be willing to increase my score if these issues are properly addressed.

**Limitations:**

Yes

**Strengths And Weaknesses:**

**Strengths**
1. The paper is generally well-written and easy to follow.
2. The experimental evaluation is extensive. Experiments on both real-world and synthetic datasets demonstrate the effectiveness of the proposed method.
3. The paper includes theoretical analysis and time complexity analysis, which helps justify the design of the proposed method.


**Weaknesses**
1. The proposed method appears to be mainly designed for a one-class classification scenario. However, in real-world datasets, normal data may consist of multiple classes or sub-distributions. It would be helpful if the authors could discuss whether the proposed method can handle such more realistic settings.
2. A potential drawback of DVM-AD is that its training time may become large for high-dimensional data. Although the authors report the training time of DVM-AD, it is not compared with the training time of other baseline methods. Including such comparisons in Table 2 would provide a clearer understanding of the computational efficiency of the proposed method.
3. The experimental protocol splits datasets into 7:3 train/test subsets, which differs from the commonly used anomaly detection setting in [1]-[4] where half of the normal data is used for training and the remaining normal data is combined with anomalies to form the testing set. The authors should clarify the motivation for adopting this different experimental setting.
4. Although the paper compares DVM-AD with many baselines, most of these methods were proposed before 2024. Including more recent anomaly detection methods published in top venues such as ICML, ICLR, and NeurIPS (e.g., [1]–[4]) would make the empirical evaluation more up-to-date and convincing.
5. For other important settings, such as anomaly detection with contaminated training data or data with noise, no baseline methods are included for comparison. Adding such baselines would strengthen the experimental evaluation.
6. The formulation and motivation of $\tilde{x}$ are somewhat difficult to understand. The current explanation appears to assume that larger feature values are more likely to correspond to anomalies. However, in many real-world datasets, larger feature values do not necessarily indicate anomalous behavior. A clearer explanation and additional experimental results would improve the clarity of this component.

[1] Livernoche V, Jain V, Hezaveh Y, et al. On diffusion modeling for anomaly detection[J]. arXiv preprint arXiv:2305.18593, 2023.

[2] Yin J, Qiao Y, Zhou Z, et al. Mcm: Masked cell modeling for anomaly detection in tabular data[C]//The Twelfth International Conference on Learning Representations. 2024.

[3] Durani W, Leiber C, Durani K, et al. Anomaly Detection by an Ensemble of Random Pairs of Hyperspheres[C]//The Thirty-ninth Annual Conference on Neural Information Processing Systems.

[4] Shenkar T, Wolf L. Anomaly detection for tabular data with internal contrastive learning[C]//International conference on learning representations. 2022.

---

> ### Author Rebuttal · Authors · 2026-03-31
>
> We thank the reviewer for the thoughtful and constructive comments. We address each concern below.
>
> ## W1: Multi-class or multi-modal normal data.
>
> DVM-AD has two structural properties that provide natural robustness to multimodal normal data: (i) The maximal-tail directions (κ ≈ 1) capture high-variance directions, which, for multimodal distributions, preserve multi-cluster geometry rather than collapsing it (unlike KNFST). This is why DVM-AD excels at cluster anomaly detection (best average rank in Figure 4a). (ii) Nearest-neighbor scoring assigns low anomaly scores to test samples near any normal mode, not only a centroid, making it inherently multimodal-aware (unlike DeepSVDD).
>
> Additionally, the two-tailed selection remains stable across all 47 datasets, including multimodal ones: the fallback never triggers (0/47) (details in our response to Reviewer ZLEQ, Q2). Empirically, our t-SNE of the "3_backdoor" dataset (d=196; Reviewer ZLEQ, Q4) shows the normal class forms multiple distinct clusters, yet DVM-AD achieves 98.48% AUROC. If sub-class labels were available, DVM-AD could be extended via multi-class Fisher discriminant analysis - an interesting direction for future work. We will add this analysis to §4.3.
>
> ## W2: Training time comparison with baselines.
>
> The reviewer correctly identifies that Table 2 reports DVM-AD training times without baseline comparisons. We note that Figure 15 (Appendix D.6.4) already provides the comparison the reviewer requests; it shows DVM-AD among the lower median training times across the 47 ADBench datasets. We will promote this result to §5.5.3.
>
> ## W3: 7:3 train/test split vs. 50/50 protocol.
>
> We follow the ADBench protocol (Han et al., 2022), a well-known tabular AD benchmark. The stratified 7:3 split preserves the natural anomaly ratio in the test set, reflecting realistic deployment conditions. In contrast, the 50/50 normal-only-train protocol concentrates all anomalies in a smaller test set, inflating the effective test anomaly ratio above the natural level, a key consideration for tuning-free deployment. Importantly, regardless of split ratio, our training set contains only normal samples - anomalies in the training split are removed, maintaining the one-class protocol. The key requirement is that the same protocol is applied consistently to all methods, which we ensure. Our multi-seed experiment (Table 8, Appendix D.7) with 5 random seeds shows stable performance, suggesting relative rankings are robust to split variation. We will add this motivation to §5.4.
>
> ## W4: Missing recent baselines.
>
> We have now run all requested methods and additional baselines under the same protocol: MCM (82.38% AUROC, rank 11.79), ICL (82.41%, rank 12.00), DTE (67.78%, rank 20.62), and ADERH (79.73%, rank 14.85) - the methods the reviewer specifically requested. Per-dataset results are in Table S3 of the supplementary. We will add a temporal scope statement to §5.2, noting the baseline set covers methods through early 2025, and a comparative discussion in §2.2, positioning DVM-AD against each method family.
>
> ## W5: No baselines in contamination/noise experiments.
>
> Thank you for this suggestion. We have now run KNN, IForest, and LUNAR under the same protocols (Appendix E.4–E.5). For noise, DVM-AD maintains the highest AUROC at all levels, with a gap of 3.42–4.02% over the best baseline (LUNAR). For contamination, DVM-AD again leads across all levels (AUROC gap of 2.69–4.02% over LUNAR). Updated Tables 12–13 with baseline comparisons are at supplementary, and we will incorporate this analysis into §5.5.3.
>
> ## W6: Proxy point and feature-wise maximum.
>
> We understand the concern - the feature-wise maximum (Eq. 5) may suggest DVM-AD assumes larger values indicate anomalies. This is not the case, and we will clarify this. The proxy serves solely as a geometric anchor to make one-class discriminant analysis well-defined; it is excluded from test-time scoring (§4.3).
>
> More critically, Proposition E.1 proves that Sw is completely independent of the proxy choice. The proxy affects only a rank-one component of the total scatter through δ, making no assumptions about the anomaly's location. Empirically, Table 10 confirms that the min-corner proxy achieves an AUROC of 90.02% vs 89.65% for max-corner - nearly identical, confirming that the proxy direction does not encode anomaly assumptions. We will add a plain-language clarification to §4.3 and promote Table 10 into the main text.
>
> ## Summary.
> All six concerns have been addressed: the multimodal and proxy questions (W1, W6) are resolved by existing structural properties of the framework; the protocol choice (W3) is methodologically justified; and we have run all requested baselines plus additional ones (W4), with DVM-AD outperforming all (W5 included). We respectfully ask the reviewer to consider raising their assessment in light of these responses and new results.
>
> **Supplementary link:** https://anonymous.4open.science/r/dvm_ad/

---

> > ### Author Rebuttal · Reviewer_VZkh · 2026-04-03
> >
> > I thank the authors for their responses.
> >
> > Regarding W2, could the authors provide time comparison results on  high-dimensional datasets and datasets with large number of samples?
> >
> > Regarding W3, it is difficult to understand why anomalous samples in training set are removed instead of used in test set. In this case, 70% of anomalous samples are dropped and the propose method is not evaluated on these samples. For this reason, I think the experimental setting is unreasonable and the evaluation is not complete.

---

> > > ### Author Response · Authors · 2026-04-07
> > >
> > > We thank the reviewer for the follow-up questions. We have completed all the requested experiments and provide the results below.
> > >
> > > ## [W2] Runtime comparison on high-dimensional and large-sample datasets.
> > >
> > > We extended the scalability analysis to two extreme synthetic regimes with 13 baselines (7 ML on CPU, 6 DL on GPU), using the same protocol as Table 2. Mean training times (in seconds) across 5 seeds are reported below. OoM = Out of Memory within 64GB RAM, 16GB VRAM.
> > >
> > > **Setting A - Large-sample (N=10⁶, d=10³):**
> > >
> > > | Method | Time (s) |
> > > |---|---|
> > > | **DVM-AD** | **32.5 ± 3.1** |
> > > | HBOS | 43.7 ± 0.8 |
> > > | KNFST | 48.2 ± 2.3 |
> > > | ECOD | 142.5 ± 0.3 |
> > > | COPOD | 156.0 ± 11.4 |
> > > | PCA | 397.3 ± 18.2 |
> > > | ContrastiveAD | 680.8 ± 1.4 |
> > > | ADERH | 845.9 ± 14.9 |
> > > | DTE | 1856.3 ± 714.2 |
> > > | MCM | 3403.4 ± 238.1 |
> > > | OCSVM, AutoEncoder, LUNAR, ALAD | OoM |
> > > |||
> > >
> > > **Setting B - High-dimensional (N=10⁵, d=10⁴):**
> > >
> > > | Method | Time (s) |
> > > |---|---|
> > > | KNFST | 38.0 ± 0.7 |
> > > | HBOS | 47.0 ± 0.7 |
> > > | DTE | 140.6 ± 5.6 |
> > > | COPOD | 145.6 ± 0.2 |
> > > | ECOD | 147.3 ± 4.1 |
> > > | **DVM-AD** | **496.0 ± 1.9** |
> > > | ADERH | 573.9 ± 1.8 |
> > > | PCA, ContrastiveAD, MCM, AutoEncoder, LUNAR, ALAD, OCSVM | OoM |
> > > |||
> > >
> > > These results align with the complexity of DVM-AD (Table 1). When N dominates (Setting A), DVM-AD is the fastest method - all DL baselines on GPU are 21×–105× slower, and four methods fail entirely. When d is extreme (Setting B), DVM-AD's O(d³) eigendecomposition cost becomes visible (496s), but it remains operable where 7 of 13 baselines fail, including all deep methods except DTE. Combined with Figure 15 (DVM-AD among the faster on the ADBench scale), this provides a complete picture of efficiency. We will incorporate the extended timing table into §5.5.3.
> > >
> > > ## [W3] Anomaly removal in 7:3 split.
> > >
> > > We have completed the full experiment the reviewer requested: 50% of normal samples used for training; all remaining normal samples combined with **all** anomalous samples placed in the test set, ensuring every anomaly in every dataset is evaluated. We ran 32 methods across all 47 ADBench datasets.
> > >
> > > Results (average across 47 datasets, all 32 methods):
> > >
> > > | Method | AUROC (%) | Rank |
> > > |---|---|---|
> > > | **DVM-AD** | **88.81** | **3.30 (1st)** |
> > > | LUNAR | 86.11 | 6.66 |
> > > | KNN | 83.85 | 8.13 |
> > > | MCM | 81.00 | 10.43 |
> > > | ContrastiveAD | 81.55 | 11.63 |
> > > | ADERH | 79.11 | 13.30 |
> > > | DTE | 68.15 | 17.83 |
> > > ||||
> > >
> > > DVM-AD ranks 1st among the 32 methods, with a substantial gap to the second-best, LUNAR (average rank 6.66). Supplementary Tables 3–4 present ranks recomputed separately for the original baselines (DVM-AD rank average 2.98, consistent with the original ranking) and the 4 reviewer-requested baselines (DVM-AD rank 1.45). Compared with the 7:3 protocol (89.65% AUROC), the drop is only 0.84 pp, consistent with the use of less normal training data, while the ranking is preserved. The full results are presented in Tables 2-4 in the supplementary link.
> > >
> > > We briefly clarify the protocol concern directly: 70% of anomalies fall into the training partition under the 7:3 split, and these are not evaluated because the training set contains only normal samples - this is a standard practice for one-class evaluation under any protocol. For instance, DAGMM (Zong et al., ICLR 2018) and Deep Orthogonal Hypersphere Compression (Zhang et al., ICLR 2024) use 50/50 splits but likewise discard anomalies from the training partition. Crucially, the removed anomalies are drawn from the same distribution as those retained in the test set - they are not a distinct subpopulation. Our multi-seed experiment (Table 8, 5 seeds) confirms this: rankings remain stable across stratifications, and across 5 random splits, the union of tested anomalies covers substantially more than any single 30% partition. The 7:3 split is well-established: it follows ADBench (Han et al., NeurIPS 2022) and is adopted by ADERH (Durani et al., NeurIPS 2025) - one of the reviewer's own suggested references.
> > >
> > > The reviewer's suggested protocol (50-50 split, all anomalies used for testing) is another approach used in some related works, and we have now completed the full experiment using it. Both protocols yield the same conclusion: DVM-AD ranks first. We will add the 50/50 results as a new appendix with per-dataset breakdown, referenced in §5.4.
> > >
> > > ## Summary.
> > >
> > > With all requested experiments now completed, runtime comparisons at extreme scales for W2, and full anomaly coverage evaluation for W3, our proposed DVM-AD ranks first under both protocols against all 32 methods, confirming the robustness of our results across evaluation settings.
> > >
> > > **Supplementary tables (W2 timing + W3 full results):** https://anonymous.4open.science/r/dvm_ad/
> > >
> > > **New references:**
> > > - Zong et al. Deep Autoencoding Gaussian Mixture Model for Unsupervised Anomaly Detection. ICLR, 2018.
> > > - Zhang et al. Deep Orthogonal Hypersphere Compression for Anomaly Detection. ICLR, 2024.

---

### Official Review · Reviewer_v1Sq · 2026-03-11

**Soundness:** 3
**Presentation:** 2
**Significance:** 3
**Originality:** 3
**Overall Recommendation:** 4
**Confidence:** 4

**Summary:**

This paper studies the problem of unsupervised (semi-supervised) anomaly detection in tabular data, where only normal data is used for training. This paper is based on Linear Discriminant Analysis, which is a very classical statistical technique. Their approach aims to learn a discriminant space in which normal data becomes compact while simultaneously preserving the informative global structure of the training set. To this end, they first construct a two-class dataset (in which normal data is given and they augment it with anomalies), then learn a projection into a discriminant space, and finally they score the test instance via its distance to the nearest neighbour in the discriminant space. By doing this, they obtain a method which does not need hyper-parameter tuning, having a computing complexity of $O(Nd^2+d^3+Ndm) $ ($N$ is number of samples, $d$ is the number of dimension) for training. They conduct experiments on 47 datasets from the ADBehch suite, and compare their method with 28 baselines from the PyOD library.

**Compliance With Llm Reviewing Policy:**

Affirmed.

**Final Justification:**

My main concerns are solved and I decided to raise my overall ratings.

**Key Questions For Authors:**

I have no questions, please check the weak points. I am willing to raise my ratings if my concerns are well addressed.

**Limitations:**

yes

**Strengths And Weaknesses:**

**String points**

1. The idea is interesting (selecting directions form both extremes of the bounded spectrum is a nice perspective).

2. The method is solid in math (the motivation is very clear, and they also show some nice analytical properties of their method).

3. They conduct many experiments (e.g., 47 + 10 datasets, with 28 baselines, on two metrics).

**Weak points**

1. The presentation of the paper can be largely improved. To be specific, the abstract and introduction are very hard to follow. Many technical terms are used without any explanations (the authors may assume the readers know everything at the beginning). Even after reading the abstract and introduction, it is difficult to figure out the main idea of this paper. (I have been working on anomaly detection for more than 6 years and am familiar with most work in this direction, but it is still very difficult for me to follow at the beginning)

2. Some important references are missing. In the related work section, the authors omitted an important family of anomaly detection methods, *generative based anomaly detector*, including GANomaly [1] based on GANs, DTE [2] based on Diffusion models, TCCM [3] based on Flow Matching; I'd like to see discussions on these methods and the authors should consider them as baselines in the experiment section.

3. Some claims may not be true. For example, the authors claims that
>"deep AD methods are not consistently superior to simpler classical detectors on tabular benchmarks (Shwartz-Ziv & Armon, 2022)"

and they cite (Shwartz-Ziv & Armon, 2022)[4] as the evidence; However, the paper of(Shwartz-Ziv & Armon, 2022)[4] only investigated classification and regression problems; They did not investigate anomaly detection at all. Therefore, they utilize wrong evidence to support a potential untrue claim; Moreover, in both DTE[2] and TCCM[3], they show that these deep methods (DTE, TCCM, LUNAR) often (if not always) outperform existing shallow AD methods on ADBench datasets. (All of these methods I mentioned are open-sourced, so I'd like to see more experiments on these competitive baselines)

4. Some important terms/concepts are not defined: the authors should formally define each type of anomalies they investigate to make their paper self-contained.

5. The complexity of the training of their method is very high. To be specific, a complexity of $O(Nd^2+d^3+Ndm)$ makes the method less practical for large and/or high-dimensional datasets. However, the authors still claim their method as (in page 7)
>""...with particularly fast training time..."

**References**

[1] Akcay, Samet, Amir Atapour-Abarghouei, and Toby P. Breckon. "Ganomaly: Semi-supervised anomaly detection via adversarial training." ACCV, 2018.

[2] Livernoche, Victor, et al. "On Diffusion Modeling for Anomaly Detection." ICLR, 2024.

[3] Li, Zhong, et al. "Scalable, Explainable and Provably Robust Anomaly Detection with One-Step Flow Matching." NeurlPS, 2025.

[4] Shwartz-Ziv, Ravid, and Amitai Armon. "Tabular data: Deep learning is not all you need." Information fusion 81 (2022): 84-90.

---

> ### Author Rebuttal · Authors · 2026-03-31
>
> We sincerely thank the reviewer for the detailed and precise feedback. The reviewer's expertise in anomaly detection makes their concerns valuable. We are especially grateful for identifying the citation error in W3 - it is the most important correction in this rebuttal.
>
> ## W1 and W4: Presentation and anomaly type definitions.
>
> We appreciate the reviewer's patience. As detailed in our response to Reviewer Tjgk (W1-W4), we have a concrete plan with four components:
>
> - (a) **Abstract and introduction:** rewrite so that the core insight, compressing the normal class and preserving its internal structure without dataset-specific tuning, is stated plainly before any technical vocabulary, replacing jargon with accessible descriptions. A full audit of §1 will replace every term requiring §4 background.
> - (b) **Anomaly type definitions (W4):** add formal definitions for local, global, cluster, and dependency anomalies at first mention following Han et al. (2022). Detailed descriptions are already in Appendix D.3.
> - (c) **Methodology structure:** restructure §4 into clearly delineated subsections - §4.0 Overview (pipeline figure), §4.1 Background, §4.2 Derivation, §4.3 Algorithm - so that motivation, derivation, and implementation are no longer mixed.
>
> ## W2: Missing generative-based AD methods.
>
> We agree Related Work should cover generative AD more explicitly. GAN-based methods - SO-GAAL, ALAD, and AnoGAN - are already included as baselines (§5.2). We will add a dedicated paragraph in §2.2 tracing the evolution of generative AD from GAN-based methods (GANomaly) through diffusion-based approaches (DTE) to recent masked modeling (MCM) and flow-matching (TCCM), and positioning DVM-AD within this landscape. We have run DTE, TCCM, and GANomaly under the same protocol. DVM-AD outperforms all: DTE 67.78% AUROC (rank 20.62), TCCM 81.76% (rank 11.62), GANomaly 78.95% (rank 15.45), vs DVM-AD 89.65% (rank 3.53). Per-dataset results at Table S3 https://anonymous.4open.science/r/dvm_ad/.
>
> ## W3: Misplaced citation of Shwartz-Ziv & Armon (2022).
>
> **(1) Citation correction.** The reviewer is correct: Shwartz-Ziv & Armon (2022) study classification and regression, not anomaly detection. We correct this error unreservedly. The evidence is already in our paper: Han et al. (2022, ADBench) conclude that none of the unsupervised algorithms is better than others; Bouman et al. (2024) further show that the best-performing algorithm varies by anomaly type. We will replace the citation accordingly.
>
> **(2) Claim alignment.** We believe our position and the reviewer's observation are compatible. The reviewer notes that deep methods "often (if not always) outperform" shallow methods, and our claim is that they are "not consistently superior." Both statements can coexist: deep methods may win on a majority of datasets while still failing to dominate universally. Our Figures 2 and 4 illustrate this: the top methods interleave deep and shallow, with no method consistently outperforming across the 47 datasets. We will revise the phrasing to make this nuance explicit.
>
> **(3) DTE and TCCM experiments.** As requested, we have run both DTE (diffusion-based) and TCCM (flow-matching-based) under the same protocol. DTE ranks 20.62 (67.78% AUROC) and TCCM ranks 11.62 (81.76%), compared to DVM-AD's rank 3.53 (89.65%), consistent with the shared view that deep methods do not consistently dominate. We will position these findings in §2.2 as suggested in W2.
>
> ## W5: Training complexity and "fast training" claim.
>
> The reviewer correctly identifies an apparent contradiction. The two statements describe different regimes: the complexity analysis is a theoretical characterization; the "fast training" claim is an empirical observation on benchmark datasets. Tabular data typically has moderate dimensionality - most ADBench datasets have d<200, with a maximum of d=1555 - meaning the O(d³) eigendecomposition rarely dominates in practice: Table 2 shows training about 32.5s for d≤10³ and N≤10⁶. Within this regime, DVM-AD's structural advantage - one pass to accumulate scatter, one eigendecomposition, no iterative optimization - makes it genuinely faster: Figure 15 shows DVM-AD among the lower median training times. Additionally, for high-d, PCA pre-reduction at 0.10d (on d=768 NLP embeddings) costs only -1.38% AUROC with 15.5x speedup (details in our response to Reviewer ZLEQ, Q3). We will scope the claim more precisely and clarify the O(d³) bottleneck in §4.4.
>
> ## Summary.
> The citation error (W3) is corrected unreservedly; the underlying claim aligns with the reviewer's own observation that deep methods perform well "often (if not always)" but not universally, confirmed by our experiments. The presentation (W1, W4) has a concrete three-component plan, and we will scope the training claim to the benchmark regime (W5). We are confident these revisions address the concerns and respectfully ask the reviewer to consider them when reassessing our work.

---

> > ### Author Rebuttal · Reviewer_v1Sq · 2026-04-02
> >
> > My main concerns are solved and I decided to raise my overall ratings. I hope the authors will incorporate the promised changes in the revised version.

---

### Official Review · Reviewer_ZLEQ · 2026-03-12

**Soundness:** 4
**Presentation:** 2
**Significance:** 3
**Originality:** 4
**Overall Recommendation:** 4
**Confidence:** 4

**Summary:**

**Summary：**This paper proposes DVM-AD, a tuning-free one-class discriminant learning framework for tabular anomaly detection. By framing the task as a generalized eigenvalue problem stabilized via Moore-Penrose pseudo-inverse, the method use a two-tailed spectral selection strategy to keep class compactness and preserve geometric structure. Extensive benchmark evaluations against baselines shows that DVM-AD outperforms them and demonstrates robustness across four anomaly types.

**Compliance With Llm Reviewing Policy:**

Affirmed.

**Final Justification:**

I will remain my original score.

**Key Questions For Authors:**

- A more detailed analysis of where the performance improvements come from would be helpful. Specifically, experiments with an ablation study on the role of Proxy Point would more clearly.
- The distribution of the number of selected discriminant vectors (m) across the 47 tabular datasets may help clarify the two-tailed selection is consistent across different domains.
- For tabular data with extremely large sample sizes or very high dimensions, would incorporating prototypes for normal samples or applying dimensionality reduction as a pre-processing step help mitigate the computational burden? Exploring the trade-off between such approximations and detection accuracy for large dataset would be valuable.
- More t-SNE visualizations comparing the original feature space with the DVM-AD discriminant space across multiple datasets would visually confirm the compactness and structure balance mentioned in the method.
- It would be useful to compare the performance of DVM-AD with the latest reconstruction-based or representation-learning methods.

**Limitations:**

yes

**Strengths And Weaknesses:**

**Strength And  Weakness**

Strengths:

- The method is tuning-free, requiring no data-specific hyperparameter adjustment beyond a fixed numerical tolerance. This makes it suitable for unsupervised deployment.
- The approach provides a simple, closed-form solution derived from discriminant analysis and provide a rigorous theoretical proof.
- The experimental study is comprehensive, using the ADBench and analyze four distinct anomaly types, while also exploring the effects of anomaly contamination and varying noise levels.
- The paper provides a detailed table of computational complexity and reports training times across varying data scales.

Weaknesses:

- The training overhead grows higher when training data with a large number of samples ($N$) or high feature dimensionality ($d$), which could limit its application on large datasets.
- The analysis of why the learned projections improve separability is difficult to understand.  In the paper, only Figure 16 shows DVM-AD representations. More intuitive evidence or a qualitative analysis across different data distributions would help to understand.
- Although paper compares with 28 baselines, it doesn't include some latest methods in tabular anomaly detection, such as NPT-AD[1], Retrival method[2], MCM[3], DTE[4], PTAD[5].

[1] Beyond Individual Input for Deep Anomaly Detection on Tabular Data
[2] Retrieval Augmented Deep Anomaly Detection for Tabular Data
[3] MCM: Masked Cell Modeling for Anomaly Detection in Tabular Data
[4] On Diffusion Modeling for Anomaly Detection
[5] PTAD: Prototype-Oriented Tabular Anomaly Detection via Mask Modeling

---

> ### Author Rebuttal · Authors · 2026-03-30
>
> We thank the reviewer for the thorough and constructive feedback. Our response below.
>
> ## Q1: Performance attribution and Proxy Point ablation.
>
> **Performance attribution.** As a closed-form method, DVM-AD has few design choices, making attribution direct. The dominant contributor is the two-tailed spectral selection (Table 9): compared to max-tail-only, 2-tail recovers +4.81 AUROC overall, with the largest gains on local (+4.75 AUROC) and dependency (+4.95 AUROC), while cluster and global gains are smaller (+1.13/+1.46 AUROC) since these are partially captured by a single tail. This confirms the compactness–structure mechanism in Section 4.2. The tolerance εsel shows limited sensitivity across magnitudes (Table 11, Appendix E.3).
>
> **Proxy ablation.** The proxy is a structural necessity - without it, κ(θ) = 1 for all θ, yielding a flat spectrum with no discriminant information - but its specific construction has minimal impact. Since Sb is rank-one and Sw is completely independent of the proxy, robustness is a property of the framework. Table 10 confirms: three constructions yield AUROC 89.65%, 90.02%, 89.44% with no strategy dominating. The full ablation is in Appendix E.2; we will add a summary in the main text
>
> ## Q2: Distribution of selected DVs (m).
>
> Across 47 datasets: the fallback never triggers (0/47); every dataset selects at least one max-tail direction (47/47); 7 datasets also select min-tail directions (7/47). The median selected fraction m/d = 0.926 indicates that selection acts as targeted filtering of mid-spectrum directions rather than aggressive dimensionality reduction. m varies predictably with scatter matrix rank, reflecting intrinsic data complexity. This confirms the two-tailed rule is stable without per-dataset tuning. Full Table S1 is at supplementary and will be added to the Appendix.
>
> ## Q3: Pre-processing for large datasets.
>
> **Large sample sizes.** DVM-AD scales linearly in N: O(Nd²) for scatter, O(Ndm) for projection (Table 1). Table 2 confirms this: at d=10³, training takes 0.37s at N=10³ and 32.5s at N=10⁶, showing near-linear growth. These results indicate that DVM-AD can handle large-sample-size datasets without compression, though prototypes could further reduce cost at extreme N.
>
> **High dimensions.** The O(d³) eigendecomposition is the computational bottleneck, though for typical tabular data (d<200 in most ADBench datasets), this completes in under one second (Table 2). For the high-d regime, we ran PCA pre-reduction (as suggested) on 5 NLP embedding datasets (d=768). At 0.50d, -0.40% AUROC with 2.5x speedup; at 0.25d, -0.61% with 6.3x; at 0.10d, -1.38% with 15.5x. This is consistent with the analysis in Section 4.2: max-tail directions are dominated by high-variance normal structure, so PCA preserves the geometry DVM-AD relies on. Full results in Table S2 supplementary.
>
> ## Q4: Visualizations of projection improvement.
>
> **t-SNE visualizations.** We have added before-and-after t-SNE comparisons at supplementary. On "3_backdoor" (98.48% AUROC), the effect is: after min-tail, normals collapse into a compact ball with anomalies ejected; after max-tail, cluster structure is preserved, but anomalies remain mixed; the 2-tail default combines both, directly visualizing the compactness–structure mechanism. We also include "39_vertebral" (57.39% AUROC, Table 8) as a harder case where anomalies remain partially embedded among normals, demonstrating that when discriminant information is limited (d=6), separation is weaker.
>
> **NN distance histograms.** We also provide NN distance histograms that measure separability in the metric DVM-AD actually uses for scoring. On "10_cover," normal and anomalous NN distances completely overlap in original space but clearly separate after projection; on "30_satellite," projection shifts anomalies into a distinct right tail. The consistent gain over KNN (same scoring, original space) across all 47 datasets confirms the projection enhances separability.
>
> ## Q5: Compare with suggested methods.
>
> We have run all suggested methods under the same default-setting protocol: NPT-AD, RAAD, MCM, DTE, and PTAD. DVM-AD (AUROC 89.65%) outperforms all baselines: NPT-AD (78.1%), RAAD (79.31%), MCM (82.38%), DTE (67.78%), and PTAD (80.56%). Per-dataset results are presented in Table S3 supplementary.
>
> ## Summary.
> The two-tailed selection is the dominant performance driver (Q1), while the proxy (Q1) and DV distribution (Q2) are stable across datasets. DVM-AD scales linearly in N and is compatible with PCA for high-d regimes (Q3). New t-SNE visualizations, including a failure case and NN histograms provide both visual and quantitative evidence of separability (Q4). DVM-AD outperforms all extended baselines (Q5). We are confident these additions strengthen the paper without altering its core technical contributions. We respectfully ask the reviewer to consider these responses when reassessing our work.
>
> **Supplementary link:** https://anonymous.4open.science/r/dvm_ad/

---

> > ### Author Rebuttal · Reviewer_ZLEQ · 2026-04-03
> >
> > Thanks for your reply.

---

### Official Review · Reviewer_Tjgk · 2026-03-13

**Soundness:** 3
**Presentation:** 1
**Significance:** 2
**Originality:** 2
**Overall Recommendation:** 4
**Confidence:** 4

**Summary:**

This paper focuses on the task of tabular anomaly detection. The paper proposes DVM-AD, a method motivated by discriminant learning. This method constructs a pseudo class by creating a deterministic reference point based on the training set, and then learns a space that separates the normal and pseudo class.

**Compliance With Llm Reviewing Policy:**

Affirmed.

**Final Justification:**

The authors promise that they will polish the paper, thus I decide to increase the score accordingly. But the authors must refine the paper as discussed, and incorporate the added results in W5 to the final revision.

**Key Questions For Authors:**

Please see the weaknesses above.

**Limitations:**

Yes.

**Strengths And Weaknesses:**

Strengths:

S1: The studied problem, i.e., tabular anomaly detection is important in real world application.
S2: The idea is interesting and requires no parameter fine-tuning.
S3: The paper provides theoretical analysis.

Weaknesses:
W1: Unclear of the motivation. This issue occur in the second and third paragraph of the introduction.
First, without detailed context, readers do not know the definition of different anomaly types like local, global, cluster, and dependency. And authors need to further explain why these two branches of existing methods will cause the sensitivity to different types of anomalies.
The authors mean when achieve normal class compactness, it is hard to preserve structure.  Are these two definitions opposite, or no methods consider these two characteristics at the same time? This is about the starting of the motivation, which is important for better understanding towards the paper.
Furthermore, there lacks of theoretical or empirical evidence that the existing methods really lacks one or both of the two characteristics.

W2: Introduction and abstract contain too much professional words, which are hard to understand. Some important concepts are introduced before enough context is given. For example, notions such as the minimal tail appear too early.

W3: The methodology section is hard to understand. The paper mixes theoretical motivation, derivation, and concrete algorithmic steps too tightly, which makes it difficult for the reader to first grasp the overall pipeline. A clearer method overview before entering the derivation would substantially improve readability.

W4: The transition from the Fisher criterion to the generalized eigenvalue problem Sbθ = μSwθ  is introduced without definition, and the purpose of the Fisher criterion is stated but its connection to the subsequent eigenproblem is omitted. This forces the reader to either be an expert in classical LDA or to spend significant effort reconstructing the standard derivation. Similarly, any other section on mathematical derivation should proceed gradually to facilitate better understanding for the reader.

W5: The related work section, appears limited and does not sufficiently position the paper against more recent relevant studies, like DRL [1], MCM [2].

[1] DRL: Decomposed Representation Learning for Tabular Anomaly Detection. ICLR 2025

[2] MCM: Masked Cell Modeling for Anomaly Detection in Tabular Data. ICLR 2024

W6: DVM-AD makes one-class discriminant learning effective via a deterministic reference point. How to ensure the robustness of this choice?

---

> ### Author Rebuttal · Authors · 2026-03-30
>
> We thank the reviewer for the thorough feedback and address each weakness below. The core concerns are presentational: theory and experiments support the claims, but we take responsibility for not making them accessible enough.
>
> ## W1: Motivation clarity.
>
> We agree that §1 Introduction asks readers to accept a complex motivation without sufficient context, and will revise §1 along four lines.
>
> **Definitions.** We will add definitional sentences at first mention: local anomalies deviate from their immediate neighborhood; global anomalies lie in low-density regions; cluster anomalies form coherent groups shifted from normal clusters; dependency anomalies violate cross-feature correlations while remaining individually plausible. We will add a reference to Appendix D.3.
>
> **Causal mechanism.** Compactness-focused methods (KNFST, PMKFN) find directions where θ⊤Swθ ≈ 0, collapsing normal data and amplifying pointwise deviations (local/global) but destroying covariance geometry needed for cluster and dependency anomalies. Structure-preserving methods (PCA) retain principal variance but are insensitive to low-variance subspaces where local anomalies hide. We will make this explicit in §1.
>
> **Are they mutually exclusive?** Not geometrically impossible, but existing formulations optimize only one end of κ(θ) = θ⊤Swθ/θ⊤Ssθ. Some methods balance both via hyperparameters (Perera & Patel, 2019; Hojjati & Armanfard, 2023), but tuning without anomaly labels is problematic (Ding et al., 2022). Theorem 4.1 shows κ is bounded in [0,1], hence both optimized directions become accessible under a fixed configuration - DVM-AD's key insight. We will add one sentence to §1 to state this distinction and clarify our motivation.
>
> **Empirical evidence.** Figure 4 confirms measurable blind spots: KNFST ranks 12.68 for dependency (Figure 4c), and PCA ranks 12.06 for local (Figure 4d). No baseline achieves the best rank in all types. Figure 5 in ADBench (Han et al., 2022) corroborates this pattern. We will promote these results in §1.
>
> ## W2: Terminology too early.
>
> The abstract introduces terms ("minimal tail," "bounded inverse-scatter spectrum") that require background from §4. We will rewrite the abstract in plain language (replacing "bounded inverse-scatter spectrum" with "a projection that captures both compactness and structural geometry") and defer formal terms to §4.2. A full audit of §1 will replace such term with a plain-language description plus a forward reference.
>
> ## W3: Methodology structure.
>
> We will restructure §4 to separate motivation, derivation, and implementation into clearly delineated subsections: §4.0 Overview (informal pipeline: proxy construction → two-tailed spectrum selection → NN scoring; and forward references); §4.1 Background (LDA foundations); §4.2 Derivation (motivation, derivation, and interpretation as labeled steps); §4.3 Algorithm (Algorithms 1–2 with step-by-step commentary).
>
> ## W4: Mathematical transitions.
>
> We will add a two-line derivation: differentiating f(θ) = θ⊤Sbθ/θ⊤Swθ and setting the gradient to zero yields Sbθ = μSwθ, with a reference to (Ghojogh et al., 2019) and a plain-language sentence: "maximizing the Fisher ratio is equivalent to finding directions that best separate the two classes." Beyond this, we have audited §4 and identified additional transitions to address:
>
> - (i) How f(θ) connect κ(θ);
> - (ii) Justification for restricting to Range(Σ);
> - (iii) Connecting Theorem 4.1 eigenvalues to compactness/structure;
> - (iv) Grounding dis_max in Eq. (9).
>
> ## W5: Enhance the Related Work.
>
> We first clarify: DRL-AD is already a baseline (Figures 2–5), and MCM is cited in §1. We will add a comparison on three axes: (i) anomaly type coverage: DRL targets global, MCM captures inter-feature correlations, DVM-AD handles all four; (ii) hyperparameter requirements: DRL/MCM need dataset-specific tuning, DVM-AD uses a fixed configuration; (iii) data modality: DVM-AD additionally covers NLP/CV embeddings. We have run MCM; DVM-AD outperforms by +7.27% in AUROC. Per-dataset results at Table S3 https://anonymous.4open.science/r/dvm_ad/
>
> ## W6: Proxy robustness.
>
> **Theoretically (Proposition E.1):** the proxy is a single point, so Sb = β δδ⊤ is rank-one; Sw is completely independent of the proxy. This means robustness is a property of the framework, not the specific construction.
>
> **Empirically (Table 10):** three constructions - max-corner, min-corner, farthest-from - yield AUROC 89.65%, 90.02%, 89.44% with no strategy dominating. These results will be promoted from the appendix to §4.3. Additionally, algorithm 1's fallback (Steps 6–8) serves as a robustness mechanism against near-degenerate configurations.
>
> ## Summary.
>
> We are confident the proposed revisions address the reviewer's concerns. The core issues are presentational rather than fundamental. We respectfully ask the reviewer to consider these responses when reassessing our work.
>
> **References:** All references are cited in the paper

---

> > ### Author Rebuttal · Reviewer_Tjgk · 2026-04-03
> >
> > Many thanks to the authors' response. The authors promise that they will polish the paper, thus I decide to increase the score accordingly. But the authors must refine the paper as discussed, and incorporate the added results in W5 to the final revision.

---

### Decision · Program_Chairs · 2026-04-30

**Decision:**

Accept (regular)

**Comment:**

Based on the reviews, rebuttal, and follow-up discussion, I recommend weak acceptance, consistent with the reviewers’ consensus. Reviewers agreed that the paper is technically sound and presents a useful, tuning-free one-class method for tabular anomaly detection, with a clear core idea, supporting theory, and strong results on broad benchmarks. The rebuttal addressed the main concerns by clarifying the proxy construction, correcting a citation issue, adding more recent baseline comparisons, and showing that the method remains top-ranked under the reviewer-requested 50/50 protocol. Several reviewers found the abstract, introduction, and derivations hard to follow, so the final version should improve clarity, define anomaly types earlier, and present the method more accessibly. The efficiency claims should also be scoped more carefully, since the method can become costly in very high-dimensional settings. Overall, this is a solid contribution whose remaining weaknesses are important but appear fixable in revision.